# IDENTIFIABILITY GUARANTEES IN TIME SERIES REPRESENTATION VIA CONTRASTIVE SPARSITY-INDUCING

## ABSTRACT

Time series representations learned from high-dimensional data, often referred to as "disentanglement" are generally expected to be more robust and better at generalizing to new and potentially out-of-distribution (OOD) scenarios. Yet, this is not always the case, as variations in unseen data or prior assumptions may insufficiently constrain the posterior probability distribution, leading to an unstable model and non-disentangled representations, which in turn lessens generalization and prediction accuracy. While identifiability and disentangled representations for time series are often said to be beneficial for generalizing downstream tasks, the current empirical and theoretical understanding remains limited. In this work, we provide results on identifiability that guarantee complete disentangled representations via Contrastive Sparsity-inducing Learning, which improves generalization and interpretability. Motivated by this result, we propose the `TimeCSL` framework to learn a disentangled representation that generalizes and maintains compositionality. We conduct a large-scale study on time series source separation, investigating whether sufficiently disentangled representations enhance the ability to generalize to OOD downstream tasks. Our results show that sufficient identifiability in time series representations leads to improved performance under shifted distributions. Our code is available at https://anonymous.4open.science/r/TimeCSL-4320.

## 1 INTRODUCTION

Time series representation learning has been proposed as a solution to the lack of robustness, transferability, systematic generalization and interpretability of current downstream task methods. However, the problem of learning meaningful representation for time series is still open. This problem is strongly related to learning *disentangled* representations pointed by Bengio et al. (2013). Informally, a representation is considered *disentangled* when its components are in *one-to-one* correspondence with natural and interpretable factors of variations. However, many works have studied the theoretical conditions under which disentanglement is possible from the point of view of identifiability. It has its origins in work on nonlinear independent analysis (ICA) (Comon, 1994; Hyvarinen & Morioka, 2017; Hyvarinen et al., 2019; Khemakhem et al.,

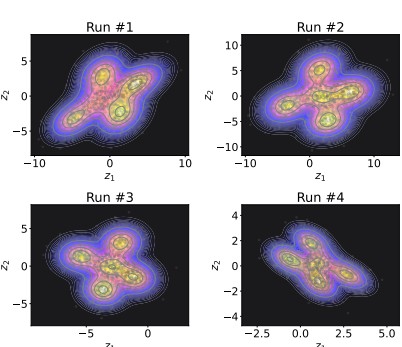

Figure 1: Recovered 5 slots latents for 4 runs of `TimeCSL` on UKDALE dataset.

2020b), which aims to recover independent latent factors from mixed observations. It has been found in (Locatello et al., 2019; Van der Maaten & Hinton, 2008; Dittadi et al., 2021; Montero et al., 2021; Lachapelle et al., 2022) that without exploiting an appropriate class of assumptions in estimation, the latent variables are not identifiable in the most general case. Existing methods like Generalized Contrastive Learning (GCL) via an auxiliary variable (Hyvarinen et al., 2019), HM-NLICA (Hälvä & Hyvärinen, 2020), Permutation Contrastive Learning (PCL) (Hyvärinen & Morioka, 2017), and SlowVAE (Klindt et al., 2021) rely on the assumption of mutually independent sources in the data generation process. However, this assumption breaks down for time-lagged or dependent latent variables, distorting identifiability. SlowVAE assumes linear relationships, while Temporally Disentangled Representation Learning (TDRL) (Yao et al., 2022) optimizes mutual

Figure 2: **Multi-view motivating setting for the *energy time series representation*.** **Left:** We consider $\{1, 2, 3, 4\}$ sources and $\{5\}$ representing measurement **noise** or other irrelevant sources. The mixed observation at different time are: $\mathbf{x}$ includes $\{1, 2, 4, 5\}$, and $\mathbf{x}'$ includes $\{2, 3, 4, 5\}$. **Center:** Training distribution combinations. **Right:** compositional consistency for OOD based recombining inferred latent slots $(\hat{\mathbf{z}}, \hat{\mathbf{z}}')$ allows for generalization, thus improving downstream tasks.

information between input and latent factors, penalizing static-dynamic interactions, and assumes only time-lagged influences. This requires matching the temporal resolution of observations and latent variables (Yao et al., 2022). A more flexible framework is needed to deal with real-world time series (*e.g.*, energy separation), where sources are often dependent, may be correlated in a general nonstationary environments with time-varying relations. Prior work on sparsity through convex optimization with sparsity-inducing norms (Bach et al., 2012) and recent findings in disentanglement using sparse task predictors (Lachapelle et al., 2023a; 2022) show impressive results empirically. An interesting question is whether these sparsity can guarantee identifiability, and resulting in disentangled representations that capture meaningful features and remain stable under distribution shifts? Indeed, without identifiability, the representation of a model can be unstable and not consistent (Locatello et al., 2019; Lenc & Vedaldi, 2015), in the sense that retraining the same model under small perturbations of the data or hyperparameters may result in wildly different representations. More formally, identifiability means that the parametrization of the model is injective (Roeder et al., 2021; Khemakhem et al., 2020b). Self-Supervised Learning (SSL) methods, known for their flexibility and efficiency (Liu et al., 2021), have improved supervised tasks via unsupervised learning, with early nonlinear ICA work unintentionally using SSL (Hyvärinen & Morioka, 2016; Bai et al., 2021a; Hyvärinen et al., 2023). However, as these methods are not always probabilistic, identifiability can be uncertain, although uniqueness is defined more broadly.

In this work, we combine SSL with probabilistic modeling and sparsity to achieve identifiability for time series representation up to affine transformations—essentially, disentangled representation for time series via Contrastive Sparsity-inducing Learning (TimeCSL) (see Fig. 1, across 4 runs, the latents are recovered, providing evidence of the latent space recovery up to the affine transformations). Importantly, this can be achieved with commonly adopted weaker assumptions. Specifically, we allow for statistically dependent latent factors, with empirical evidence indicating that relaxing independence improves OOD generalization (Roth et al., 2023; Oublal et al., 2024). Moreover, it requires no complete auxiliary data, handles nonlinear predictors and latent relationships for time series, and reduces reliance on labeled data via contrastive learning. Our contributions include:

[1] We rely on the sparsity assumption of time series representation, and provide theoretical insight and empirical arguments on how, and under which condition, identifiability up to affine transformation is preserved. We show that TimeCSL outperform an affine transformation *e.g.*, permutation and element-wise transformation.

[2] Unlike many existing identifiability results, we allow for arbitrary dependencies without parametric assumptions, achieving slot latent disentanglement through *Partial Selective Pairing*. This approach is particularly suitable for time series, where obtaining fully labeled data can be challenging.

[3] Building on this result, we propose generalization consistency for uncommon OOD correlations as in Fig. 2. We validate it by showing that TimeCSL effectively disentangles latent slots in real-world source separation tasks (*e.g.*, energy disaggregation). Notably, existing architectures

(*e.g.*, D3VAE, RNN-VAE) improve by **+11%** `RMSE` in downstream tasks with disentangled representations. We also release over 221 trained models as baselines for future research[1].

**Notation** Vectors and vector-valued functions are denoted by bold letters. Vectors with factorized dimensionality, such as the latent variable $\mathbf{z} \in \mathbb{R}^{d_{\mathcal{Z}}}$, where the latent space $\mathcal{Z}$ has dimension $d_{\mathcal{Z}} = d \times n$, or functions with factorized outputs, like the encoder $\boldsymbol{f}_\phi \colon \mathcal{X} \to \mathbb{R}^{2d_{\mathcal{Z}}}$, where $\boldsymbol{f}_\phi(\mathbf{x}) = [\boldsymbol{\mu}_\phi(\mathbf{x}), \boldsymbol{\sigma}_\phi(\mathbf{x})]^\top$, are used in this context. We refer to $(\boldsymbol{f}_\phi, \boldsymbol{g}_\theta)$ as the ground truth encoder-decoder, and $(\hat{\boldsymbol{f}}_\phi, \hat{\boldsymbol{g}}_\theta)$ as the learned encoder-decoder, and $\hat{\mathbf{z}} := \{\hat{\boldsymbol{z}}_1, \ldots, \hat{\boldsymbol{z}}_n\}$ is the learned representation of $\mathbf{z} := \{\boldsymbol{z}_1, \ldots, \boldsymbol{z}_n\}$. When indexing with $k$, we refer to the $k$-th contiguous sub-vector, such as the learned slot latent $\hat{\boldsymbol{z}}_k := \hat{\boldsymbol{\mu}}_{\phi\,k}(\mathbf{x}) + \hat{\boldsymbol{\sigma}}_{\phi\,k}(\mathbf{x}) \odot \boldsymbol{\epsilon}$, where $\boldsymbol{\epsilon} \sim \mathcal{N}(0, \mathbf{I})$, and both $\hat{\boldsymbol{\mu}}_{\phi\,k}(\mathbf{x})$, $\hat{\boldsymbol{\sigma}}_{\phi\,k}(\mathbf{x}) \in \mathbb{R}^d$. Additionally, for a positive integer $n$, we denote the set $\{1, \ldots, n\}$ as $[n]$.

## 2 BACKGROUND AND PRELIMINARIES

We formalize our setting for time series representation learning, in which we have a set of high-dimensional time series observations $\mathbf{x}$ as $C$-variate time series observed at times $t = 1, \ldots, T$. We denote by $\mathbf{x} \in \mathbb{R}^{C \times T}$ resulting matrix with rows denoted by $\boldsymbol{x}_1, \ldots, \boldsymbol{x}_C$. Each row can be seen as a univariate time series in $\mathbb{R}^T$. Without loss of generality, we consider the case where $C = 1$. In the source separation problem, the observed signal $\mathbf{x} \in \mathcal{X}$ is assumed to be a mixture of $n$ sources, denoted as $\mathbf{y} := \{\boldsymbol{y}_1, \ldots, \boldsymbol{y}_n\} \in \mathcal{Y}$, where each $\boldsymbol{y}_k \in \mathbb{R}^T$, with additive independent noise $\xi \in \mathbb{R}^T$: $\mathbf{x} = \sum_{k=1}^n \boldsymbol{y}_k + \xi$. The space $\mathcal{Y}$ representing the individual source signals, satisfies $\mathcal{Y} \subseteq \mathcal{X}$ [2]. Given a data set of $N$ samples, denoted as $\{\mathbf{x}_i, \mathbf{y}_i\}_{i=1}^N$, the goal is to recover $\mathbf{y}$ from $\mathbf{x}$. Although the observed signal is a sum of sources, the mixing process is inherently *nonlinear* due to interactions from multi-state appliances, power distortions, and continuously fluctuating power in NILM (Yue et al., 2020), similar to harmonic distortions and reverberations in audio (Lu et al., 2021).

To formalize this idea, we consider a Euclidean observation space $\mathcal{X}$, and denote by $\mathcal{M}_+^1(\mathcal{X})$ the set of probability measures on $\mathcal{X}$. The standard framework for learning representations typically relies on VAEs (Kingma & Welling, 2014), which consist of two main components: i) the encoder network with parameters $\phi$, and ii) the decoder network with parameters $\theta$. The encoder parameterized a distribution $q_\phi(\mathbf{z}|\mathbf{x})$ over the latent space $\mathcal{Z} = \mathbb{R}^{d_{\mathcal{Z}}}$, with $d_{\mathcal{Z}} = d \times n$ representing the dimensionality, serves as a variational approximation of the Bayesian posterior $p_\theta(\mathbf{z}|\mathbf{x})$. The likelihood $p_\theta(\mathbf{x}|\mathbf{z})$ is parameterized by the decoder network. In standard setup, we assume a standard Gaussian prior $p(\mathbf{z}) = \mathcal{N}(\mathbf{0}, \mathbf{I})$ on $\mathcal{Z}$ and Gaussian distributions $q_\phi(\mathbf{z}|\mathbf{x})$. More precisely, for any $\mathbf{x} \in \mathcal{X}$, the distribution $q_\phi(\mathbf{z}|\mathbf{x})$ is a Gaussian distribution with a diagonal covariance matrix $\mathcal{N}(\boldsymbol{\mu}_\phi(\mathbf{x}), \operatorname{diag}(\boldsymbol{\sigma}_\phi^2(\mathbf{x})))$, where $\boldsymbol{\mu}_\phi \colon \mathcal{X} \to \mathcal{Z}$ and $\boldsymbol{\sigma}_\phi \colon \mathcal{X} \to \mathcal{Z}_{\geq 0}$. In order to simplify some of the expressions below, it may be useful to express the encoder network as a function $\boldsymbol{f}_\phi \colon \mathcal{X} \to \mathbb{R}^{2d_{\mathcal{Z}}}$, where $\boldsymbol{f}_\phi(\mathbf{x}) = [\boldsymbol{\mu}_\phi(\mathbf{x}), \boldsymbol{\sigma}_\phi(\mathbf{x})]^\top$ and the decoder is a compositional function $\boldsymbol{g}_\theta \colon \mathbb{R}^{d_{\mathcal{Z}}} \to \mathbb{R}^{T \times n}$, defined as $\boldsymbol{g}_\theta(\mathbf{z}) = \sum_{k=1}^n \boldsymbol{g}_{\theta\,k}(\mathbf{z})$, where each $\boldsymbol{g}_{\theta\,k} \colon \mathbb{R}^{d_{\mathcal{Z}}} \to \mathbb{R}^{T \times 1}$, mainly, $\boldsymbol{y}_k = \boldsymbol{g}_{\theta\,k}(\mathbf{z})$. The encoder and decoder networks are jointly trained on data set of $N$ samples by minimizing the following objective:

$$\mathcal{L}_{\text{VAE}}(\phi, \theta) = \frac{1}{N} \sum_{i=1}^N \left[ \mathbb{E}_{\mathbf{z} \sim q_\phi(\mathbf{z}|\mathbf{x}_i)} [\log p_\theta(\mathbf{x}_i|\mathbf{z})] - \beta \mathrm{KL}(q_\phi(\mathbf{z}|\mathbf{x}_i) \,\|\, p(\mathbf{z})) \right], \tag{2.1}$$

where the first part of Eq. (2.1) is the reconstruction loss and the second part is the KL-divergence between the latent distributions (associated to the training samples) and the prior over the latent space, weighted by a hyper-parameter $\beta > 0$ (Higgins et al., 2016). The reconstruction loss measures the similarity between the true source measurements $\mathbf{y} = \{\boldsymbol{y}_1, \ldots, \boldsymbol{y}_n\}$ and its reconstruction given by a multi-output decoder $\boldsymbol{g}_\theta(\mathbf{z}) =: \{\boldsymbol{g}_{\theta\,1}(\mathbf{z}), \ldots, \boldsymbol{g}_{\theta\,n}(\mathbf{z})\}$, and can be defined in many ways. With a Gaussian likelihood, the reconstruction loss is the squared $L_2$ norm: $\|\sum_{k=1}^n (\boldsymbol{y}_k - \boldsymbol{g}_{\theta\,k}(\mathbf{z}))\|^2$, or in an unsupervised fashion, *i.e.,* when the label source $\mathbf{y}$ is absent, the reconstruction loss becomes $\|\mathbf{x} - \boldsymbol{g}_\theta(\mathbf{z})\|^2$. After training, the VAE defines a generative model using the prior $p(\mathbf{z})$ and the decoder $\boldsymbol{g}_\theta$. The VAE's generated distribution denote by $\boldsymbol{g}_\theta \sharp p(\mathbf{z}) \in \mathcal{M}_+^1(\mathcal{X})$ allows one to generate new samples by first sampling a latent vector from the prior, then passing it through the decoder. We further assume the following:

---

[1]Pretrained models and usage guidelines: https://anonymous.4open.science/r/TimeCSL-4320

[2]When $\mathbf{x}$ is sparse, it may equal a single source $\boldsymbol{y}_1$, so $\mathcal{Y} \subseteq \mathcal{X}$.

**Assumption 2.1.** The decoder $g_\theta$ is a piecewise affine function, such as a multilayer perceptron with ReLU (or leaky ReLU) activations.

A special case of this model is well-studied in theory and applications and in deep generative models literature (Burgess & Kim, 2018; Ahuja et al., 2022). We consider the following generative process:

**Data-generating process.** We assume Asm 2.1, and we consider the following generative model for observations $\mathbf{x}$:

$$\mathbf{x} = \sum_{k=1}^{n} \boldsymbol{g}_{\theta\,k}(\mathbf{z}) + \xi, \quad \mathbf{z} = (\boldsymbol{z}_1, \ldots, \boldsymbol{z}_n) \in \mathbb{R}^{d \times n}, \text{vec}(\mathbf{z}) \sim \sum_{j=1}^{J} \omega_j \mathcal{N}(\text{vec}(\boldsymbol{\mu}_j), \boldsymbol{\Sigma}_j), \quad (2.2)$$

where $\xi \in \mathbb{R}^T$, denote independent random noise. Our results include the noiseless case $\xi = 0$ as a special case (*i.e.,* when all sources are well-known). The notation $\text{vec}(\mathbf{z}) \in \mathbb{R}^{d \cdot n}$ denotes the vectorization [3] of $\mathbf{z}$ that follows a multivariate Gaussian Mixture Model (GMM), and $\omega_j$ are the mixture weights (with $\sum_{j=1}^{J} \omega_j = 1$), with mean $\text{vec}(\boldsymbol{\mu}_j) \in \mathbb{R}^{d \cdot n}$ and $\boldsymbol{\Sigma}_j = \boldsymbol{\Sigma}_d \otimes \boldsymbol{\Sigma}_n$ with $\boldsymbol{\Sigma}_d$ being the $d \times d$ covariance and $\boldsymbol{\Sigma}_n$ the $n \times n$ covariance between sub-components *i.e.,* $\boldsymbol{z}_k$. Here, $\otimes$ denotes the Kronecker product. The GMM prior assumption can be generalized to exponential family mixtures (Kivva et al., 2022), provided the prior is analytic and affine-closed. Additionally, GMMs can approximate complex distributions (Nguyen & McLachlan, 2019). This maintains the flexibility and generalization of Eq. (2.2), and we impose no constraints on: 1) ReLU architectures, 2) independence of $\mathbf{z}$, or 3) the complexity of the mixture model or neural network.

**Objective.** Our goal is to identify the latent variables $\mathbf{z}$ from a set of observations $\mathbf{x}$ that lead to better reconstruction of true sources $\boldsymbol{y}_k = \boldsymbol{g}_{\theta\,k}(\mathbf{z})$, thus $\mathbf{y}$, which means, recovering $\mathbf{x}$ up to an additive error $\xi$. Thus, as far as disentanglement is considered to mean finding the original components $\mathbf{z}$ in a nonlinear mixing such Eq. (2.2), the very problem seems to be ill-defined. This is a fundamental problem which is receiving increasing attention in the deep learning community, and forms the basic motivation for nonlinear ICA theory (Hyvärinen & Pajunen, 1999). Unlike (Hyvärinen et al., 2023), our setting via Eq. (2.2) does not require $\boldsymbol{z}_k$ to be independent, recognizing the interdependencies in real-world data, and instead imposes structure on the nonlinear mixing Asm 2.1. Identifiability here ensures a linear mapping between ground truth and learned variables but does not guarantee disentanglement. Following (Lachapelle et al., 2022; Locatello et al., 2020), we extend this to define slot identifiability up to element-wise linear transformations below:

**Definition 2.2** (**Slot Identifiability and Disentangled Representation**). An autoencoder $\hat{\boldsymbol{g}}_\theta, \hat{\boldsymbol{f}}_\phi$ *slot-identifies* $\mathbf{z}$ on $\mathcal{Z}$ w.r.t. the true decoder $\boldsymbol{g}_\theta$ if $\hat{\mathbf{z}} = \hat{\boldsymbol{f}}_\phi(\boldsymbol{g}_\theta(\mathbf{z}))$ minimizes the reconstruction loss in Eq. (2.1) (first term), and there exists an invertible transformations $\mathbf{h} := \{\boldsymbol{h}_1, \boldsymbol{h}_2 \ldots, \boldsymbol{h}_n\}$, with $\boldsymbol{h}_k \in \mathbb{R}^d$, such that $\hat{\boldsymbol{z}}_k = \boldsymbol{h}_k(\boldsymbol{z}_k) \forall k \in [n]$, ensuring a one-to-one mapping. The learned representation $\hat{\mathbf{z}}$ identified *up to permutation, scaling, and element wise linear transformation* $\mathbf{z}$, if there exist a permutation matrix $\boldsymbol{\Pi}$ of $[n]$, an invertible diagonal matrix $\boldsymbol{\Lambda}$ constructed from the scaling factors of $\mathbf{h}$, and an offset $\mathbf{b}$, such that $\hat{\mathbf{z}} = \boldsymbol{\Lambda}\boldsymbol{\Pi}\mathbf{z} + \mathbf{b}$.

## 3 RELATED WORK

**On the Nonlinear ICA for Time Series Representation Learning.** Recent advances in nonlinear ICA has increasingly focused on utilizing temporal structures and nonstationarities for identifiability. (Hyvärinen & Morioka, 2016) introduced Time-Contrastive Learning (TCL), which assumes independent sources and leverages variance differences across data segments. Similarly, Permutation-based Contrastive Learning (PCL) identifies independent sources under the assumption of uniform dependency. i-VAE (Khemakhem et al., 2020a) extended this by using VAEs to approximate joint distributions in nonstationary regimes, relaxing the independence assumption with promising results. Further, (Roth et al., 2023) and (Oublal et al., 2024) explored using contrastive learning for latent space recovery without assuming source independence. Latent tEmporally cAusal Processes estimation (LEAPS) (Yao et al., 2021) introduces a nonparametric approach to causal discovery, but is limited by assumptions of no instantaneous causal influence and causal constancy. Work by (Lachapelle et al., 2022), and (Klindt et al., 2021) also requires source independence or some

---

[3] The vectorization of $\mathbf{z}$ (*i.e.,* stacks the columns of $\mathbf{z}$ in a single column vector), following a multivariate Gaussian mixture model, is equivalent to $\mathbf{z}$ following a Matrix Gaussian mixture, as shown in App. A.4.2.

intervention (Ahuja et al., 2023) to achieve identifiability. In contrast, our work extends identifiability theory by relaxing the independence assumption. We impose no constraints on $p(\mathbf{z})$ beyond its definition in Eq. (2.2), offering a more flexible framework. Recent studies have explored structural assumptions like orthogonality (Gresele et al., 2021; Zheng et al., 2022) or fixed sparsity (Moran et al., 2022), but our approach generalizes these further. Our intuitive argument is that sparsity and contrastive learning complement each other, potentially improving disentanglement.

**Time Series Representation with Out-Of-Distribution.** Handling out-of-distribution (OOD) data in time series representation has led to methods like RNNVAE (Chung et al., 2015), Slow-VAE (Klindt et al., 2021), and D3VAE (Li et al., 2023). Other approaches, such as CoTS (Woo et al., 2022), and CDSVAE (Bai et al., 2021b) focus on sequential data with contrastive disentanglement. Transformer-based models, such as Transformer (Zerveas et al., 2021), TimesNet (Wu et al., 2022), Autoformer (Wu et al., 2021), and Informer (Zhou et al., 2021), are designed to capture long-term dependencies but do not focus on identifiability or disentanglement. Understanding whether they preserve disentanglement representation across runs is crucial for robust representation learning. Inspired by OOD generalization frameworks in object-centric models (Zhao et al., 2022; Netanyahu et al., 2023), this ideas can be extend to time series. OOD generalization has been demonstrated in additive models (Dong & Ma, 2022) and slot-wise functions with nonlinearity (Wiedemer et al., 2023b), assuming identifiability for images. Work by (Lachapelle et al., 2023b) and (Wiedemer et al., 2023a) shows that additivity of the decoder (see § 2) ensures identifiability and decoder generalization under certain assumptions, which we apply to time series for an enhanced generalization.

## 4 IDENTIFIABILITY GUARANTEES VIA CONTRASTIVE SPARSITY-INDUCING

In this section, we begin with the intuition behind the proposed approach, which leverages sparsity in the mixing process to achieve identifiability. Previous methods relying on independence or non-Gaussian priors for identifiability often fail in nonlinear cases, as marginal transformations can preserve independence without revealing true structure (Hyvärinen & Pajunen, 1999; Hyvärinen et al., 2019). We build on the insight that any alternative solution introducing indeterminacy, beyond permutations or component-wise transformations, would result in a denser structure. Rather than constraining functional forms (Taleb & Jutten, 1999; Ahuja et al., 2023) or relying on auxiliary variables (Khemakhem et al., 2020a), we assume Partial Contrastive Sparsity for time series. This enables learning identifiable and disentangled representations without requiring independence or parametric assumptions on $p(\mathbf{z})$. In the following subsection, we present Partial contrastive Pairing.

① **Partial Contrastive Pairing for Time Series** For instance, in multiview object-centric settings (Bengio et al., 2020) or time series (see Fig. 2), a view $\mathbf{x}$ and its augmentation $\mathbf{x}'$ typically share limited information rather than complete overlap. To address this, we propose a more general case, *Partial Selective Pairing*, which allows pairs to share only a subset of relevant factors, serving as a relaxation of *Selective Pairing* in SSL. Assuming the data process generating Eq. (2.2), we define the shared support indices $\mathcal{S}$ of all sources that actively contribute to $\mathbf{x}$ as $\mathcal{S}(\mathbf{x}) := \{k \mid \boldsymbol{y}_k \neq 0, \ k = 1, 2, \dots, n\}$. The *Partial Selective Pairing* between observations $\mathbf{x}$ and $\mathbf{x}'$ is based on *shared support* $\mathbf{I}(\mathbf{x}, \mathbf{x}') := \mathcal{S}(\mathbf{x}) \cap \mathcal{S}(\mathbf{x}')$.

**Assumption 4.1** (**Sufficient Partial Selective Pairing**). For each factor $k \in [n]$, there exist observations $(\mathbf{x}, \mathbf{x}') \in \mathcal{X}$ such that the union of the shared support indices $\mathbf{i} = \mathbf{I}(\mathbf{x}, \mathbf{x}')$ that do not include $k$ must cover all other factors. Formally:

$$\bigcup_{\mathbf{i} \in \mathcal{I} \mid k \notin \mathbf{i}} \mathbf{i} = [n] \setminus \{k\} \quad , \quad \mathcal{I} := \{\mathbf{i} \subseteq [n] \mid p(\mathbf{i}) > 0\} \tag{4.1}$$

where $\mathcal{I}$ is the set of shared support indices and $p(\mathbf{i}) := \frac{1}{\#\mathcal{X}} \cdot \#\{\mathcal{S}(\mathbf{x}) = \mathbf{i}, \ \mathbf{x} \in \mathcal{X}\}$ gives the probability that the factors indexed by $\mathbf{i}$ are active, with $k \notin \mathbf{i}$ inactive.

In nonlinear ICA, sufficient variability assumes the auxiliary variable diversely affects source distributions (Hyvärinen & Morioka, 2016; Hyvarinen et al., 2019), while (Lachapelle et al., 2023a) adapted this concept for task supports. Similarly, Structural Variability (Ng et al., 2023) ensures each pair of sources influences distinct observed variables. However, overlapping influences often occur in real-world time series, posing practical challenges (see App. A.5). Instead, our Partial Selective Pairing assumption Eq. (4.1) allows some overlap, provided the union of shared support indices (excluding the specific source) spans all sources, enabling flexible modeling of source dependencies.

② **Identifiability via Contrastive Sparsity-inducing.** According to Asm 4.1, the sparsity-inducing nature arises from the existence of a source $k \notin \mathbf{i}$. However, this source is still well-defined within the support indicating that existing source $k$ remains inactive in either $\mathbf{x}$ or $\mathbf{x}'$. The use of a sparsity constraint or regularization is inspired by prior work (Ahuja et al., 2023; Lachapelle et al., 2023a) in the context of sparse multitask learning. The loss of zero reconstruction ensures that the encoding $\boldsymbol{f}_\phi(\mathbf{x})$ retains all information, implying that $(\hat{\mathbf{z}}, \hat{\mathbf{z}}')$ achieves sparsity comparable to the ground truth $(\mathbf{z}, \mathbf{z}')$. This sparsity in a latent representation $\hat{\mathbf{z}}$, means only a subset of latent variables are active for a given input $\mathbf{x}$. If $\frac{|\hat{\boldsymbol{\mu}}_{k,\phi}(\mathbf{x})|}{\hat{\boldsymbol{\sigma}}_{k,\phi}(\mathbf{x})}$ is small (e.g., close to zero), it suggests the $k$-th latent variable is not contributing, thus making it inactive $\boldsymbol{y}_k = 0$. However, when $\frac{|\hat{\boldsymbol{\mu}}_{k,\phi}(\mathbf{x})|}{\hat{\boldsymbol{\sigma}}_{k,\phi}(\mathbf{x})}$ is large (e.g., $\geq 1$), it implies the source $k$ contribute to $\mathbf{x}$. Bounding this ratio ensures that only the most relevant latent variables remain active, indirectly enforcing sparsity by limiting the number of significant variables. This raises the question of whether minimizing the $l_0$-norm of the learned latents variables, with sufficient partial pairing, can identify $\mathbf{z}$ through $\hat{\boldsymbol{g}}_\theta^{-1}(\mathbf{x})$ up to permutation and element-wise linear transformations. While $\boldsymbol{g}_\theta$ is nonlinear, sparsity alone is only valid for the linear case (Lachapelle et al., 2022) which is a strong assumption and may not be sufficient to resolve the ambiguities introduced by nonlinearities in many real-world cases. Sparsity without additional constraints, does not guarantee identifiability in practice, as $\hat{\boldsymbol{y}}_k = \hat{\boldsymbol{g}}_{\boldsymbol{\theta}\,k}^{-1} \circ \boldsymbol{g}_{\theta\,k}(\hat{\mathbf{z}})$ can depends on multiple components of $\mathbf{z}$. According to Darmois' theory (Darmois, 1953), this issue persists even when $\hat{\mathbf{z}}$ is sparse, further exacerbating unidentifiability. Building on this insight, we extend the concept of sparsity to contrastive sparsity by assuming Asm 2.1, without requiring bijectivity, and provide conditions under which $\mathbf{z}$ can be identified up to permutation and element-wise transformations.

**Theorem 4.2** (Element-wise Identifiability given index support $\mathbf{i}$ for Piecewise Linear $\boldsymbol{g}_\theta$). *Let $\boldsymbol{f}_\phi : \mathbb{R}^{d \times n} \to \mathbb{R}^{T \times n}$ be a continuous invertible piecewise linear function and $\hat{\boldsymbol{g}}_\theta : \mathbb{R}^{d \times n} \to \mathbb{R}^{T \times n}$ be a continuous invertible piecewise linear function onto its image. Assume that Asm 4.1, Asm 2.1 holds, and the mixed observations $(\mathbf{x}, \mathbf{x}') \overset{i.i.d.}{\sim} \mathcal{X}$, follows the data-generating process Eq. (2.2). The learnable latent $\hat{\mathbf{z}}$ (resp. $\hat{\mathbf{z}}'$) of $\mathbf{z}$ (resp. $\mathbf{z}'$). If all following conditions hold:*

$$\mathbb{E}\|\hat{\mathbf{z}}\|_0 \leq \mathbb{E}\|\mathbf{z}\|_0 \quad and \quad \mathbb{E}\|\hat{\mathbf{z}}'\|_0 \leq \mathbb{E}\|\mathbf{z}'\|_0, and, \tag{4.2}$$

$$\mathcal{R}_{alig}(\hat{\mathbf{z}}, \hat{\mathbf{z}}', \mathbf{i}) := \sum_{i \in \mathbf{i}} \left| \frac{\hat{\boldsymbol{z}}_i'^\top \hat{\boldsymbol{z}}_i}{\|\hat{\boldsymbol{z}}_i'\|_2 \|\hat{\boldsymbol{z}}_i\|_2} - 1 \right| = 0. \tag{4.3}$$

*then $\mathbf{z}$ is identified by $\mathbf{h} := \hat{\boldsymbol{g}}_\theta^{-1}(\mathbf{x})$, i.e., $\hat{\boldsymbol{g}}_\theta^{-1} \circ \boldsymbol{g}_\theta$ is a permutation composed with element-wise invertible linear transformations (Def. 2.2).*

*Proof Sketch.* Intuitively, based result (Kivva et al., 2022) combined with contrastivity between two latent based on their shared support indices $\mathbf{i}$. This means that for the data that satisfy Asm 4.1, $\boldsymbol{g}_\theta(\mathbf{z})$ and $\hat{\boldsymbol{g}}_\theta(\hat{\mathbf{z}})$ are equally distributed, then there exists an invertible affine transformation such that $\mathbf{h}(\mathbf{z}) = \mathbf{z}'$. Second, we use the strategy of linear identifiability (Lachapelle & Lacoste-Julien, 2022) to obtain element wise identifiability. The complete proof are given in App. A.3. This approach is similar to SparseVAE (Moran et al., 2022), which enforces constraints using Spike-and-Slab Lasso. However, our method ensures slot identifiability through Partial Selective Pairing, without requiring strong assumptions or extra constraints on $\mathcal{Z}$. In contrast, SparseVAE uses separate decoders for each feature. Another line of work can dive to constrains the generator $\boldsymbol{g}_\theta$ via its Jacobian $\mathbf{J}\boldsymbol{g}_\theta(\mathbf{z})$, known as compositionality and irreducibility (Von Kügelgen et al., 2021; Brady et al., 2023). Definitions are provided in App. A.2. Within our framework, compositionality means that each high-dimensional source is controlled by only one latent slot $\boldsymbol{z}_k$, enforcing local sparsity. However, minimizing compositionality in $\hat{\boldsymbol{g}}_\theta$ on $\mathcal{Z}$ is computationally infeasible [4].

③ **Invariance for Compositional Generalization Representation** From Thm. 4.2, it follows that $\hat{\boldsymbol{g}}_\theta$ faithfully maps each inferred slot $\boldsymbol{h}_k(\boldsymbol{z}_{\pi(k)})$ to its corresponding source in $\mathbf{x}$ for all possible values of $\boldsymbol{z}_{\pi(k)}$, ensuring identifiability (ID). We extend this to ensuring OOD scenarios by simply composing the latents from the training set and applying a stop gradient to prevent the gradients from flowing back into the recomposed latent during training (see Fig. 2). During training, simultaneously, we perform ID and OOD, ensuring that the combined latent remains consistent *i.e.,* compositional with the original latent, allowing the model to generalize OOD samples while retaining the ID. Assuming the conditions stated in Thm. 4.2 are satisfied, this implies the existence of transformations

---

[4]For a CNN with 1 million parameters and a batch size of 32, at least 250GB of GPU memory is required.

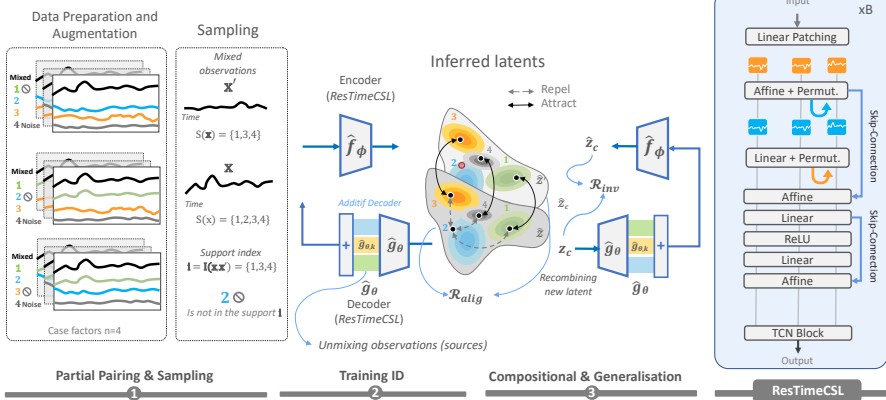

Figure 3: **Overview of `TimeCSL` framework using ResTimeCSL Architecture.** After linearly projecting the time series patches into high dimensional embeddings the ResTimeCSL is affine.

$\mathbf{h}$, along with a permutation $\pi$, that enable the slot-identification $\mathbf{z}$ for any composition of slots, whether ID or OOD, over $\mathcal{Z}$, as given by

$$\mathbf{z}_c = \boldsymbol{f}_\phi(\boldsymbol{h}_1(\boldsymbol{z}_{\pi(1)}), \dots, \boldsymbol{h}_n(\boldsymbol{z}_{\pi(n)})), \text{ and } \mathcal{Z}_c = \boldsymbol{f}_\phi(\boldsymbol{h}_1(\mathcal{Z}_{\pi(1)}) \times \dots \times \boldsymbol{h}_n(\mathcal{Z}_{\pi(n)})). \quad (4.4)$$

The compositional generalization consistency on $\mathcal{Z}_c$, holds, *i.e.*, $\hat{\boldsymbol{g}}_\theta^{-1}(\boldsymbol{g}_\theta(\mathbf{z})) = \mathbf{z}_c$ and $\hat{\boldsymbol{g}}_\theta(\mathbf{z}_c) = \boldsymbol{g}_\theta(\mathbf{z})$, if and only if $\mathbf{z}_c$ minimizes the invariance such that,

$$\mathcal{R}_{inv}(\mathbf{z}_c) := \sum_{i \neq k} \left( \frac{\boldsymbol{z}_{ci}^\top \boldsymbol{z}_{ck}}{\|\boldsymbol{z}_{ci}\|_2 \|\boldsymbol{z}_{ck}\|_2} \right)^2, \text{ for some } \gamma_{inv} > 0, \gamma_{inv} \mathcal{R}_{inv}(\mathbf{z}_c) = 0. \quad (4.5)$$

The condition in Eq. (4.5) ensure that $\hat{\boldsymbol{f}}_\phi$ inverts $\hat{\boldsymbol{g}}_\theta$ on ID and OOD by re-encoding the latent from inferred ones (see Fig. 3). Implementation details and sampling process of $\mathbf{z}_c$ for this regularization is discussed in § 4.1. To validate Eq. (4.5), we have just to verify the compositional consistency error *i.e.*, $\hat{\boldsymbol{g}}_\theta^{-1}(\hat{\boldsymbol{g}}_\theta(\mathbf{z}_c) = \mathbf{z}_c$ over $\forall \mathbf{z}_c \in \mathcal{Z}_c$. Formally:

$$\mathcal{L}_{cons} := \mathbb{E}_{\mathbf{z}_c \sim q_\phi(\mathbf{z}_c)}[||\hat{\boldsymbol{f}}_\phi(\hat{\boldsymbol{g}}_\theta(\mathbf{z}_c) - \mathbf{z}_c||] = 0, \text{ where, } supp(q_\phi(\mathbf{z}_c)) = \mathcal{Z}' \text{ Eq. (4.4)}. \quad (4.6)$$

## 4.1 PUTTING IT ALL TOGETHER IN PRACTICE

**On the Possibility of Sufficient Partial Pairing** In Thm. 4.2, we demonstrated how slot identifiability can be achieved on $\mathcal{Z}$ and OOD $\mathcal{Z}_c$ under the compositionality condition in Eq. (4.6). A key insight is the sufficient partial pairing for contrastive learning (Asm 4.1). This assumption can be relaxed to factor groups when the dataset is complex enough to discern varying features (e.g., in weather time series). For such cases, grouping factors avoids assumption violations. We validated our results on synthetic time series data (assumptions fully satisfied) and energy separation tasks, were used to relax assumptions via grouping factors. Data was prepared in pairs $(\mathbf{x}, \mathbf{x}')$, with additional samples generated as needed to cover all factors.

**Conditions on the Network.** We proposed ResTimeCSL (see Fig. 3), an efficient architecture for time series modeling that doesn't violate Asm 2.1. It projects time series patches into high-dimensional embeddings and processes them sequentially using a cross-patch linear sublayer and a cross-channel two-layer MLP, similar to the Transformer's FCN sublayer. Each sublayer includes residual connections, two affine element-wise transformations, and uses ReLU or LeakyReLU activations. For training, we leverage a VAE model with a mixture of Gaussians (Jiang et al., 2016) for a fixed latent dimension by $n$ and $d$, optimizing the objective $\mathcal{L}_{\text{VAE}}$. We sample i.i.d. pairs $(\mathbf{x}, \mathbf{x}') \in \mathcal{X}$. Using a learnable encoder $\hat{\boldsymbol{f}}_\phi$, $\mathbf{x}$ (resp. $\mathbf{x}'$) is encoded into $[\hat{\boldsymbol{\mu}}_{\phi k}(\mathbf{x}), \hat{\boldsymbol{\sigma}}_{\phi k}(\mathbf{x})]^\top$ (resp. $[\hat{\boldsymbol{\mu}}_{\phi k}(\mathbf{x}'), \hat{\boldsymbol{\sigma}}_{\phi k}(\mathbf{x}')]^\top$) with reparameterization noise terms (Kingma & Welling, 2022). The inferred latents are $(\hat{\mathbf{z}}, \hat{\mathbf{z}}')$. A learnable decoder $\hat{\boldsymbol{g}}_\theta$ maps $\hat{\mathbf{z}}$ (resp. $\hat{\mathbf{z}}'$) to single-source outputs $\hat{\boldsymbol{y}}_k = \boldsymbol{g}_{\theta k}(\hat{\mathbf{z}})$ (resp. $\hat{\boldsymbol{y}}'_k = \hat{\boldsymbol{g}}_{\theta k}(\hat{\mathbf{z}}')$) for $k = 1, \dots, n$. Summing over these outputs reconstructs the mixed signals $\hat{\mathbf{x}}$ (resp. $\hat{\mathbf{x}}'$). In practice, the sparsity of the ground truth variables $\mathbf{z}$ is unknown, so we instead set a hyperparameters $\eta$ for the sparsity constraint. Furthermore, for more stability, instead of $\mathbb{E} \|\mathbf{z}\|_0 \leq \eta$ we consider $\|\mathbf{v}\|_{s,\text{norm}} = \frac{1}{d_z} \sum_{i=1}^{d_z} \sum_{j=1}^{n_a+1} |v_{ij}|$. The `TimeCSL` objective serves then as

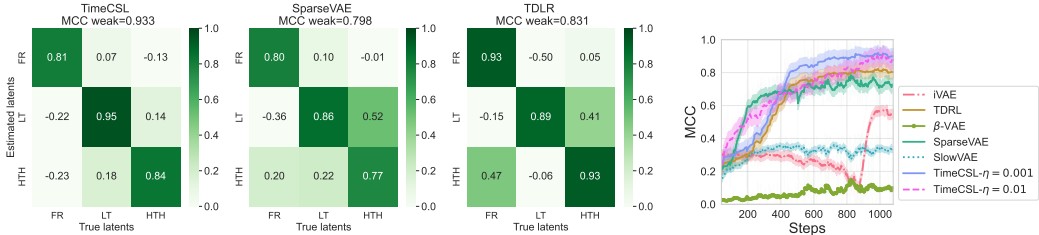

Figure 4: **Identifiability Validation.** MCC for factors $\{$FR, LT, HTR$\}$ on synthetic data; **Left:** Weak MCC for TimeCSL, SparseVAE, and TDRL. **Right:** Baseline comparisons over training steps.

a regularization term for the loss $\mathcal{L}^*_{\mathrm{VAE}}$, that denote the sum of $\mathcal{L}_{\mathrm{VAE}}$ computed for time series $\mathbf{x}$ and $\mathbf{x}'$. Thus, the final objective can be expressed as follows:

$$\mathcal{L}_{\texttt{TimeCSL}}(\phi, \theta; \mathcal{B}) = \mathcal{L}^*_{\mathrm{VAE}}(\phi, \theta; \mathcal{B}) + \mathbb{E}_{\mathcal{B}}[\gamma_{alig}\mathcal{R}_{alig}(\mathbf{z}, \mathbf{z}', \mathbf{i})] + \mathbb{E}_{\mathcal{B}}[\gamma_{inv}\mathcal{R}_{inv}(\mathbf{z}_c, \mathbf{i})] \quad (4.7)$$
$$+ \mathbb{E}_{\mathcal{B}}\| \max(0, \|\hat{\mathbf{z}}\|_s - \eta) + \max(0, \|\hat{\mathbf{z}}'\|_s - \eta)\|,$$

where $\mathcal{B}$ is a batch of data. The alignment term $\mathcal{R}_{alig}$ penalizes deviations from cosine similarity between corresponding latents, scaled by $\gamma_{alig}$. The invariance term $\mathcal{R}_{inv}$, scaled by $\gamma_{inv}$, reduce invariance of the latent composed $\mathbf{z}_c$ from $\hat{\mathbf{z}}$ and $\hat{\mathbf{z}}'$. In our experiments, we use $\eta = 0.01$ or $0.001$.

# 5 EXPERIMENTS

## 5.1 VALIDATION OF THE THEORY

**Datasets and Evaluation Setup.** We conducted experiments for time series representation with separation task on three public **real datasets:** UK-DALE (Kelly & Knottenbelt, 2015), REDD (Kolter & Johnson, 2011), and REFIT (Murray et al., 2017) providing power measurements from multiple homes. $60\%$ of the data is used for training with additional $10\%$ of data augmentation, while the remaining $40\%$ of real data is evenly divided between validation and testing. Inputs are zero-mean normalized, we consider $T = 256$, $C = 1$ and number factors/sources $n = 5$: Fridge (FR), Dishwasher (DW), Washing Machine (WM), Heater (HTR), and Lighting (LT). The mixed observation may include unlabeled noise factors. **Synthetic Dataset:** we generate a nonlinear mixing observations with $n = 3$, from ground truth available signals of $\{$FR, LT, HTR$\}$ from UK-DALE, REDD, and REFIT with adding some Gaussian noise. To generate OOD scenarios Tab. 2 *i.e.,* strong correlation between factors, we adopt the methodology outlined in (Träuble et al., 2021) where $p(y_1, y_2) \propto \exp\left(-||y_1 - \alpha y_2||^2 / 2\sigma^2\right)$ and adjusting the parameter $\sigma$ to control the correlation.

**Metrics.** To assess slot identifiability, we follow (Locatello et al., 2020) by fitting nonlinear regressors to predict each ground-truth slot $\mathbf{z}_k$ from inferred slots $\hat{\mathbf{z}}_j$, evaluating the fit with the $R^2$ score. Slot assignments are optimized via the Hungarian algorithm (Kuhn, 1955), and we report the average $R^2$ over matched slots. Additionally, we use the Mean Correlation Coefficient (MCC) metric (Khemakhem et al., 2020a), reporting both *strong* MCC (before affine alignment) and *weak* MCC (after alignment). All MCCs are computed out-of-sample: the affine map $\Gamma$ is fitted on one half of the data and evaluated on the other. RMIG (Robust Mutual Information GAP) (Do & Tran, 2019), and DCI (Disentanglement, Completeness and Informativeness) (Eastwood & Williams, 2018) adapted for time series are used to evaluate the disentanglement of factors *i.e.,* sources. We provide in-depth details of metrics and their implementation in App. B.4.

**Contrastive Partial Selective Pairing Pipeline.** Four augmentations were sequentially applied to all contrastive methods' pipeline branches. The parameters from the random search are: 1) **Crop and delay:** applied with a $0.5$ probability and a minimum size of $50\%$ of the initial sequence. 2) **Cutout or Masking:** time cutout of $5$ steps with a $0.8$ probability. 3) **Channel Masks powers:** each time series is randomly masked out with a $0.4$ probability. 4) **Gaussian noise:** random Gaussian noise is added to window input $\mathbf{x}$ with a standard deviation form $0.1$ to $0.3$. Further details in App. B.3.

**Baselines & Implementations.** Nonlinear ICA methods are used; $\beta$-VAE, iVAE and TCL which leverage nonstationarity establish identifiability but assumes independent factors, and SlowVAE/SlowVAE which exploit temporal constraints but assume independent sources. We provide also variant $\beta$-TC/Factor/-VAE such as D3VAE and CDSVAE implemented for time series sequence modeling. We

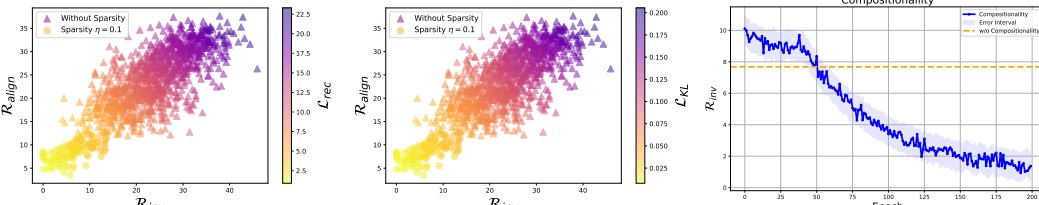

Figure 5: **Experimental validation. Left**: As predicted by Eq. (4.2), inducing sparsity in models that minimize $\mathcal{R}_{alig}$ and $\mathcal{R}_{inv}$ results in representations that are slot-identifiable both in ID and OOD, provided the reconstruction loss $\mathcal{L}_{rec}$ (as in Eq. (2.1)) is also minimized (see heat-map). A similar trend is observed for the $\mathcal{L}_{KL}$. **Right**: Compositional error Eq. (4.6) decreases throughout training, indicating that the decoder is implicitly optimized to be compositional, then validating Eq. (4.5).

Table 1: Average performance, considering factors {FR, DW, WM, HTR, LT} with 5 seed on real datasets REFIT and REDD. Metrics reported are: DCI, RMIG and RMSE. Lower values are better for all metrics. (↓ lower is better, ↑ higher is worse  Top-1 ,  Top-2 ).

| Sc. | Methods | $\sigma = \infty$ | | | $\sigma = 0.3$ | | | $\sigma = 0.8$ | | |
|---|---|---|---|---|---|---|---|---|---|---|
| | Metrics ⇒ | DCI ↓ | RMIG ↓ | RMSE ↓ | DCI ↓ | RMIG ↓ | RMSE ↓ | DCI ↓ | RMIG ↓ | RMSE ↓ |
| REFIT | ○ BertNILM | - | - | $56.4 \pm 2.58$ | - | - | $70.2 \pm 1.45$ | - | - | $70.92 \pm 1.15$ |
| | ○ S2S | - | - | $54.3 \pm 3.12$ | - | - | $69.95 \pm 3.56$ | - | - | $69.95 \pm 3.26$ |
| | ○ Autoformer | - | - | $49.7 \pm 0.81$ | - | - | $50.5 \pm 2.15$ | - | - | $52.95 \pm 1.63$ |
| | ○ Informer | - | - | $50.3 \pm 2.41$ | - | - | $53.5 \pm 1.98$ | - | - | $58.95 \pm 1.89$ |
| | ● TimesNet | - | - | $49.24 \pm 2.87$ | - | - | $51.10 \pm 2.64$ | - | - | $54.91 \pm 2.31$ |
| | ● CoST | $68.4 \pm 2.41$ | $0.94 \pm 0.03$ | $47.7 \pm 1.35$ | $73.7 \pm 2.41$ | $0.98 \pm 0.27$ | $53.2 \pm 1.02$ | $71.95 \pm 1.63$ | $1.00 \pm 0.02$ | $58.45 \pm 0.82$ |
| | ○ SlowVAE | $78.0 \pm 1.09$ | $0.94 \pm 0.13$ | $43.2 \pm 2.23$ | $81.0 \pm 1.82$ | $0.94 \pm 0.13$ | $49.2 \pm 1.13$ | $79.74 \pm 0.84$ | $1.07 \pm 0.11$ | $54.65 \pm 1.43$ |
| | ● SlowVAE+HDF | $79.8 \pm 0.10$ | $0.64 \pm 0.05$ | $57.2 \pm 2.15$ | $81.1 \pm 0.34$ | $0.71 \pm 0.14$ | $59.3 \pm 1.82$ | $80.37 \pm 0.05$ | $0.72 \pm 0.03$ | $61.64 \pm 1.52$ |
| | ● TDRL | $64.85 \pm 1.48$ | $0.42 \pm 0.12$ | $28.56 \pm 2.15$ | $76.23 \pm 1.32$ | $0.48 \pm 0.02$ | $26.33 \pm 1.97$ | $77.13 \pm 1.00$ | $0.58 \pm 0.24$ | $31.99 \pm 1.64$ |
| | ○ D3VAE | $63.12 \pm 2.84$ | $0.40 \pm 0.14$ | $42.28 \pm 2.13$ | $63.66 \pm 1.31$ | $0.51 \pm 0.38$ | $46.11 \pm 1.58$ | $66.73 \pm 1.88$ | $0.67 \pm 0.08$ | $50.10 \pm 0.74$ |
| | ○ C-DSVAE | $72.42 \pm 3.10$ | $0.91 \pm 0.15$ | $48.6 \pm 2.32$ | $73.12 \pm 1.43$ | $0.95 \pm 0.41$ | $52.9 \pm 1.71$ | $76.29 \pm 2.04$ | $1.08 \pm 0.09$ | $57.45 \pm 0.81$ |
| | ● C-DSVAE + HDF | $67.80 \pm 2.91$ | $0.85 \pm 0.14$ | $45.45 \pm 2.18$ | $68.76 \pm 1.34$ | $0.90 \pm 0.39$ | $49.69 \pm 1.60$ | $71.50 \pm 1.92$ | $1.01 \pm 0.08$ | $53.85 \pm 0.76$ |
| | ● SparseVAE | $61.51 \pm 1.31$ | $0.39 \pm 0.13$ | $21.01 \pm 1.89$ | $67.29 \pm 1.17$ | $0.43 \pm 0.62$ | $22.71 \pm 1.73$ | $68.19 \pm 0.88$ | $0.51 \pm 0.21$ | $28.91 \pm 1.89$ |
| | ● TimeCSL | $\mathbf{59.71 \pm 1.27}$ | $\mathbf{0.36 \pm 0.11}$ | $\mathbf{18.44 \pm 1.84}$ | $\mathbf{65.22 \pm 1.13}$ | $\mathbf{0.41 \pm 0.23}$ | $\mathbf{19.11 \pm 1.69}$ | $\mathbf{66.01 \pm 0.86}$ | $\mathbf{0.48 \pm 0.08}$ | $\mathbf{22.21 \pm 1.41}$ |
| | Avg. | $69.74 \pm 1.95$ | $0.80 \pm 0.10$ | $47.3 \pm 1.92$ | $73.4 \pm 1.22$ | $0.90 \pm 0.17$ | $52.25 \pm 1.47$ | $74.98 \pm 1.38$ | $1.00 \pm 0.08$ | $54.9 \pm 1.25$ |
| REDD | ○ BertNILM | - | - | $61.42 \pm 3.47$ | - | - | $67.61 \pm 1.95$ | - | - | $69.06 \pm 1.43$ |
| | ○ S2S | - | - | $59.08 \pm 4.15$ | - | - | $68.60 \pm 3.91$ | - | - | $70.68 \pm 3.25$ |
| | ○ Autoformer | - | - | $49.87 \pm 0.92$ | - | - | $51.53 \pm 1.48$ | - | - | $51.88 \pm 1.34$ |
| | ○ Informer | - | - | $54.61 \pm 1.41$ | - | - | $58.13 \pm 0.67$ | - | - | $62.45 \pm 1.76$ |
| | ● TimesNet | - | - | $51.37 \pm 2.41$ | - | - | $55.35 \pm 2.23$ | - | - | $58.47 \pm 2.21$ |
| | ● CoST | $62.60 \pm 2.20$ | $0.86 \pm 0.03$ | $43.53 \pm 1.23$ | $67.51 \pm 2.11$ | $0.89 \pm 0.25$ | $48.71 \pm 0.94$ | $65.98 \pm 1.50$ | $0.92 \pm 0.02$ | $53.32 \pm 0.75$ |
| | ○ SlowVAE | $71.14 \pm 0.96$ | $0.86 \pm 0.12$ | $39.46 \pm 2.05$ | $74.34 \pm 1.60$ | $0.86 \pm 0.12$ | $45.02 \pm 1.04$ | $73.19 \pm 0.77$ | $0.98 \pm 0.10$ | $49.94 \pm 1.31$ |
| | ● SlowVAE+HDF | $73.12 \pm 0.09$ | $0.59 \pm 0.05$ | $52.34 \pm 1.97$ | $74.40 \pm 0.31$ | $0.65 \pm 0.13$ | $54.48 \pm 1.67$ | $73.75 \pm 0.05$ | $0.66 \pm 0.03$ | $56.28 \pm 1.40$ |
| | ● TDRL | $59.39 \pm 1.31$ | $0.38 \pm 0.11$ | $26.12 \pm 1.97$ | $69.82 \pm 1.19$ | $0.44 \pm 0.02$ | $24.10 \pm 1.78$ | $70.82 \pm 0.91$ | $0.53 \pm 0.22$ | $29.27 \pm 1.51$ |
| | ○ D3VAE | $59.39 \pm 2.56$ | $0.74 \pm 0.13$ | $39.56 \pm 1.92$ | $59.65 \pm 1.17$ | $0.78 \pm 0.34$ | $43.13 \pm 1.42$ | $62.62 \pm 1.69$ | $0.89 \pm 0.07$ | $47.07 \pm 0.66$ |
| | ○ C-DSVAE | $66.44 \pm 2.84$ | $0.83 \pm 0.14$ | $44.51 \pm 2.13$ | $67.06 \pm 1.31$ | $0.87 \pm 0.38$ | $48.48 \pm 1.58$ | $70.24 \pm 1.88$ | $0.99 \pm 0.08$ | $52.74 \pm 0.74$ |
| | ● C-DSVAE + HDF | $62.20 \pm 2.67$ | $0.78 \pm 0.13$ | $41.65 \pm 2.01$ | $63.23 \pm 1.24$ | $0.83 \pm 0.36$ | $45.71 \pm 1.48$ | $65.73 \pm 1.77$ | $0.93 \pm 0.07$ | $49.54 \pm 0.70$ |
| | ● SparseVAE | $56.39 \pm 1.21$ | $0.36 \pm 0.12$ | $19.21 \pm 1.74$ | $61.60 \pm 1.07$ | $0.45 \pm 0.57$ | $20.81 \pm 1.60$ | $62.65 \pm 0.81$ | $0.47 \pm 0.19$ | $26.42 \pm 1.74$ |
| | ● TimeCSL | $\mathbf{54.74 \pm 1.17}$ | $\mathbf{0.33 \pm 0.10}$ | $\mathbf{16.93 \pm 1.70}$ | $\mathbf{60.10 \pm 1.04}$ | $\mathbf{0.38 \pm 0.21}$ | $\mathbf{17.50 \pm 1.56}$ | $\mathbf{60.31 \pm 0.79}$ | $\mathbf{0.44 \pm 0.07}$ | $\mathbf{20.39 \pm 1.30}$ |
| | Avg. | $69.25 \pm 1.87$ | $0.67 \pm 0.09$ | $47.4 \pm 1.83$ | $74.2 \pm 1.36$ | $0.73 \pm 0.10$ | $53.16 \pm 1.55$ | $75.55 \pm 1.23$ | $0.80 \pm 0.08$ | $56.31 \pm 1.48$ |

compare TimeCSL with downstream task models in energy disaggregation, BertNILM (Yue et al., 2020) and S2S (Chen et al., 2018a) as a baseline, for those models, we keep the same configuration as the original implementation. We run experiments with 5 seeds, reporting average results and standard deviations, using 8 NVIDIA A100 GPUs. Hyperparameters and training details are in App. B.

**Results.** Fig. 4 shows that standard nonlinear ICA models like $\beta$-VAE/C-DSVAE, and SlowVAE struggle with identifiability, while SparseVAE and iVAE perform comparatively better on synthetic data. TimeCSL with strong sparsity ($\eta = 0.01$) achieves the best identifiability. Fig. 5 provides convincing probes of the compositional generalization consistency condition Eq. (4.5), where minimizing $\mathcal{R}_{alig}$ and $\mathcal{R}_{inv}$, both with and without sparsity, aligns with the predictions of Thm. 4.2. Slot identifiability improves as reconstruction error decreases, with similar trends observed for $\mathcal{L}_{KL}$. Additionally, Fig. 5 (Left) illustrates a reduction in compositional error as $\mathcal{R}_{inv}$ is minimized, confirming the compositional nature of the decoder as predicted by Eq. (4.5). Empirically, Tab. 1 summarizes the performance of different models as data complexity increases, controlled by correlation levels. The findings show that TimeCSL surpasses SparseVAE, demonstrating better disentanglement and reconstruction. However, at higher correlation levels, models without tailored designs for identifiability and disentanglement face challenges, underscoring potential limitations in real-world applications.

### 5.2 ABLATION STUDIES AND DISCUSSION

**When and how to perform disentanglement?** In Tab. 2, we use TimeCSL as regularizer, and we train models only on (REFIT+REDD), while testing them on possible OOD dataset *i.e.,* UKDALE.

Figure 6: Relative RMSE (%) improvement over baseline BertNILM Yue et al., 2020 for {FR, DW, WM, HTR, LT} devices, with the amount of labeled training data as a variable parameter.

We explore its application with alternative structures especially tailored for time series, focusing on the analysis of the impact of nonlinearity of the decoder induced by the activation function, (Asm 2.1 does not hold), especially those residual in Diffusion based VAE model (D3VAE). The model demonstrates improved generalization when `TimeCSL` is combined with another method, leading to slightly better results. Secondly, `TimeCSL` displays improved performance as sparsity increases, with $R^2$ positively correlating with performance. `RMIG` further indicates that integrating attention with `TimeCSL` yields well-disentangled representations. The attention mechanism, which intro-

Table 2: Average $R^2$, RMIG and weaker/strong MCC scores on UK-DALE dataset with factors {FR, DW, WM, HTR, LT}. (↓ lower is better, ↑ higher is better Top-1 , Top-2 ).† indicates implemented.

| Method | Activation | $R^2$ ↑ | RMIG ↓ | weak MCC ↑ | strong MCC ↑ |
|---|---|---|---|---|---|
| ○ CoST | ReLU | 0.165 | 0.405 | 0.395 | -0.010 |
| ○ RNN-VAE (baseline) | LeakyReLU | 0.065 | 0.660 | 0.340 | 0.080 |
| ○ RNN-VAE+TimeCSL | LeakyReLU | 0.169 | 0.562 | 0.400 | 0.038 |
| ○ C-DSVAE | ReLU | 0.127 | 0.415 | 0.685 | 0.070 |
| ○ C-DSVAE+TimeCSL | ReLU | 0.167 | 0.511 | 0.578 | 0.167 |
| ○ SlowVAE | LeakyReLU | 0.263 | 0.860 | 0.671 | 0.082 |
| ○ SlowVAE+TimeCSL | LeakyReLU | 0.272 | 0.560 | 0.387 | 0.074 |
| ○ DIOSC | Softmax | 0.280 | 0.368 | 0.562 | 0.194 |
| ○ D3VAE (Diffusion) | Softmax | 0.271 | 0.791 | 0.544 | 0.188 |
| ○ D3VAE+TimeCSL (Diffusion) | Softmax | 0.285 | 0.682 | 0.573 | 0.198 |
| ○ iVAE | LeakyReLU | 0.230 | 0.408 | 0.479 | 0.177 |
| ○ TDRL | LeakyReLU | 0.223 | 0.380 | 0.464 | 0.172 |
| ○ TCL | LeakyReLU | 0.115 | 0.748 | 0.448 | 0.165 |
| ○ LEAP | LeakyReLU | 0.138 | 0.340 | 0.538 | 0.198 |
| ● TimeCSL $\eta = 0.001$ | ReLU | 0.292 | 0.330 | 0.629 | 0.258 |
| ● TimeCSL†+self-attention | Softmax | 0.231 | 0.478 | 0.373 | 0.106 |
| ● TimeCSL † $\eta = 0.01$ | ReLU | **0.305** | **0.367** | **0.633** | **0.266** |

duces nonlinearities, still improves model performance, though less than `TimeCSL`, and with reduced identifiability, indicating possible empirical weak disentanglement, even when nonlinearity preexists.

**Is the sparsity enough to ensure robustness in downstream tasks?** We provide evidence that `TimeCSL` exhibits robustness across different correlation scenarios as illustrated in Fig. 6. In addition, we conduct experiments using different sate of the art architecture for time series representation. The results in Fig. 6 and Tab. 2 demonstrate that `TimeCSL` with sparsity $\eta = 0.1$ is more consistent than `TimeCSL` with lower sparsity *i.e.,* $\eta = 0.01$, outperforming the baseline across all three correlation settings ($\sigma = \{0.3, 0.5, 0.7\}$). This underscores its effectiveness and adaptability in scenarios with strongly correlated data. For more in-depth analysis, additional results are available in App. B.9.1.

## 6 CONCLUSION

In this work, we delved into the effectiveness of contrastive sparsity-inducing techniques in attaining both identifiability and generalization. We showcased that disentangled representations, complemented by sparse-inducing methods through contrastive learning, improve generalization, particularly when the downstream task can be tackled using only a portion of the underlying factors of variation. Looking ahead, future investigations could explore leveraging such meaningful representations for downstream tasks, as evidenced by our primary experiments demonstrating performance enhancement. Furthermore, we posit that such representations could prove efficient in scenarios characterized by limited labeled data for time series representation. We have demonstrated generalization through compositional representations. We built on the literature in generative models and nonlinear ICA (Kivva et al., 2022; Hyvarinen et al., 2019; Lachapelle et al., 2022) and made two key assumptions: i) partial sufficiency holds, which enables sparsity through contrastive learning, and ii) the decoder $g_\theta$ is injective. Our results are a step toward identifiability and disentanglement in time series models.

**Limitations & Future Work** We acknowledge that our assumptions on time series representation and source separation have room for extension. The piecewise injectivity assumption (Asm 2.1), though potentially violated in practice, could be revised to incorporate structures like attention mechanisms or instance normalization. The Sufficient Partial Pairing assumption (Asm 4.1) depends on having sufficient data, and as noted in § 4.1, it can also be relaxed to group factors. Looking ahead, these extensions offer exciting opportunities for further improving the model's robustness and flexibility.

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

# Supplementary Material:

To ensure a comprehensive understanding of our paper and to support reproducibility and reliability, we present additional results and provide complete proofs for the theorems articulated in the main paper. This supplementary material is meticulously organized as follows:

# Table of Contents

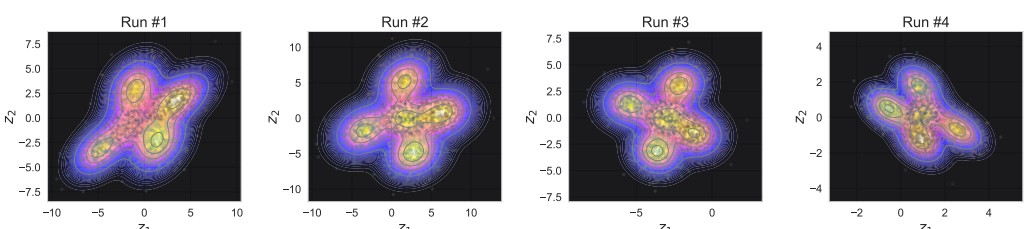

Figure 7: Recovered latent spaces for 4 runs of `TimeCSL` on REDD dataset with 5 latents ($n = 5, d = 16$) {FR, DW, WM, HTR, LT}.

## A  EXTENDED RELATED WORK AND PROOFS

In this section, we detail the contributions of the paper, including all the details. Although there is no change in their contents, the formulation of some definitions and theorems are slightly altered here to be more precise and cover edge cases omitted in the main text. Hence, the numbering of the restated elements is reminiscent of that used in the main text.

## A.1 EXTEND THE DISCUSSION ON RELATED WORK

Self-supervised learning (SSL) methods have moved away from using negative pairs, as in contrastive learning (CL), and instead focus on alignment with various forms of regularization to prevent collapsed representations. For example, BYOL (Grill et al., 2020) and SimSiam (Chen & He, 2021) use architectural regularization with moving-average updates for a separate *target* network (BYOL only) or a stop-gradient operation (for both). Meanwhile, BarlowTwins (Zbontar et al., 2021) promotes redundancy reduction and alignment by optimizing the cross-correlation between $\mathbf{z}$ and $\mathbf{z}'$ to match the identity matrix, ensuring zero off-diagonals and ones on the diagonal. We can interpret positive augmentation as a modified representation $\mathbf{z}'$ that is connected to the original $\mathbf{z}$ through a conditional distribution $p(\mathbf{z}' \mid \mathbf{z})$. This implies that the augmented observation $\mathbf{x}'$ shares similar information with the anchor observation $\mathbf{x}$, and is generated by applying the same mixing function $g_\theta$ as defined in data-generating process Eq. (2.2).

Table 3: Related work in nonlinear ICA for time series. A blue check denotes that a method has an attribute, whereas a red cross denotes the opposite. † indicates an approach we implemented.

| Approach | Temporal Data | Dependent Factors | Nonparametric Expression | Stationary Process |
|---|---|---|---|---|
| TCL (Hyvarinen & Morioka, 2016) | ✓ | ✗ | ✗ | ✗ |
| PCL (Hyvarinen & Morioka, 2017) | ✓ | ✗ | ✓ | ✓ |
| GCL (Hyvarinen et al., 2019) | ✓ | ✗ | ✓ | ✗ |
| iVAE (Khemakhem et al., 2020b) | ✗ | ✗ | ✗ | ✗ |
| GIN (Sorrenson et al., 2020) | ✗ | ✗ | ✗ | ✗ |
| HM-NLICA (Hälvä & Hyvärinen, 2020) | ✓ | ✗ | ✓ | ✗ |
| SlowVAE (Klindt et al., 2021) | ✓ | ✗ | ✗ | ✓ |
| (Yao et al., 2021) **LEAP (Theorem 1)** | ✓ | ✓ | ✓ | ✗ |
| (Yao et al., 2021) **LEAP (Theorem 2)** | ✓ | ✓ | ✗ | ✓ |
| TimeCSL (our)† **TimeCSL (Theorem 1)** | ✓ | ✓ | ✓ | ✓ + ✗ |

## A.2 GENERALIZATION, COMPOSITIONALITY AND IRREDUCIBILITY ASSUMPTIONS

**Compositional contrast** In recent work on compositionality (Assouel et al., 2022; Zhao et al., 2022; Kurth-Nelson et al., 2022) and its importance in learning models that can generalize well to novel situations, the concept of *compositional contrast* has emerged as a powerful tool for evaluating how well a model separates information into independent, non-interacting components. This concept is particularly relevant in the context of time series analysis or image generation, where the model's ability to decompose an input into distinct parts, or "slots," can significantly impact the quality of predictions and interpretability. Compositionality ensures that each slot, or latent variable, corresponds to a specific factor or component of the data. In highly compositional models, these components do not interact with each other—each one affects a distinct aspect of the output. In contrast, non-compositional models tend to mix these components, making it harder to disentangle the factors and interpret the model's output. Evaluating how well a model adheres to compositionality principles can be challenging, as it requires quantifying how independent the slots are in their contribution to the final output. To address this, Brady et al. (2023) introduced the notion of *compositional contrast*, which measures the extent to which the model's latent variables (slots) interact when producing the final output. This measure is particularly useful in determining whether a decoder is truly compositional—that is, whether each slot contributes independently of the others, or if there are unwanted interactions between them. Before we introduce the formal definition of compositional contrast, it is important to understand the underlying principle. The intuition behind the compositional contrast is that if a model is fully compositional, each slot should affect only a specific subset of the output (e.g., one region of an image or one time series variable) and have no influence on other components. Conversely, if the model is not compositional, changes in one slot will influence multiple components of the output simultaneously, indicating that the slots are not independent. The compositional contrast function captures this idea by calculating how much the gradients of each slot (with respect to the model's output) overlap. If the gradients of different slots with respect to the same output component are non-zero, this suggests interaction between the slots, indicating a lack of compositionality. The function sums these interactions across all slots and output components, providing a single value that quantifies the degree of interaction. A lower compositional contrast value suggests higher compositionality, while a higher value indicates more interaction between slots. Formally, the compositional contrast is defined as follows:

**Definition A.1** (Compositional Contrast). Let $g_\theta : \mathcal{Z} \to \mathcal{X}$ be differentiable. The *compositional contrast* of $g_\theta$ at $\mathbf{z}$ is

$$C_{\text{comp}}(g_\theta, \mathbf{z}) = \sum_{n=1}^{N} \sum_{k=1}^{K} \sum_{j=k+1}^{K} \left\| \frac{\partial g_{\theta\,n}}{\partial \mathbf{z}_k}(\mathbf{z}) \right\| \left\| \frac{\partial g_{\theta\,n}}{\partial \mathbf{z}_j}(\mathbf{z}) \right\| . \tag{A.1}$$

This contrast function was proven to be zero if and only if $g_\theta$ is compositional according to Eq. (4.5). The function can be understood as computing each pairwise product of the (L2) norms for each pixel's gradients with respect to any two distinct slots $k \neq j$ and taking the sum. This quantity is non-negative and will only be zero if each pixel is affected by at most one slot, ensuring that $g_\theta$ satisfies Eq. (4.5). We can use this function to measure the compositional of a decoder in our experiments (see § 4), where it serves as a key indicator of how effectively the model decomposes its inputs into independent components. More empirical and theoretical details on the function can be found in Brady et al. (2023).

### A.3 Element-wise Identifiability given index support i for Piecewise Linear

In this section, we present the proof of Thm. 4.2. To establish a solid foundation for the argument, we first restate Asm 4.1, which plays a pivotal role in the proof.

**Assumption 4.1** (**Sufficient Partial Selective Pairing**). For each factor $k \in [n]$, there exist observations $(\mathbf{x}, \mathbf{x}') \in \mathcal{X}$ such that the union of the shared support indices $\mathbf{i} = \mathbf{I}(\mathbf{x}, \mathbf{x}')$ that do not include $k$ must cover all other factors. Formally:

$$\bigcup_{\mathbf{i} \in \mathcal{I} \mid k \notin \mathbf{i}} \mathbf{i} = [n] \setminus \{k\} \quad , \quad \mathcal{I} := \{\mathbf{i} \subseteq [n] \mid p(\mathbf{i}) > 0\} \tag{4.1}$$

where $\mathcal{I}$ is the set of shared support indices and $p(\mathbf{i}) := \frac{1}{\#\mathcal{X}} \cdot \# \{\mathcal{S}(\mathbf{x}) = \mathbf{i}, \mathbf{x} \in \mathcal{X}\}$ gives the probability that the factors indexed by $\mathbf{i}$ are active, with $k \notin \mathbf{i}$ inactive.

Additionally, we introduce some notation. For $\mathbf{i} \in \mathcal{I}$, we assume that the probability measure $\mathbb{P}_{\mathbf{z}_\mathbf{i}}$ admits a density with respect to the Lebesgue measure on $\mathbb{R}^{|\mathbf{i}|}$. We let $\equiv$ denote equality in the distribution.

**Theorem 4.2** (Element-wise Identifiability given index support i for Piecewise Linear $g_\theta$). *Let* $f_\phi : \mathbb{R}^{d \times n} \to \mathbb{R}^{T \times n}$ *be a continuous invertible piecewise linear function and* $\hat{g}_\theta : \mathbb{R}^{d \times n} \to \mathbb{R}^{T \times n}$ *be a continuous invertible piecewise linear function onto its image. Assume that* Asm 4.1, Asm 2.1 *holds, and the mixed observations* $(\mathbf{x}, \mathbf{x}') \overset{i.i.d.}{\sim} \mathcal{X}$, *follows the data-generating process* Eq. (2.2). *The learnable latent* $\hat{\mathbf{z}}$ *(resp.* $\hat{\mathbf{z}}'$*) of* $\mathbf{z}$ *(resp.* $\mathbf{z}'$*). If all following conditions hold:*

$$\mathbb{E}\|\hat{\mathbf{z}}\|_0 \leq \mathbb{E}\|\mathbf{z}\|_0 \quad and \quad \mathbb{E}\|\hat{\mathbf{z}}'\|_0 \leq \mathbb{E}\|\mathbf{z}'\|_0, and, \tag{4.2}$$

$$\mathcal{R}_{alig}(\hat{\mathbf{z}}, \hat{\mathbf{z}}', \mathbf{i}) := \sum_{i \in \mathbf{i}} \left| \frac{\hat{z}_i'^\top \hat{z}_i}{\|\hat{z}_i'\|_2 \|\hat{z}_i\|_2} - 1 \right| = 0. \tag{4.3}$$

*then* $\mathbf{z}$ *is identified by* $\mathbf{h} := \hat{g}_\theta^{-1}(\mathbf{x})$, *i.e.,* $\hat{g}_\theta^{-1} \circ g_\theta$ *is a permutation composed with element-wise invertible linear transformations (Def. 2.2).*

*Proof.* The proving strategy has three steps: Intuitively, based result (Kivva et al., 2022) combined with contrastivity beteween tow latent based their shared support indices $\mathbf{i}$. This means that for the data that satisfy Asm 4.1, $g_\theta(\mathbf{z})$ and $\hat{g}_\theta(\hat{\mathbf{z}})$ are equally distributed, then there exists an invertible affine transformation such that $\mathbf{h}(\mathbf{z}) = \mathbf{z}'$. Second, we use the strategy of linear identifiability (Lachapelle & Lacoste-Julien, 2022) to obtain element wise identifiabiltiy:

**Step 1) Contrastive Sparsity and Linear Identifiability given pairs i** We begin by recalling the result from Kivva et al. (2022) on the existing of an invertible function affine transformation $\mathbf{h}_k$, we adapt this for the case where if the reconstruction objective is minizzed and alignment. The theorem on identifiability of MVNs states:

**Theorem A.2.** *Let $g_\theta, g'_\theta : \mathbb{R}^{d \times n} \to \mathbb{R}^{C \times T}$ be piecewise affine functions satisfying 2.1. Let $\mathbf{z} \sim \sum_{i=1}^{J} \omega_i \mathcal{N}(\boldsymbol{\mu}_i, \boldsymbol{\Sigma}_i)$ and $\mathbf{z}' \sim \sum_{j=1}^{J'} \omega'_j \mathcal{N}(\boldsymbol{\mu}'_j, \boldsymbol{\Sigma}'_j)$ be a pair of GMMs (in reduced form). Suppose that $g_\theta(\mathbf{z})$ and $g'_\theta(\mathbf{z}')$ are equally distributed. Then there exists an invertible affine transformation $\mathbf{h} : \mathbb{R}^{d \times n} \to \mathbb{R}^{d \times n}$ such that $\mathbf{h}(\mathbf{z}) \equiv \mathbf{z}'$, i.e., $J = J'$ and for some permutation $\pi$ we have $\omega_i = \omega'_{\pi(k)}$ and $\mathbf{h} \sharp \mathcal{N}(\boldsymbol{\mu}_i, \boldsymbol{\Sigma}_i) = \mathcal{N}(\boldsymbol{\mu}'_{\pi(i)}, \boldsymbol{\Sigma}'_{\pi(i)})$.*

We recall that the transformation and the number of components can be unknown and arbitrary, and that no assumption of separation or independence is necessary for the distribution.

By Theorem C.2 (Kivva et al., 2022), since contrastive learning involves the minimisation of a contrastive loss which ensures that similar data points (positive pairs) are moved closer together and dissimilar data points (negative pairs) are moved further apart. Let the inferred latent representation $(\mathbf{z}, \mathbf{z}')$ be handled by the exact same function $\boldsymbol{f}_\phi$, and we consider the zero reconstruction under $\mathcal{R}_{aling} = 0$ for all slot indices in $\mathbf{i}$. Alongside this, contrastive loss minimization induces the distributions of $g_\theta(\mathbf{z})$ and $g_\theta(\mathbf{z}')$ to become indistinguishable on $i \in \mathbf{i}$ to be well-aligned, apart from for $k \notin \mathbf{i}$, but as we consider the Asm 4.1 on the sufficient partial pairing that will cover this factor $k$ in another pairing sample of the pair $(\mathbf{x}, \mathbf{x}')$. Thus, according to Theorem C.2 (Kivva et al., 2022), there must exist an invertible affine transformation $\mathbf{h}$ such that $\mathbf{h}(\mathbf{z}) \equiv \mathbf{z}'\mathbf{z})$. It is more likely to observe that :

$$\sum_{j=1}^{J} \omega_k g_\theta \sharp \mathcal{N}(\mu_k, \sigma_k) \sim g_\theta \sharp \boldsymbol{f}_\phi \Big( \sum_{j=1}^{J} \omega_k \mathcal{N}(\mu_k, \sigma_k) \Big). \tag{A.2}$$

In other words, minimizing to hold (i) and zeros error construction, implies a mixture model whose components are piecewise affine transformations identifiable.

**Step 2) Sparsity Pattern of an Invertible Matrix with an element-wise linear transformation**
Since $\mathbf{x} = g_\theta(\mathbf{z})$, we can rewrite perfect reconstruction as:

$$\mathbb{E}\|g_\theta(\mathbf{z}) - \hat{g}_\theta(\boldsymbol{f}_\phi(g_\theta(\mathbf{z})))\|_2^2 = 0 \tag{10}$$

This means $g_\theta$ and $\hat{g}_\theta \circ \boldsymbol{f}_\phi \circ g_\theta$ are equal $\mathbb{P}_{\mathbf{z}}$-almost everywhere. Both of these functions are continuous, $g_\theta$ by Asm 2.1, and $\hat{g}_\theta \circ \boldsymbol{f}_\phi \circ g_\theta$ because $\hat{g}_\theta$ is continuous, and $g_\theta, \boldsymbol{f}_\phi$ are linear. Since they are continuous and equal $\mathbb{P}_{\mathbf{z}}$-almost everywhere $\mathcal{Z}$, this means that they must be equal over the support of $\mathcal{Z}$, i.e.,

$$g_\theta(\mathbf{z}) = \hat{g}_\theta \circ \boldsymbol{f}_\phi \circ g_\theta(\mathbf{z}), \quad \forall \mathbf{z} \in \mathcal{Z}. \tag{11}$$

This can be easily shown by contradiction considering any slot latent $\mathbf{z}' \in \mathcal{Z}$ on which $g_\theta$ and $\hat{g}_\theta \circ \boldsymbol{f}_\phi \circ g_\theta$ are different, i.e., $\hat{g}_\theta \circ \boldsymbol{f}_\phi \circ \hat{g}_\theta(\mathbf{z}') \neq g_\theta(\mathbf{z}')$. This would imply that $(g_\theta - \hat{g}_\theta \circ \boldsymbol{f}_\phi \circ g_\theta)$, which is also a continuous function, is non-zero at $\mathbf{z}'$ and in its neighborhood, which contradict the assumption that $g_\theta$ and $\hat{g}_\theta \circ \boldsymbol{f}_\phi \circ g_\theta$ are the same $\mathbb{P}_{\mathbf{z}}$-almost everywhere. We can now apply the inverse of $\hat{g}_\theta$ on both sides to obtain

$$\hat{g}_\theta^{-1} \circ g_\theta(\mathbf{z}) = \boldsymbol{f}_\phi \circ g_\theta(\mathbf{z}) = \mathbf{h}(\mathbf{z}), \quad \forall \mathbf{z} \in \mathcal{Z}. \tag{12}$$

Since both $g_\theta$ and $\boldsymbol{f}_\phi$ are invertible linear functions, given the fisrt part of the proof (**Step** 1-App. A.3) $\mathbf{h}$ is also an invertible linear function. We now show that $\mathbf{h}$ is a permutation composed with an element-wise linear transformation. To do this, we leverage the sparsity constraint:

$$\mathbb{E}\|\hat{\mathbf{z}}\|_0 \leq \mathbb{E}\|\mathbf{z}\|_0 \tag{A.3}$$

$$\mathbb{E}\|\boldsymbol{f}_\phi(g_\theta(\mathbf{z}))\|_0 \leq \mathbb{E}\|\mathbf{z}\|_0 \tag{A.4}$$

$$\mathbb{E}\|\mathbf{h}(\mathbf{z})\|_0 \leq \mathbb{E}\|\mathbf{z}\|_0 \tag{A.5}$$

$$\tag{A.6}$$

Since $\boldsymbol{h}_k$ is invertible linear transformation, we have that $\boldsymbol{h}_k(\mathbf{z}) = \mathbf{w}_k \cdot \mathbf{z}$ and its determinant is non-zero, i.e.,

$$\det(\mathbf{h}) := \sum_{\pi \in \mathcal{P}} \text{sign}(\pi) \prod_{k=1}^{n} \mathbf{h}_{k, \pi(k)} \neq 0, \tag{A.7}$$

where $\mathcal{P}$ denotes the set of all $n$-permutations. This expression implies that at least one term in the sum is non-zero, meaning there exists a permutation $\pi \in \mathcal{P}$ such that for every $k \in [n]$, $\frac{\partial \boldsymbol{h}_k}{\partial \mathbf{z}_{\pi(k)}} \neq 0$.

Following the steps outlined in Theorem B.4 by (Lachapelle et al., 2022), and under the assumption of Asm 4.1, we extend the disentanglement analysis to our setting. This leads to the conclusion that $\mathbf{h}$ can be expressed as a permutation composed with an element-wise invertible linear transformation, based on the shared support indices $\mathbf{i}$ of the latent slot within the subspace $\mathcal{Z}_{\mathbf{i}}$. Specifically, there exists a permutation $\pi$ on $[n]$ such that, for each latent slot $k$, the corresponding permutation is given by $\pi(k)$. Since $\mathcal{I}$ is a finite set, which allows us to order its elements as $\{\mathbf{i}_1, \ldots, \mathbf{i}_{|\mathcal{I}|}\}$. Therefore, we can express $\mathcal{Z}$ as the union $\mathcal{Z} = \bigcup_{i=1}^{|\mathcal{I}|} \mathcal{Z}^{(\mathbf{i}_i)}$. While we have already shown that $\mathbf{h}$ is affine on each $\mathcal{Z}_{\mathbf{i}}$, we now demonstrate that $\mathbf{h}$ is linear on $\mathcal{Z}$, i.e., $\mathbf{h}(\mathbf{z})$ is a linear function on the entire set $\mathcal{Z} = \bigcup_{\mathbf{i} \in \mathcal{I}} \mathcal{Z}_{\mathbf{i}}$. This completes the proof. $\qquad \square$

## A.4 THE GENERATIVE PROCESS AND THE ELBO FOR MULTIVARIATES MIXTURE GAUSSIAN

We in this subsection how `TimeCSL` is trained based an a VAE process does similar to (Kivva et al., 2022; Jang et al., 2017), whcih more kind of unsupervised generative approach for clustering that performance well, we herein first describe the generative process of `TimeCSL`. Specifically, suppose there are $n$ slots latents each has a dimension $d$, an observed sample $\mathbf{x} \sim \mathcal{X}$ is generated by the following process:

---

**Algorithm 1 Generative Process**

---

1: **Input:** Prior probabilities $\boldsymbol{w}$, neural network parameters $\boldsymbol{\theta}$
2: **for** $j = 1, 2, \ldots, N$ **do**
3:     Sample slot $k \sim \text{Cat}(\boldsymbol{w})$
4:     Sample latent vector $\mathbf{z}^{(j)} \sim \mathcal{N}(\boldsymbol{\mu}_k^{(j)}, \boldsymbol{\sigma}_k^{(j)} \cdot \boldsymbol{\sigma}_k^{(j)} \mathbf{I})$
5:     Compute $[\boldsymbol{\mu}_\phi\left(\mathbf{x}^{(j)}\right); \log \boldsymbol{\sigma}_\phi\left(\mathbf{x}^{(j)}\right)^2] = \boldsymbol{g}_\theta(\mathbf{z}^{(j)})$
6:     Sample observation $\mathbf{x}_j \sim \mathcal{N}(\boldsymbol{\mu}_\theta\left(\mathbf{x}^{(j)}\right), \boldsymbol{\sigma}_\theta\left(\mathbf{x}^{(j)}\right)^2 \mathbf{I})$ or $\text{Ber}(\boldsymbol{\mu}_\theta\left(\mathbf{x}^{(j)}\right))$
7: **end for**
8: **return** $\{\mathbf{x}^{(j)}, \mathbf{z}^{(j)}, k\}_{j=1}^N$

---

**Lemma A.3.** *Given two multivariate Gaussian distributions $q(\mathbf{z}) = \mathcal{N}(\mathbf{z}; \hat{\boldsymbol{\mu}}, \hat{\boldsymbol{\sigma}}^2 \mathbf{I})$ and $p(\mathbf{z}) = \mathcal{N}(\mathbf{z}; \boldsymbol{\mu}, \boldsymbol{\sigma}^2 \mathbf{I})$, we have:*

$$\int q(\mathbf{z}) \log p(\mathbf{z}) \, d\mathbf{z} = \sum_{j=1}^J -\frac{1}{2} \log \left(2\pi \sigma_j^2\right) - \frac{\hat{\sigma}_j^2}{2\sigma_j^2} - \frac{(\hat{\mu}_j - \mu_j)^2}{2\sigma_j^2}, \qquad (A.8)$$

*where $\mu_j$, $\sigma_j$, $\hat{\mu}_j$ and $\hat{\sigma}_j$ simply denote the $j^{th}$ element of $\boldsymbol{\mu}$, $\boldsymbol{\sigma}$, $\hat{\boldsymbol{\mu}}$ and $\hat{\boldsymbol{\sigma}}$, respectively, and $J = d \times n$ is the dimensionality of $\mathbf{z}$.*

*Proof.*

$$\int q(\mathbf{z}) \log p(\mathbf{z}) \, d\mathbf{z} = \int \mathcal{N}(\mathbf{z}; \hat{\boldsymbol{\mu}}, \hat{\boldsymbol{\sigma}}^2 \mathbf{I}) \log \mathcal{N}(\mathbf{z}; \boldsymbol{\mu}, \boldsymbol{\sigma}^2 \mathbf{I}) \, d\mathbf{z}$$

$$= \int \prod_{j=1}^{J} \frac{1}{\sqrt{2\pi\hat{\sigma}_j^2}} \exp(-\frac{(z_j - \hat{\mu}_j)^2}{2\hat{\sigma}_j^2}) \log \left[ \prod_{j=1}^{J} \frac{1}{\sqrt{2\pi\sigma_j^2}} \exp(-\frac{(z_j - \mu_j)^2}{2\sigma_j^2}) \right] d\mathbf{z}$$

$$= \sum_{j=1}^{J} \int \frac{1}{\sqrt{2\pi\hat{\sigma}_j^2}} \exp(-\frac{(z_j - \hat{\mu}_j)^2}{2\hat{\sigma}_j^2}) \log \left[ \frac{1}{\sqrt{2\pi\sigma_j^2}} \exp(-\frac{(z_j - \mu_j)^2}{2\sigma_j^2}) \right] dz_j$$

$$= \sum_{j=1}^{J} \int \frac{1}{\sqrt{2\pi\hat{\sigma}_j^2}} \exp(-\frac{(z_j - \hat{\mu}_j)^2}{2\hat{\sigma}_j^2}) \left[ -\frac{1}{2} \log(2\pi\sigma_j^2) \right] dz_j - \int \frac{1}{\sqrt{2\pi\hat{\sigma}_j^2}} \exp(-\frac{(z_j - \hat{\mu}_j)^2}{2\hat{\sigma}_j^2}) \frac{(z_j - \mu_j)^2}{2\sigma_j^2} \, dz_j$$

$$= \sum_{j=1}^{J} -\frac{1}{2} \log(2\pi\sigma_j^2) - \int \frac{1}{\sqrt{2\pi\hat{\sigma}_j^2}} \exp(-\frac{(z_j - \hat{\mu}_j)^2}{2\hat{\sigma}_j^2}) \frac{(z_j - \hat{\mu}_j)^2 + 2(z_j - \hat{\mu}_j)(\hat{\mu}_j - \mu_j) + (\hat{\mu}_j - \mu_j)^2}{2\hat{\sigma}_j^2} \frac{\hat{\sigma}_j^2}{\sigma_j^2} \, dz_j$$

$$= \boldsymbol{b} - \frac{\hat{\sigma}_j^2}{\sigma_j^2} \int \frac{1}{\sqrt{2\pi\hat{\sigma}_j^2}} \exp(-\frac{(z_j - \hat{\mu}_j)^2}{2\hat{\sigma}_j^2}) \frac{(z_j - \hat{\mu}_j)^2}{2\hat{\sigma}_j^2} \, dz_j - \int \frac{1}{\sqrt{2\pi\hat{\sigma}_j^2}} \exp(-\frac{(z_j - \hat{\mu}_j)^2}{2\hat{\sigma}_j^2}) \frac{(\hat{\mu}_j - \mu_j)^2}{2\sigma_j^2} \, dz_j$$

$$= \boldsymbol{b} - \frac{\hat{\sigma}_j^2}{\sigma_j^2} \int \frac{1}{\sqrt{2\pi}} \exp(-\frac{x_j^2}{2}) \frac{x_j^2}{2} \, dx_j - \frac{(\hat{\mu}_j - \mu_j)^2}{2\sigma_j^2}$$

$$= \boldsymbol{b} - \frac{\hat{\sigma}_j^2}{\sigma_j^2} \int \frac{1}{\sqrt{2\pi}} (-\frac{x_j}{2}) \, d(\exp(-\frac{x_j^2}{2})) - \frac{(\hat{\mu}_j - \mu_j)^2}{2\sigma_j^2}$$

$$= \boldsymbol{b} - \frac{\hat{\sigma}_j^2}{\sigma_j^2} \left[ \frac{1}{\sqrt{2\pi}} (-\frac{x_j}{2}) \exp(-\frac{x_j^2}{2}) \Big|_{-\infty}^{\infty} - \int \frac{1}{\sqrt{2\pi}} \exp(-\frac{x_j^2}{2}) \, d(-\frac{x_j}{2}) \right] - \frac{(\hat{\mu}_j - \mu_j)^2}{2\sigma_j^2}$$

$$= \sum_{j=1}^{J} -\frac{1}{2} \log(2\pi\sigma_j^2) - \frac{\hat{\sigma}_j^2}{2\sigma_j^2} - \frac{(\hat{\mu}_j - \mu_j)^2}{2\sigma_j^2}$$

where $\boldsymbol{b}$ denotes $\sum_{j=1}^{J} -\frac{1}{2} \log(2\pi\sigma_j^2)$ for simplicity.

$\square$

### A.4.1 VARIATIONAL LOWER BOUND FOR TIMECSL

A `TimeCSL` instance is tuned to maximize the likelihood of the given data points. Given the generative process in Section A.4, by using Jensen's inequality, the log-likelihood of `TimeCSL` can be written as:

$$\log p(\mathbf{x}) = \log \int_{\mathbf{z}} \sum_k p(\mathbf{x}, \mathbf{z}, k) d\mathbf{z}$$

$$\geq E_{q(\mathbf{z}, k | \mathbf{x})}[\log \frac{p(\mathbf{x}, \mathbf{z}, k)}{q(\mathbf{z}, k | \mathbf{x})}] = \mathcal{L}_{\text{ELBO}}(\mathbf{x}) \tag{A.9}$$

where $\mathcal{L}_{\text{ELBO}}$ is the evidence lower bound (ELBO), $q(\mathbf{z}, k | \mathbf{x})$ is the variational posterior to approximate the true posterior $p(\mathbf{z}, k | \mathbf{x})$. In `TimeCSL`, we assume $q(\mathbf{z}, k | \mathbf{x})$ to be a mean-field distribution and can be factorized as:

$$q(\mathbf{z}, k | \mathbf{x}) = q(\mathbf{z} | \mathbf{x}) q(k | \mathbf{x}). \tag{A.10}$$

Then, according to Equation A.10, the $\mathcal{L}_{\text{ELBO}}(\mathbf{x})$ in Equation A.9 can be rewritten as:

$$
\begin{aligned}
\mathcal{L}_{\text{ELBO}}(\mathbf{x}) &= E_{q(\mathbf{z},k|\mathbf{x})}\left[\log \frac{p(\mathbf{x},\mathbf{z},k)}{q(\mathbf{z},k|\mathbf{x})}\right] \\
&= E_{q(\mathbf{z},k|\mathbf{x})}\left[\log p(\mathbf{x},\mathbf{z},k) - \log q(\mathbf{z},k|\mathbf{x})\right] \\
&= E_{q(\mathbf{z},k|\mathbf{x})}[\log p(\mathbf{x}|\mathbf{z}) + \log p(\mathbf{z}|k) \\
&\quad + \log p(k) - \log q(\mathbf{z}|\mathbf{x}) - \log q(k|\mathbf{x})]
\end{aligned}
\tag{A.11}
$$

In `TimeCSL`, similar to VAE, we use a neural network $g$ to model $q(\mathbf{z}|\mathbf{x})$:

$$
[\hat{\boldsymbol{\mu}}; \log \hat{\boldsymbol{\sigma}}^2] = \boldsymbol{f}_\phi(\mathbf{x};\boldsymbol{\phi}) \tag{A.12}
$$

$$
q(\mathbf{z}|\mathbf{x}) = \mathcal{N}(\mathbf{z};\hat{\boldsymbol{\mu}},\hat{\boldsymbol{\sigma}}^2\mathbf{I}) \tag{A.13}
$$

where $\phi$ is the parameter of network $g$.

By substituting the terms in Equation A.11 and using the SGVB estimator and the *reparameterization* trick, the $\mathcal{L}_{\text{ELBO}}(\mathbf{x})$ can be rewritten as: [5]

$$
\begin{aligned}
\mathcal{L}_{\text{ELBO}}(\mathbf{x}) = &\frac{1}{N}\sum_{l=1}^{N}\sum_{i=1}^{C\times T}\left[x_i \log \boldsymbol{\mu}_{x_i}^{(l)} + (1-x_i)\log \boldsymbol{f}_\phi(1-\boldsymbol{\mu}_{x_i}^{(l)})\right] \\
&-\frac{1}{2}\sum_{k=1}^{n}\gamma_k\sum_{j=1}^{J}\left(\log \boldsymbol{\sigma}_k^2|_j + \frac{\hat{\boldsymbol{\sigma}}^2|_j}{\boldsymbol{\sigma}_k^2|_j} + \frac{\left(\hat{\boldsymbol{\mu}}|_j - \boldsymbol{\mu}_k|_j\right)^2}{\boldsymbol{\sigma}_k^2|_j}\right) \\
&+\sum_{k=1}^{n}\gamma_k \log \frac{w_k}{\gamma_k} + \frac{1}{2}\sum_{j=1}^{J}\left(1+\log \hat{\boldsymbol{\sigma}}^2|_j\right)
\end{aligned}
\tag{A.14}
$$

where $N$ is the number of Monte Carlo samples in the SGVB estimator, $C \times T$ is the dimensionality of $\mathbf{x}$, $n$ is number of slots or factors, and $\boldsymbol{\mu}_x^{(l)}$, $x_i$ is the $i^{\text{th}}$ element of $\mathbf{x}$, $J$ is the dimensionality of $\boldsymbol{\mu}_k$, $\boldsymbol{\sigma}_k^2$, $\hat{\boldsymbol{\mu}}$ and $\hat{\boldsymbol{\sigma}}^2$, and $*|_j$ denotes the $j^{\text{th}}$ element of $*$, $n$ is the number of slots, $w_k$ is the prior probability of slot $k$, and $\gamma_k$ denotes $q(k|\mathbf{x})$ for simplicity. In Equation A.14, we compute $\boldsymbol{\mu}_x^{(l)}$ as

$$
\boldsymbol{\mu}_x^{(l)} = f_\phi(\mathbf{z}^{(l)};\theta), \tag{A.15}
$$

where $\mathbf{z}^{(l)}$ is the $l^{\text{th}}$ sample from $q(\mathbf{z}|\mathbf{x})$ by Equation A.13 to produce the Monte Carlo samples. According to the *reparameterization* trick, $\mathbf{z}^{(l)}$ is obtained by

$$
\mathbf{z}^{(l)} = \hat{\boldsymbol{\mu}} + \hat{\boldsymbol{\sigma}} \circ \boldsymbol{\epsilon}^{(l)}, \tag{A.16}
$$

where $\boldsymbol{\epsilon}^{(l)} \sim \mathcal{N}(0,\mathbf{I})$, $\circ$ is element-wise multiplication, and $\hat{\boldsymbol{\mu}}$, $\hat{\boldsymbol{\sigma}}$ are derived by Equation A.12. We now describe how to formulate $\gamma_c \triangleq q(k|\mathbf{x})$ in Equation A.14 to maximize the ELBO. Specifically, $\mathcal{L}_{\text{ELBO}}(\mathbf{x})$ can be rewritten as:

$$
\begin{aligned}
\mathcal{L}_{\text{ELBO}}(\mathbf{x}) &= E_{q(\mathbf{z},c|\mathbf{x})}\left[\log \frac{p(\mathbf{x},\mathbf{z},c)}{q(\mathbf{z},c|\mathbf{x})}\right] \\
&= \int_{\mathbf{z}}\sum_c q(k|\mathbf{x})q(\mathbf{z}|\mathbf{x})\left[\log \frac{p(\mathbf{x}|\mathbf{z})p(\mathbf{z})}{q(\mathbf{z}|\mathbf{x})} + \log \frac{p(k|\mathbf{z})}{q(k|\mathbf{x})}\right]d\mathbf{z} \\
&= \int_{\mathbf{z}}q(\mathbf{z}|\mathbf{x})\log \frac{p(\mathbf{x}|\mathbf{z})p(\mathbf{z})}{q(\mathbf{z}|\mathbf{x})}d\mathbf{z} - \int_{\mathbf{z}}q(\mathbf{z}|\mathbf{x})D_{KL}(q(k|\mathbf{x})||p(k|\mathbf{z}))d\mathbf{z}
\end{aligned}
\tag{A.17}
$$

Once the training is done by maximizing the ELBO w.r.t the parameters of $\{\boldsymbol{\pi},\boldsymbol{\mu}_k,\boldsymbol{\sigma}_k,\boldsymbol{\theta},\boldsymbol{\phi}\}$, $k \in \{1,\ldots,K\}$, a latent representation $\mathbf{z}$ can be extracted for each observed sample $\mathbf{x}$. This is done by Equation A.12 and Equation A.13.

---

[5]This is the case when the observation $\mathbf{x}$ is binary. For the real-valued situation, the ELBO can be obtained in a similar way.

### A.4.2 THE EQUIVALENCE BETWEEN MATRIX NORMAL AND MULTIVARIATE NORMAL DISTRIBUTIONS

In our formulation, we use a vectorization of the matrix $\mathbf{z} \in \mathbb{R}^{d \times n}$, which follows a multivariate Gaussian model. We now show that this can also be interpreted as a Matrix Normal distribution. The equivalence between the Matrix Normal and the Multivariate Normal density functions can be established using properties of the trace and the Kronecker product.

*Proof.* Let $\mathbf{z}$ be modeled as a mixture of $J$ Matrix Normal distributions. Each component of this mixture is characterized by a mean matrix $\boldsymbol{\mu}_j \in \mathbb{R}^{d \times n}$ and a covariance matrix $\boldsymbol{\Sigma}_j = \boldsymbol{\Sigma}_n \otimes \Sigma_n \in \mathbb{R}^{d \times d} \otimes \mathbb{R}^{n \times n}$, where $\boldsymbol{\Sigma}_n$ and $\boldsymbol{\Sigma}_n$ are the row and column covariance matrices, respectively. The probability density function of $\mathbf{z}$ is thus given by

$$f_{\mathbf{z}}(\mathbf{z}) = \sum_{j=1}^{J} \omega_j \mathcal{N}(\mathbf{z} \mid \boldsymbol{\mu}_j, \boldsymbol{\Sigma}_j),$$

where $\omega_j$ are the mixing weights such that $\omega_j > 0$ and $\sum_{j=1}^{J} \omega_j = 1$.

The Matrix Normal distribution is defined as

$$\mathcal{N}(\mathbf{z} \mid \boldsymbol{\mu}_j, \boldsymbol{\Sigma}_j) = \frac{1}{(2\pi)^{\frac{dn}{2}} |\boldsymbol{\Sigma}_j|^{\frac{n+d}{2}}} \exp\left( -\frac{1}{2} \operatorname{tr}\left[ \boldsymbol{\Sigma}_d^{-1}(\mathbf{z} - \boldsymbol{\mu}_j)^T \boldsymbol{\Sigma}_n^{-1}(\mathbf{z} - \boldsymbol{\mu}_j) \right] \right),$$

where $\mathbf{z}$ is a $d \times n$ matrix, and the covariance matrix $\boldsymbol{\Sigma}_j$ is the Kronecker product $\boldsymbol{\Sigma}_n \otimes \boldsymbol{\Sigma}_n$, with $\boldsymbol{\Sigma}_n$ and $\boldsymbol{\Sigma}_n$ being the covariance matrices of the rows and columns of $\mathbf{z}$, respectively.

To connect the Matrix Mixture Normal distribution with the Mixture of Multivariate Normal distributions, we vectorize the matrix $\mathbf{z}$. The vectorization of a matrix $\mathbf{z} \in \mathbb{R}^{d \times n}$ is given by

$$\operatorname{vec}(\mathbf{z}) = \begin{bmatrix} z_{11} & z_{21} & \cdots & z_{d1} & z_{12} & \cdots & z_{dn} \end{bmatrix}^T \in \mathbb{R}^{1 \times (d \cdot n)}$$

where $\mathbf{z}_i$ denotes the $i$-th column of $\mathbf{z}$, and the resulting vector $\operatorname{vec}(\mathbf{z})$ is a $d \cdot n$-dimensional vector.

Now, substituting the vectorized form of $\mathbf{z}$ into the Matrix Normal distribution, we have

$$\mathcal{N}(\operatorname{vec}(\mathbf{z}) \mid \operatorname{vec}(\boldsymbol{\mu}_j), \boldsymbol{\Sigma}_j) = \frac{1}{(2\pi)^{\frac{dn}{2}} |\boldsymbol{\Sigma}_j|^{\frac{d+n}{2}}} \exp\left( -\frac{1}{2} \bar{\mathbf{z}}^T \boldsymbol{\Sigma}_j^{-1} \bar{\mathbf{z}} \right), \tag{A.18}$$

where $\bar{\mathbf{z}} = \operatorname{vec}(\mathbf{z}) - \operatorname{vec}(\boldsymbol{\mu}_j)$. Next, observe that the mixture model for $\mathbf{z}$ in the original form becomes

$$f_{\mathbf{z}}(\mathbf{z}) = \sum_{j=1}^{J} \omega_j \mathcal{N}(\operatorname{vec}(\mathbf{z}) \mid \operatorname{vec}(\boldsymbol{\mu}_j), \boldsymbol{\Sigma}_n \otimes \Sigma_n), \tag{A.19}$$

which is a mixture of multivariate normal distributions in the vectorized space $\mathbb{R}^{d \cdot n}$. This shows that the Matrix Mixture Normal distribution is equivalent to a Mixture of Multivariate Normal distributions upon vectorization. To complete the proof, we use the determinant property of the Kronecker product:

$$|\boldsymbol{\Sigma}_n \otimes \boldsymbol{\Sigma}_n| = |\boldsymbol{\Sigma}_n|^n |\boldsymbol{\Sigma}_n|^d. \tag{A.20}$$

Thus, the determinant of the covariance matrix $\boldsymbol{\Sigma}_n \otimes \boldsymbol{\Sigma}_n$ can be written as the product of the determinants of $\boldsymbol{\Sigma}_n$ and $\boldsymbol{\Sigma}_n$, raised to the appropriate powers. This confirms that the matrix mixture normal distribution is indeed equivalent to the mixture of multivariate normal distributions. $\qquad \square$

## A.5 STRUCTURAL SPARSITY AND SUFFICIENT PARTIAL SELECTIVE PAIRING ASSUMPTIONS

**Comparison of Structural Sparsity and Sufficient Partial Selective Pairing Assumptions**   We compare two important assumptions in the context of source separation: the Structural Sparsity assumption from (Ng et al., 2023) and the Sufficient Partial Selective Pairing assumption. The Structural Sparsity assumption for sources $\boldsymbol{y} = \{\boldsymbol{y}_1, \ldots, \boldsymbol{y}_n\}$ in the mixing matrix $\boldsymbol{A}$ stipulates that for any pair of sources $k$ and $\ell$, their supports (denoted $\mathrm{supp}(\boldsymbol{y}_k)$ and $\mathrm{supp}(\boldsymbol{y}_\ell)$) must differ in at least two observed variables, i.e.,

$$|\mathrm{supp}(\boldsymbol{y}_k) \cup \mathrm{supp}(\boldsymbol{y}_\ell)| - |\mathrm{supp}(\boldsymbol{y}_k) \cap \mathrm{supp}(\boldsymbol{y}_\ell)| > 1$$

Here, $\mathrm{supp}(\boldsymbol{y}_k)$ represents the indices of the observed variables affected by the source $\boldsymbol{y}_k$. This assumption ensures that the sources $\boldsymbol{y}_k$ and $\boldsymbol{y}_\ell$ are distinguishable in terms of the observed variables they influence.

**Example of Structural Sparsity Assumption**   Consider a scenario where we have three sources $\boldsymbol{y}_1, \boldsymbol{y}_2, \boldsymbol{y}_3$ and four observed variables $\mathbf{x}_1, \mathbf{x}_2, \mathbf{x}_3, \mathbf{x}_4$. The observed data $\mathbf{x} = [\mathbf{x}_1, \mathbf{x}_2, \mathbf{x}_3, \mathbf{x}_4]$ is a mixture of the sources. The supports for the sources are defined as follows:

$$\mathrm{supp}(\boldsymbol{y}_1) = \{1\}, \quad \mathrm{supp}(\boldsymbol{y}_2) = \{2\}, \quad \mathrm{supp}(\boldsymbol{y}_3) = \{3\}$$

For the Structural Sparsity assumption to hold between sources $\boldsymbol{y}_1$ and $\boldsymbol{y}_2$, the supports must differ in at least two observed variables. For example, we have:

$$|\mathrm{supp}(\boldsymbol{y}_1) \cup \mathrm{supp}(\boldsymbol{y}_2)| - |\mathrm{supp}(\boldsymbol{y}_1) \cap \mathrm{supp}(\boldsymbol{y}_2)| = 2 - 0 = 2$$

This satisfies the assumption, as the supports of sources $\boldsymbol{y}_1$ and $\boldsymbol{y}_2$ differ in at least two variables. If, however, both sources share the same support:

$$\mathrm{supp}(\boldsymbol{y}_1) = \{1\}, \quad \mathrm{supp}(\boldsymbol{y}_2) = \{1\}$$

Then the assumption would not hold because the supports are identical, and they do not differ by at least two observed variables.

**Sufficient Partial Selective Pairing Assumption (Assumption 1)**   The Sufficient Partial Selective Pairing assumption requires that for each factor $k \in [n]$, there exist observations $(\mathbf{x}, \mathbf{x}') \in \mathcal{X}$ such that the union of the shared support indices $\mathbf{i} = \mathbf{I}(\mathbf{x}, \mathbf{x}')$ that do not include $k$ must cover all other factors. Formally, we have:

$$\bigcup_{\mathbf{i} \in \mathcal{I} \mid k \notin \mathbf{i}} \mathbf{i} = [n] \setminus \{k\}, \quad \mathcal{I} := \{\mathbf{i} \subseteq [n] \mid p(\mathbf{i}) > 0\} \tag{A.21}$$

Here, $\mathcal{I}$ is the set of shared support indices, and $p(\mathbf{i})$ is the probability that the factors indexed by $\mathbf{i}$ are active, with $k \notin \mathbf{i}$ inactive. The assumption ensures that when one factor is inactive, the shared support indices from the remaining factors provide enough information to reconstruct all active factors.

**Example of Sufficient Partial Selective Pairing Assumption**   In the same scenario with three sources $\boldsymbol{y}_1, \boldsymbol{y}_2, \boldsymbol{y}_3$ and observed variables $\mathbf{x}_1, \mathbf{x}_2, \mathbf{x}_3, \mathbf{x}_4$, we can define the shared support indices for each observation. Let's assume that the following shared support indices hold:

- Observation 1: $\mathbf{i} = \{1, 2\}$ - Observation 2: $\mathbf{i} = \{2, 3\}$ - Observation 3: $\mathbf{i} = \{3, 4\}$

Now, for the Sufficient Partial Selective Pairing assumption to hold for factor $k = 1$, we must ensure that the union of the shared supports where factor 1 is inactive covers all other factors. For example, if we exclude $k = 1$, the union of the shared supports for the remaining factors should cover $\boldsymbol{y}_2$ and $\boldsymbol{y}_3$:

$$\bigcup_{\mathbf{i}|1\notin\mathbf{i}} \mathbf{i} = \{2, 3, 4\} = [2, 3, 4]$$

This satisfies the assumption because when $\boldsymbol{y}_1$ is inactive, the shared support indices from $\boldsymbol{y}_2$ and $\boldsymbol{y}_3$ cover all remaining factors.

**Why the Sufficient Partial Selective Pairing Assumption is More Flexible**

- It does not require the supports of every pair of sources to differ by exactly two observed variables.

- It only requires that when one factor is inactive, the shared support indices must still cover all other active factors, which allows for more overlap between the supports of different sources.

- This assumption is better suited for real-world scenarios where the supports of factors may not be completely distinct but still provide enough information to disentangle the factors.

In contrast, the Structural Sparsity assumption proposed in (Ng et al., 2023) can be too strict in cases where factors share common supports, and it would fail to identify factors in such cases.

**Example.1 (Assumption-1 fails)** This ensures distinct influences across observed variables. If the supports are nearly identical, Assumption-1 fails. For example, consider the mixing matrix $\boldsymbol{A}$:

$$\begin{bmatrix} \mathbf{x}_1(t) \\ \mathbf{x}_2(t) \\ \mathbf{x}_3(t) \\ \mathbf{x}_4(t) \end{bmatrix} = \begin{bmatrix} 1 & 0.5 & 0 & 0.2 \\ 0.3 & 1 & 0.4 & 0 \\ 0 & 0.2 & 1 & 0.5 \\ 0.1 & 0 & 0.6 & 1 \end{bmatrix} \begin{bmatrix} \boldsymbol{y}_1(t) \\ \boldsymbol{y}_2(t) \\ \boldsymbol{y}_3(t) \\ \boldsymbol{y}_4(t) \end{bmatrix} + \epsilon$$

with supports $\text{supp}(\mathbf{a}_1) = \{1, 2, 4\}$, $\text{supp}(\mathbf{a}_2) = \{1, 2, 3\}$, $\text{supp}(\mathbf{a}_3) = \{2, 3, 4\}$, and $\text{supp}(\mathbf{a}_4) = \{1, 3, 4\}$. For $\boldsymbol{y}_1$ and $\boldsymbol{y}_2$, the difference in support is 2 (validating Assumption-1), as is the case for $\boldsymbol{y}_3$ and $\boldsymbol{y}_4$. However, the significant overlap in the observed variables they influence ($\boldsymbol{y}_1$ and $\boldsymbol{y}_2$ both affect $\mathbf{x}_1(t)$, $\mathbf{x}_2(t)$, and $\boldsymbol{y}_3$ and $\boldsymbol{y}_4$ affect $\mathbf{x}_3(t)$, $\mathbf{x}_4(t)$) limits the ability to uniquely identify each source, pointing to a practical challenge in real-world data.

## B  EXPERIMENTS AND IMPLEMENTATION SETTINGS

### B.1  IMPLEMENTATION SOURCE. (TIMECSL-LIB)

We have implemented the ResTimeCSL architecture from scratch, and our code is available at `https://anonymous.4open.science/r/TimeCSL-4320`. Some components of our code are inspired by the following works:

- The GMM-based VAE sampling is inspired by VaDE (Jiang et al., 2016), and we adapted the implementation from `https://github.com/mperezcarrasco/Pytorch-VaDE`.

- For the Diffusion model D3VAE (Li et al., 2023), we utilized the authors' implementation from `https://github.com/PaddlePaddle/PaddleSpatial/tree/main/research/D3VAE`.

- Regarding the methods listed in Tab. 3, the TCL model was adapted from `https://github.com/hmorioka/TCL/tree/master/tcl`, while the other models are derived from `https://github.com/rpatrik96/nl-causal`.

- For iVAE (Khemakhem et al., 2020b), we used the implementation available at `https://github.com/MatthewWilletts/algostability`.

Our experiments were conducted with 5 different random seeds, and we report the average results along with standard deviations. The experiments were run using 8 NVIDIA A100 GPUs.

## B.2 DATASETS.

In this section, we provide details about the datasets used for our experiments. We consider both real-world and synthetic datasets, each with specific characteristics relevant to the study. The table below summarizes the key properties of these datasets, including the number of samples, input dimensions, the number of sources/factors, and the names of the factors. The real-world datasets include REDD, REFIT, and UKDALE, which are commonly used in energy consumption modeling. Additionally, we employ synthetic datasets (Synthetic-1, Synthetic-2, and Synthetic-3) to simulate various scenarios with varying factors and input sizes. These datasets allow for comprehensive testing of our proposed method across different contexts.

Table 4: Synthetic and real-world datasets

| Dataset | # Samples | Input Dim | # Sources/Factors | Factors name |
|---------|-----------|-----------|-------------------|--------------|
| REDD | 5400 | 256 | 3 | {FR, DW, WM, HTR, LT} |
| REFIT | 1299 | 256 | 5 | {FR, DW, WM, HTR, LT} |
| UKDALE | 1300 | 256 | 5 | {FR, DW, WM, HTR, LT} |
| Synthetic-1 | 12000 | 24 | 3 | {FR, LT, HTR} |
| Synthetic-2 | 11000 | 96 | 5 | {FR, LT, HTR} |
| Synthetic-3 | 11000 | 64 | 3 | {FR, LT, HTR} |
| Synthetic-4 | 23000 | 256 | 5 | {FR, DW, WM, HTR, LT} |

## B.3 CONTRASTIVE PARTIAL SELECTIVE PAIRING - DATA AUGMENTATIONS

Four augmentations were sequentially applied to all contrastive methods' pipeline branches. The parameters from the random search are: 1) **Crop and delay:** applied with a 0.5 probability and a minimum size of $50\%$ of the initial sequence. 2) **Cutout or Masking:** time cutout of 5 steps with a 0.8 probability. 3) **Channel Masks powers:** each time series is randomly masked out with a 0.4 probability. 4) **Gaussian noise:** random Gaussian noise is added to window input $\mathbf{x}$ with a standard deviation form 0.1 to 0.3. Further details in App. B.3. Also in our experiments, we utilize a composition of three data augmentations, applied in the following order - scaling, shifting, and jittering, activating with a probability of 0.3 to 0.5.

**Scaling** The time-series is scaled by a single random scalar value, obtained by sampling $\epsilon \sim \mathcal{N}(0, 0.5)$, and each time step is $\mathbf{x}'_t = \epsilon x_t$.

**Shifting** The time-series is shifted by a single random scalar value, obtained by sampling $\epsilon \sim \mathcal{N}(0, 0.5)$ and each time step is $\mathbf{x}'_t = x_t + \epsilon$.

**Jittering** I.I.D. Gaussian noise is added to each time step, from a distribution $\epsilon_t \sim \mathcal{N}(0, 0.5)$, where each time step is now $\mathbf{x}'_t = x_t + \epsilon_t$.

## B.4 IMPLEMENTATION OF METRICS AND STUDY CASE

Previous work has relied on the Mean Correlation Coefficient (MCC) as a metric to quantify identifiability. For consistency with previous work, we report this metric, but also propose a new metric to quantify identifiability up to an affine transformation. There are two challenges in designing such a metric: Firstly, for two Gaussian mixtures, standard distance metrices such as TV-distance or KL-divergence do not have a closed form. Secondly, we need to find an affine map $A$ that best aligns a pair of Gaussian mixtures. Therefore, developing a metric to quantify identifiability up to an affine transformation has natural challenges. We propose $\boldsymbol{d}_{\text{aff}, L2}$, defined below, as an additional metric in this setting.

### B.4.1    ALIGNMENT PRIOR TO MEASURING WEAK MCC

We seek an affine map $\Gamma$ to align two GMMs using two methods. One approach, used in previous works on MCC, is Canonical Correlation Analysis (CCA). Alternatively, we explore a different method. For two GMMs, we iterate over all permutations of the components, and for each permutation, we compute the optimal map $\Gamma$ that aligns the components. While ideally $\Gamma$ would align both the means and the covariance matrices, solving this as an optimization problem is challenging. Thus, we focus on aligning the means of the first GMM to those of the second GMM. The map $\Gamma$ is found by solving the least-squares problem:

$$\min_{\Gamma} \sum_i \| \boldsymbol{\mu}_1^{(i)} - \Gamma \boldsymbol{\mu}_2^{(i)} \|^2 \tag{B.1}$$

This can be efficiently solved using Singular Value Decomposition (SVD). Empirically, aligning the means provides good results.

### B.4.2    MEASURING IDENTIFIABILITY STRONG-MCC AND WEAK-MCC

The other metric we consider is the Mean Correlation Coefficient (MCC) metric which had been used in prior works (Khemakhem et al., 2020a). There are two versions of MCC that have been used:

1. The *weak* MCC is defined to be the MCC after alignment via the affine map $\Gamma$ transformation see App. B.4.1.

2. The *strong* MCC is defined to be the MCC before alignment.

Furthermore, in this work, we consider two different metrics. For a pair of distributions $p_1, p_2$, we define $\boldsymbol{d}_{\text{aff},L2}$ loss as

$$\boldsymbol{d}_{\text{aff},L2}(p_1, p_2) = \min_{\substack{A:\mathbb{R}^m \to \mathbb{R}^m, \\ \text{affine}}} \Delta_{L_2}(\Gamma_\sharp p_1, p_2), \quad \text{where} \quad \Delta_{L_2}(p_1, p_2) = \frac{\|p_1 - p_2\|_{L_2}}{\|p_1\|_{L_2}^{1/2} \|p_2\|_{L_2}^{1/2}} \tag{B.2}$$

In our experiments, we report both the strong MCC and weak MCC. Moreover, all reported MCC s are out-of-sample, i.e. the optimal affine map $\Gamma$ is computed over half the dataset and then reused for the other half of the dataset.

### B.4.3    MEASURING DISENTANGLEMENT OF THE LEARNED REPRESENTATION

In implementing the disentanglement metrics, we adhere to the methodology outlined in (Locatello et al., 2019), expanding it to accommodate time series data. For the computation of DCI metrics, we employ a gradient boosted tree from the scikit-learn package.

**$\beta$-VAE Metric**    Disentanglement is then measured as the accuracy of a linear classifier that predicts the index of the fixed factor based on the coordinate-wise sum of absolute differences between the representation vectors in the two mini-batches. (Higgins et al., 2016) suggest fixing a random attributes of variation in the underlying generative model and sampling two mini-batches of observations $\mathbf{x}$. We sample two batches of 256 points with a random factor fixed to a randomly sampled value across the two batches, and the others varying randomly. We compute the mean representations for these points and take the absolute difference between pairs from the two batches. We then average these 64 values to form the features of a training (or testing) point.

**FactorVAE Metric (Kim & Mnih, 2019)**    (Kim & Mnih, 2019) address several issues with this metric by using a majority vote classifier that predicts the index of the fixed ground-truth attribute based on the index of the representation vector with the least variance. First, we estimate the variance of each latent dimension by embedding $10k$ random samples from the data set, excluding collapsed dimensions with variance smaller than .05. Second, we generate the votes for the majority vote classifier by sampling a batch of 64 points, all with a factor fixed to the same random value. Third, we compute the variance of each dimension of their latent representation and divide it by the variance of that dimension computed on the data without interventions. The training point for the majority vote classifier consists of the index of the dimension with the smallest normalized variance. We train on $10k$ points and evaluate on $5k$ points.

**Mutual Information Gap Metric (Chen et al., 2018b)** $\beta$-VAE metric and the FactorVAE metric are neither general nor unbiased as they depend on some hyperparameters (Chen et al., 2018b). They compute the mutual information between each ground-truth factor and each dimension in the computed representation $r(\mathbf{x})$. For each ground-truth factor $\mathbf{z}_k$, they then consider the two dimensions in $r(\mathbf{x})$ that have the highest and second highest mutual information with $\mathbf{z}_k$. The Robust Mutual Information Gap (MIG) is then defined as the average, normalized difference between the highest and second highest mutual information of each factor with the dimensions of the representation. The original metric was proposed evaluating the sampled representation. Instead, we consider the mean representation, in order to be consistent with the other metrics. We estimate the mutual information by binning each dimension of the representations. Then, the score is computed as follows:

$$RMIG = \frac{1}{K} \sum_{k=1}^{K} [I(v_{jk}, z_k) - \max I(v_j, z_k)]$$

Where $\mathbf{z}_k$ is a factor of variation, $\mathbf{v}_i$ is a dimension of the latent representation. The MIG score of all factors are averaged to report one score.

**Disentanglement, Completeness and Informativeness (DCI)** In (Carbonneau et al., 2022), a framework is proposed to evaluate disentangled representations using metrics for modularity, compactness, and explicitness, referred to as disentanglement, completeness, and informativeness. Regressors predict factors from codes, with modularity and compactness estimated by importance weights $R_{ij}$. These weights are computed using a lasso regressor or random forests. The compactness for factor $\mathbf{v}_i$ is defined as:

$$C_i = 1 + \sum_{j=1}^{d} p_{ij} \log_d p_{ij}, \quad p_{ij} = \frac{R_{ij}}{\sum_{k=1}^{d} R_{ik}}.$$

Compactness for the entire representation is the average over all factors. The modularity for code dimension $\mathbf{z}_j$ is:

$$D_j = 1 + \sum_{i=1}^{M} p_{ij} \log_M p_{ij}, \quad p_{ij} = \frac{R_{ij}}{\sum_{k=1}^{M} R_{kj}}.$$

The modularity score is the weighted average over all code dimensions, with weights $\rho_j$ reflecting their importance in predicting factors. Explicitness is defined by the MSE of the regressor, normalized between 0 and 1:

$$\text{Explicitness} = 1 - 6 \cdot \text{MSE}, \quad \text{MSE} = E[(\mathbf{x} - \mathbf{y})^2] = \frac{1}{6}.$$

.

**Time Disentanglement Score TDS** Time series data often exhibit variations that may not always align with conventional metrics, especially when considering the presence or absence of underlying attributes. To address this challenge, (Oublal et al., 2024) introduce the Time Disentanglement Score (TDS), a metric designed to assess the disentanglement of attributes in time series data. The foundation of TDS lies in an Information Gain perspective, which measures the reduction in entropy when an attribute is present compared to when it's absent.

$$TDS = \frac{1}{dim(\mathbf{z})} \sum_{n \neq m} \sum_{k} \frac{||z_m - z_{n,k}^+||^2}{\text{Var}[z_m]}, \tag{B.3}$$

In the context of TDS, we augment factor $m$ in a time series window $\mathbf{x}$ with a specific objective: to maintain stable entropy when the factor is present and reduce entropy when it's absent. This augmentation aims to capture the essence of attribute-related information within the data.

### B.5 RESTIMECSL ARCHITECTURE

The architecture employs multiple ResTimeCSL residual units Fig. 8 to model both the encoder and decoder for temporal sequential data. The input size is $T = 256$ (time steps) with $C = 1$ (features).

The encoder compresses the input into a latent representation of size $n = 5 \times d = 16$, while the decoder reconstructs the sequence into an output of size $T = 256 \times n = 5$. An additive layer is applied after decoding to sum the $n$ components at each time step $t$, ensuring the output matches the input dimensions. Let $\mathbf{x} \in \mathbb{R}^{T \times C}$ represent the input sequence. A linear patching operation is applied to preprocess the input: $\mathbf{x}_{\text{patch}} = \text{LinearPatching}(\mathbf{x})$. The encoder comprises multiple stacked "ResTimeCSL" residual units to map the input into a latent representation $\mathbf{z} \in \mathbb{R}^{n \times d}$, where $n = 5$ and $d = 16$. Each "ResTimeCSL" block performs:

$$\mathbf{h}_{\text{out}} = \text{TCN}(\text{Affine}(\mathbf{h}_{\text{in}}) + \text{SkipConnections}),$$

with $\mathbf{h}_{\text{in}}$ and $\mathbf{h}_{\text{out}}$ denoting the input and output of a block, respectively. Similarly, the decoder uses multiple "ResTimeCSL" blocks to reconstruct the sequence, producing an output $\mathbf{y} \in \mathbb{R}^{T \times n}$, where $n = 5$. Finally, an additive layer combines the $n$ components at each time step $t$:

$$\mathbf{y}_{\text{final}}(t) = \sum_{i=1}^{n} \mathbf{y}_i(t),$$

ensuring that the final output size matches the input: $\mathbf{y}_{\text{final}} \in \mathbb{R}^{C \times T}$, with $C = 1$. This hierarchical structure, powered by multiple "ResTimeCSL" units, ensures effective representation learning and reconstruction while maintaining temporal and feature dimensions.

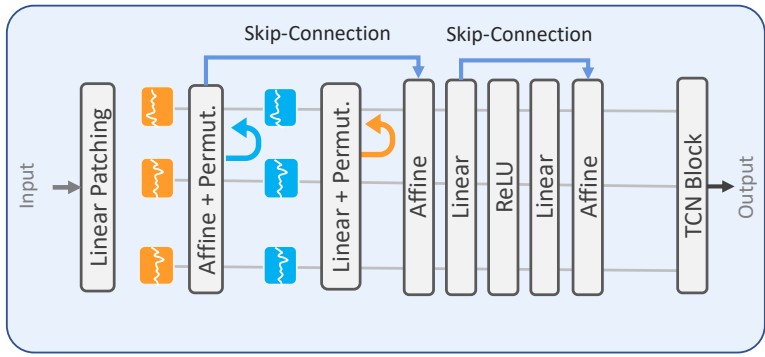

Figure 8: The residual unit ResTimeCSL, is employed in both the encoder and decoder.

The training process uses the Adamax (Kingma & Ba, 2014) optimizer with an initial learning rate of $10^{-3}$ and $\beta_1 = 0.9, \beta_2 = 0.999$. A cosine annealing learning rate decay is applied to improve convergence

### B.6 PIPELINE CORRELATED SAMPLES.

Robustness of the model to correlations between data is assessed by examining different pairs. We focus mainly on linear correlations between two different devices and on the case where one device correlates with two others. To do this, we parameterize the correlations by sampling a dataset from the common distribution. We build on the correlation time series framework by introducing a pairwise correlation between the attributes $y_m$ and $y_n$ as follows: $p(y_m, y_n) \propto \exp\left(-||y_m - \alpha y_n||^2 / 2\sigma^2\right)$, where $\alpha$ is a scaling factor. A high value of $\sigma$ indicates a lower correlation between the normalised attributes $y_m$ and $y_n$ (No.Corr, $\sigma = \infty$). We also extend this framework to cover correlations between several attributes in the time window $T$. Therefore, we consider correlation pair scenarios such as : *No correlation*; *Pair:1* washer-dryer; *Pair:2* dryer-oven and, finally, a *Random pair:* approach with randomly selected appliances.

### B.7 IMPACT OF RELU/LEAKYRELU AND ATTENTION LAYER WITH GELU ACTIVATION ON DECODER BEHAVIOR

In this study, we evaluate the impact of different activation functions on the decoder's behavior to satifies Asm 2.1. Specifically, we compare the use of ReLU (a piecewise affine activation) and GELU (a smooth, nonlinear activation) within an MLP decoder. The results suggest that the choice of activation function has a significant impact on the latent representation produced by the model.

**ReLU Activation:** The decoder becomes piecewise affine, meaning that it can be broken down into affine transformations over different regions of the input space. This causes the decoder to create latent representations that reflect distinct linear transformations in various regions of the input. As a result, the learned latent space is structured around these distinct affine regions, potentially making the model more sensitive to certain regions of the data space and leading to more discrete or sharply defined latent representations.

**LeakyReLU Activation:** In contrast, the GELU activation is smooth and nonlinear across the entire input space. This means that the decoder no longer operates piecewise affine, and the latent space learned by the model is more continuous and smooth. Since GELU smoothly transforms the input, it enables the decoder to create more nuanced, continuous latent representations. The absence of piecewise linear behavior allows for better modeling of complex, smooth relationships in the data, which may improve generalization to unseen data or tasks that require such smooth transformations.

### B.8   VALIDATION OF RESULTS ON SYNTHETIC DATA GENERATION

We simulate time-series data for energy disaggregation by leveraging the appliance signatures $y_k \in \mathbb{R}^T$ from the REDD and REFIT datasets, where $T$ is the number of time steps. The observed mixed signal $x \in \mathbb{R}^T$ is generated as the sum of the individual appliance contributions, i.e., $x_t = \sum_{k=1}^{n} y_{k,t} + \epsilon_t$ where $\epsilon_t \sim \mathcal{N}(0, \sigma^2)$ is Gaussian noise. Each appliance signature $y_k$ represents the time-series power consumption of appliance $k$, and these signatures are directly taken from the dataset. The final mixed signal $x$ is the result of combining the contributions from multiple appliances, with each $y_k$ corresponding to the power usage of a particular appliance in the dataset. This model serves as a foundation for evaluating energy disaggregation methods.

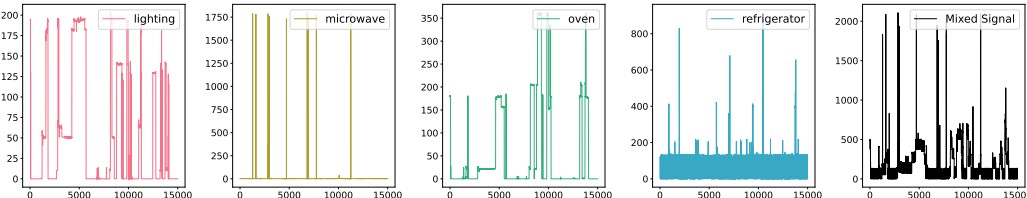

Figure 9: An example of a mixed signal from four sources in the REDD dataset.

### B.9   ADDITIONAL EXPERIMENT RESULTS.

#### B.9.1   EXPERIMENT ON REDD AND REFIT DATASETS

*Remark* B.1. In Tab. 6 , we observe a similarity in metrics across the REDD and REFIT datasets (with 5 seed experiments), despite their differences, can be explained by the fact that certain factors, particularly "FR", are highly represented in both datasets. This suggests that these common factors capture underlying patterns relevant to both datasets, leading to similar model performance. However, factors like "LT" and "HTR" are less prominent, which means their influence on the results is smaller. To address this and more accurately evaluate our approach in real datasets, we consider a broader set of factors such as {FR, DW, WM, HTR, LT}for REDD and UKDALE datasets, which would better capture the unique characteristics of each dataset and provide a more nuanced evaluation.

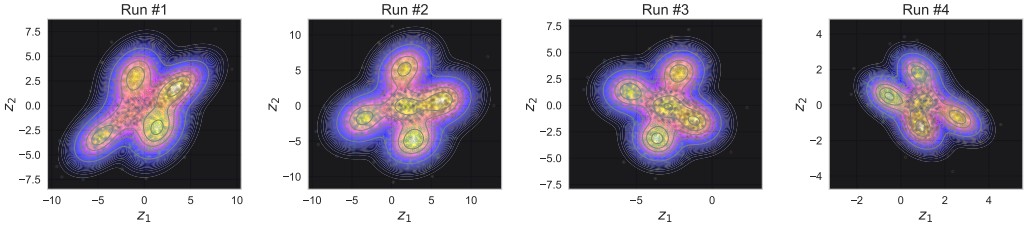

Figure 10: Recovered latent spaces for 4 runs of `TimeCSL` on REDD dataset with 5 latents ($n = 5, d = 16$) {FR, DW, WM, HTR, LT}.

Table 5: Average performance, considering factors {FR, DW, WM, HTR, LT} with 5 seed on real datasets REDD and REFIT. Metrics reported are DCI, RMIG and RMSE. Lower values are better for all metrics. (↓ lower is better, ↑ higher is worse Top-1, Top-2).

| Sc. | Methods | $\sigma = \infty$ | | | $\sigma = 0.3$ | | | $\sigma = 0.8$ | | |
|---|---|---|---|---|---|---|---|---|---|---|
| | Metrics ⇒ | DCI ↓ | RMIG ↓ | RMSE ↓ | DCI ↓ | RMIG ↓ | RMSE ↓ | DCI ↓ | RMIG ↓ | RMSE ↓ |
| Synthetic-1 | ○ BertNILM | - | - | 52.81 ± 25.41 | - | - | 75.78 ± 7.76 | - | - | 66.50 ± 6.69 |
| | ○ S2S | - | - | 47.99 ± 24.45 | - | - | 63.64 ± 20.56 | - | - | 67.93 ± 15.57 |
| | ○ Autoformer | - | - | 61.52 ± 7.66 | - | - | 52.23 ± 11.25 | - | - | 48.45 ± 9.31 |
| | ○ Informer | - | - | 48.59 ± 10.89 | - | - | 59.29 ± 11.36 | - | - | 63.45 ± 10.52 |
| | ● TimesNet | - | - | 63.57 ± 10.61 | - | - | 67.02 ± 9.10 | - | - | 69.93 ± 9.89 |
| | ○ C-DSVAE | 72.83 ± 11.71 | 1.08 ± 0.45 | 40.50 ± 6.45 | 71.76 ± 9.74 | 1.08 ± 0.44 | 51.67 ± 7.88 | 72.64 ± 10.89 | 1.23 ± 0.51 | 55.26 ± 7.80 |
| | ○ SlowVAE | 82.31 ± 11.96 | 1.08 ± 0.47 | 43.46 ± 7.93 | 81.65 ± 10.75 | 1.08 ± 0.46 | 54.81 ± 5.93 | 84.09 ± 6.93 | 1.27 ± 0.49 | 53.65 ± 7.48 |
| | ● CoST | 79.86 ± 10.86 | 1.16 ± 0.23 | 50.14 ± 6.77 | 79.16 ± 10.49 | 1.15 ± 0.22 | 55.91 ± 5.72 | 80.16 ± 9.68 | 1.25 ± 0.20 | 58.76 ± 5.51 |
| | ● SlowVAE+HDF | 88.69 ± 1.11 | 1.11 ± 0.24 | 65.87 ± 8.13 | 85.99 ± 1.34 | 0.97 ± 0.21 | 69.94 ± 7.29 | 89.47 ± 0.58 | 1.14 ± 0.24 | 72.21 ± 7.47 |
| | ● C-DSVAE + HDF | 76.94 ± 6.38 | 0.89 ± 0.37 | 33.61 ± 5.80 | 75.66 ± 6.53 | 0.84 ± 0.33 | 37.92 ± 5.88 | 74.45 ± 5.78 | 0.89 ± 0.40 | 42.58 ± 6.49 |
| | ● SparseVAE | 71.35 ± 8.48 | 0.67 ± 0.25 | 26.46 ± 5.68 | 72.67 ± 8.54 | 0.68 ± 0.27 | 31.07 ± 5.34 | 73.98 ± 8.23 | 0.74 ± 0.29 | 32.56 ± 5.16 |
| | ● TimeCSL | 75.44 ± 6.93 | 0.59 ± 0.17 | 25.53 ± 6.69 | 74.50 ± 6.29 | 0.61 ± 0.19 | 29.23 ± 6.57 | 76.66 ± 5.70 | 0.74 ± 0.16 | 33.76 ± 6.73 |
| Synthetic-2 | ○ BertNILM | - | - | 60.83 ± 5.80 | - | - | 72.63 ± 2.25 | - | - | 71.02 ± 2.55 |
| | ○ S2S | - | - | 53.73 ± 5.84 | - | - | 65.57 ± 5.35 | - | - | 69.21 ± 4.06 |
| | ○ Autoformer | - | - | 54.60 ± 1.70 | - | - | 50.48 ± 2.82 | - | - | 50.39 ± 2.26 |
| | ○ Informer | - | - | 45.92 ± 3.03 | - | - | 53.77 ± 2.86 | - | - | 61.08 ± 2.51 |
| | ● TimesNet | - | - | 54.68 ± 3.68 | - | - | 55.28 ± 3.02 | - | - | 59.24 ± 3.41 |
| | ○ C-DSVAE | 74.83 ± 5.72 | 1.12 ± 0.23 | 47.04 ± 3.14 | 73.42 ± 2.40 | 1.10 ± 0.21 | 53.02 ± 3.49 | 75.29 ± 3.34 | 1.21 ± 0.14 | 54.81 ± 3.46 |
| | ● SlowVAE | 80.92 ± 2.73 | 1.10 ± 0.20 | 44.58 ± 3.11 | 79.95 ± 2.64 | 1.09 ± 0.18 | 51.92 ± 2.58 | 81.45 ± 1.57 | 1.21 ± 0.14 | 50.69 ± 2.99 |
| | ● CoST | 71.18 ± 3.83 | 1.04 ± 0.06 | 47.10 ± 1.66 | 71.01 ± 3.86 | 1.05 ± 0.05 | 53.58 ± 1.39 | 70.56 ± 3.50 | 1.14 ± 0.04 | 55.29 ± 1.22 |
| | ● SlowVAE+HDF | 81.13 ± 0.17 | 0.85 ± 0.08 | 60.50 ± 3.01 | 80.21 ± 0.19 | 0.79 ± 0.07 | 62.72 ± 2.77 | 81.68 ± 0.10 | 0.89 ± 0.05 | 64.03 ± 2.99 |
| | ● C-DSVAE + HDF | 74.77 ± 1.56 | 0.78 ± 0.05 | 35.62 ± 2.52 | 74.39 ± 1.51 | 0.75 ± 0.05 | 38.40 ± 1.83 | 74.88 ± 0.98 | 0.79 ± 0.07 | 39.95 ± 1.62 |
| | ● SparseVAE | 69.84 ± 4.10 | 0.62 ± 0.06 | 27.28 ± 2.59 | 69.95 ± 4.15 | 0.60 ± 0.05 | 29.61 ± 1.67 | 72.52 ± 3.77 | 0.65 ± 0.07 | 30.35 ± 1.45 |
| | ● TimeCSL | 71.72 ± 3.23 | 0.46 ± 0.04 | 25.02 ± 2.77 | 71.21 ± 2.58 | 0.51 ± 0.03 | 25.91 ± 2.62 | 72.68 ± 2.33 | 0.61 ± 0.02 | 28.82 ± 2.83 |

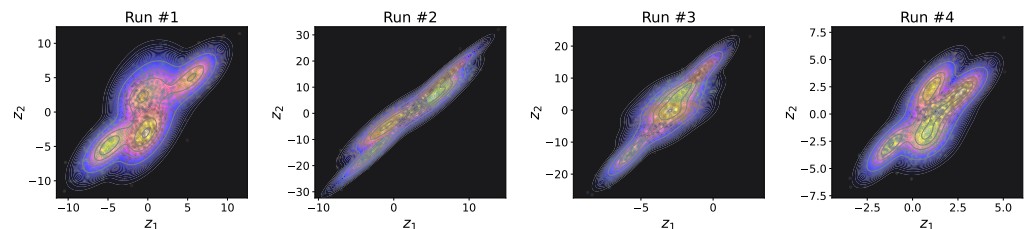

Figure 11: Recovered latent spaces for 4 runs of TDRL on REDD dataset with 5 latents ($n = 5, d = 16$) {FR, DW, WM, HTR, LT}.

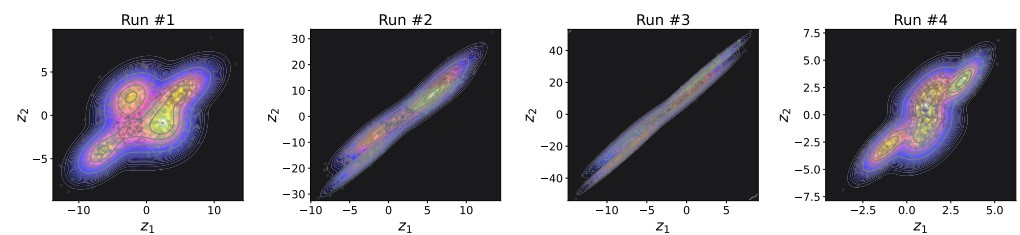

Figure 12: Recovered latent spaces for 4 runs of SlowVAE on REDD dataset with 5 latents ($n = 5, d = 16$) {FR, DW, WM, HTR, LT}.

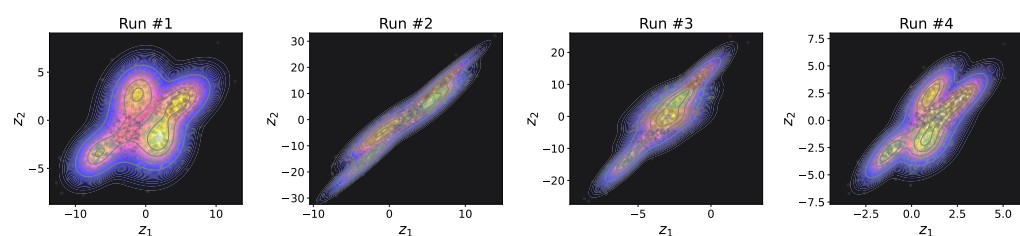

Figure 13: Recovered latent spaces for 4 runs of iVAE on REDD dataset with 5 latents ($n = 5, d = 16$) {FR, DW, WM, HTR, LT}.

### B.9.2 EXPERIMENT ON SYNTHETIC DATASETS

Table 6: Average performance, considering factors {FR, LT, HTR} with 5 seed on synthetics datasets (1 & 2). Metrics reported are: DCI, RMIG and RMSE. Lower values are better for all metrics. (↓ lower is better, ↑ higher is worse  Top-1 ,  Top-2 ).

| Sc. | Methods | $\sigma = \infty$ | | | $\sigma = 0.3$ | | | $\sigma = 0.8$ | | |
|---|---|---|---|---|---|---|---|---|---|---|
| | Metrics ⇒ | DCI ↓ | RMIG ↓ | RMSE ↓ | DCI ↓ | RMIG ↓ | RMSE ↓ | DCI ↓ | RMIG ↓ | RMSE ↓ |
| Synthetic-1 | BertNILM | - | - | 36.86 ± 1.68 | - | - | 45.84 ± 1.00 | - | - | 46.29 ± 0.76 |
| | S2S | - | - | 35.46 ± 2.04 | - | - | 45.36 ± 2.47 | - | - | 45.76 ± 2.26 |
| | Autoformer | - | - | 32.45 ± 0.56 | - | - | 33.02 ± 1.49 | - | - | 34.68 ± 1.13 |
| | Informer | - | - | 32.92 ± 1.67 | - | - | 35.03 ± 1.71 | - | - | 38.47 ± 1.54 |
| | TimesNet | - | - | 32.12 ± 1.99 | - | - | 33.38 ± 1.83 | - | - | 35.84 ± 1.61 |
| | CoST | 44.68 ± 1.57 | 0.61 ± 0.02 | 31.14 ± 0.93 | 48.01 ± 1.57 | 0.64 ± 0.09 | 34.81 ± 0.71 | 46.98 ± 1.13 | 0.65 ± 0.01 | 38.14 ± 0.57 |
| | SlowVAE | 50.96 ± 0.71 | 0.61 ± 0.09 | 28.26 ± 1.54 | 53.04 ± 1.26 | 0.61 ± 0.09 | 32.15 ± 0.78 | 52.14 ± 0.58 | 0.70 ± 0.08 | 35.74 ± 1.03 |
| | SlowVAE+HDF | 52.17 ± 0.07 | 0.42 ± 0.02 | 37.35 ± 1.49 | 53.00 ± 0.12 | 0.46 ± 0.05 | 38.86 ± 1.26 | 52.53 ± 0.03 | 0.47 ± 0.01 | 40.22 ± 1.06 |
| | TDRL | 42.34 ± 1.02 | 0.28 ± 0.04 | 18.64 ± 1.41 | 49.75 ± 0.87 | 0.31 ± 0.01 | 17.18 ± 1.36 | 50.43 ± 0.69 | 0.38 ± 0.08 | 20.91 ± 1.07 |
| | D3VAE | 41.30 ± 1.97 | 0.26 ± 0.05 | 27.64 ± 1.40 | 41.55 ± 0.91 | 0.33 ± 0.26 | 30.11 ± 1.10 | 43.47 ± 1.31 | 0.44 ± 0.03 | 32.77 ± 0.51 |
| | C-DSVAE | 47.35 ± 2.14 | 0.59 ± 0.05 | 31.78 ± 1.61 | 47.79 ± 0.99 | 0.62 ± 0.26 | 34.55 ± 1.18 | 50.02 ± 1.42 | 0.71 ± 0.03 | 37.57 ± 0.53 |
| | C-DSVAE + HDF | 44.31 ± 1.93 | 0.56 ± 0.05 | 29.68 ± 1.51 | 45.01 ± 0.92 | 0.59 ± 0.25 | 32.42 ± 1.04 | 46.68 ± 1.33 | 0.66 ± 0.03 | 35.12 ± 0.50 |
| | SparseVAE | 40.15 ± 0.86 | 0.25 ± 0.09 | 13.72 ± 1.30 | 43.98 ± 0.81 | 0.28 ± 0.21 | 14.81 ± 1.20 | 44.53 ± 0.58 | 0.31 ± 0.07 | 18.89 ± 1.30 |
| | TimeCSL | 39.02 ± 0.87 | 0.23 ± 0.07 | 12.03 ± 1.26 | 42.51 ± 0.74 | 0.27 ± 0.15 | 12.72 ± 1.16 | 42.91 ± 0.59 | 0.31 ± 0.05 | 14.76 ± 0.92 |
| | **Avg.** | 45.62 ± 1.27 | 0.52 ± 0.07 | 31.02 ± 1.26 | 48.02 ± 0.85 | 0.58 ± 0.12 | 34.08 ± 1.04 | 48.92 ± 1.18 | 0.64 ± 0.06 | 35.67 ± 0.91 |
| Synthetic-2 | BertNILM | - | - | 40.06 ± 2.41 | - | - | 44.14 ± 1.22 | - | - | 45.04 ± 0.99 |
| | S2S | - | - | 38.48 ± 2.87 | - | - | 45.07 ± 2.71 | - | - | 46.22 ± 2.26 |
| | Autoformer | - | - | 33.56 ± 0.79 | - | - | 34.13 ± 2.07 | - | - | 37.51 ± 1.81 |
| | Informer | - | - | 36.02 ± 2.37 | - | - | 37.61 ± 1.98 | - | - | 38.81 ± 2.36 |
| | TimesNet | - | - | 36.69 ± 2.08 | - | - | 39.08 ± 2.71 | - | - | 42.55 ± 2.35 |
| | CoST | 50.87 ± 1.13 | 0.58 ± 0.06 | 28.93 ± 1.81 | 53.10 ± 1.23 | 0.61 ± 0.14 | 30.72 ± 1.31 | 52.63 ± 1.19 | 0.67 ± 0.14 | 33.15 ± 1.12 |
| | SlowVAE | 48.11 ± 1.06 | 0.45 ± 0.05 | 31.73 ± 2.19 | 50.15 ± 1.35 | 0.47 ± 0.06 | 34.12 ± 1.57 | 50.97 ± 0.78 | 0.55 ± 0.02 | 35.27 ± 1.06 |
| | SlowVAE + HDF | 51.09 ± 1.64 | 0.34 ± 0.04 | 32.85 ± 2.40 | 51.97 ± 1.07 | 0.39 ± 0.05 | 35.72 ± 2.17 | 51.85 ± 1.58 | 0.43 ± 0.06 | 37.38 ± 2.51 |
| | TDRL | 45.12 ± 2.15 | 0.39 ± 0.05 | 22.87 ± 1.36 | 50.61 ± 1.53 | 0.44 ± 0.03 | 23.98 ± 1.41 | 51.18 ± 0.90 | 0.49 ± 0.08 | 27.13 ± 2.30 |
| | D3VAE | 43.77 ± 1.31 | 0.36 ± 0.06 | 28.43 ± 1.61 | 48.42 ± 1.89 | 0.39 ± 0.04 | 30.14 ± 1.35 | 48.02 ± 1.23 | 0.44 ± 0.06 | 32.46 ± 1.10 |
| | C-DSVAE | 49.68 ± 2.12 | 0.55 ± 0.07 | 31.03 ± 2.15 | 49.92 ± 1.05 | 0.58 ± 0.08 | 33.60 ± 1.77 | 51.51 ± 1.76 | 0.61 ± 0.03 | 35.38 ± 1.42 |
| | C-DSVAE + HDF | 47.38 ± 1.19 | 0.53 ± 0.05 | 30.76 ± 2.13 | 48.85 ± 1.62 | 0.56 ± 0.03 | 32.89 ± 2.04 | 49.98 ± 1.34 | 0.60 ± 0.05 | 34.25 ± 1.22 |
| | SparseVAE | 46.56 ± 2.49 | 0.44 ± 0.08 | 19.88 ± 2.06 | 50.49 ± 1.07 | 0.47 ± 0.06 | 21.42 ± 2.53 | 50.83 ± 1.73 | 0.53 ± 0.05 | 23.59 ± 2.17 |
| | TimeCSL | 43.45 ± 1.12 | 0.33 ± 0.02 | 16.32 ± 2.16 | 47.33 ± 1.29 | 0.35 ± 0.04 | 17.22 ± 2.01 | 48.09 ± 0.81 | 0.39 ± 0.06 | 18.95 ± 2.08 |
| | **Avg.** | 47.02 ± 1.56 | 0.45 ± 0.06 | 28.04 ± 1.84 | 50.43 ± 1.19 | 0.48 ± 0.09 | 30.32 ± 1.56 | 50.95 ± 1.26 | 0.54 ± 0.07 | 32.83 ± 1.57 |

Table 7: Average performance, considering factors {FR, DW, WM, HTR, LT} with 5 seed on synthetics datasets. Metrics reported are DCI, RMIG and RMSE. Lower values are better for all metrics. (↓ lower is better, ↑ higher is worse  Top-1 ,  Top-2 ).

| Sc. | Methods | $\sigma = \infty$ | | | $\sigma = 0.3$ | | | $\sigma = 0.8$ | | |
|---|---|---|---|---|---|---|---|---|---|---|
| | Metrics ⇒ | DCI ↓ | RMIG ↓ | RMSE ↓ | DCI ↓ | RMIG ↓ | RMSE ↓ | DCI ↓ | RMIG ↓ | RMSE ↓ |
| Synthetic-3 | BertNILM | - | - | 56.4 ± 2.58 | - | - | 70.2 ± 1.45 | - | - | 70.92 ± 1.15 |
| | S2S | - | - | 54.3 ± 3.12 | - | - | 69.5 ± 3.56 | - | - | 69.95 ± 3.26 |
| | Autoformer | - | - | 49.7 ± 0.81 | - | - | 50.5 ± 2.15 | - | - | 52.95 ± 1.63 |
| | Informer | - | - | 50.3 ± 2.41 | - | - | 53.5 ± 1.98 | - | - | 58.95 ± 1.89 |
| | FEDformer | - | - | 50.3 ± 2.12 | - | - | 52.5 ± 2.45 | - | - | 59.01 ± 1.76 |
| | TimesNet | - | - | 49.24 ± 2.87 | - | - | 51.10 ± 2.64 | - | - | 54.91 ± 2.31 |
| | C-DSVAE | 72.42 ± 3.10 | 0.96 ± .15 | 48.6 ± 2.32 | 73.12 ± 1.43 | 0.95 ± .15 | 52.9 ± 2.31 | 74.29 ± 2.04 | 1.08 ± .09 | 52.99 ± 1.91 |
| | SlowVAE | 78.0 ± 1.09 | 0.94 ± .13 | 43.2 ± 2.23 | 78.0 ± 1.09 | 0.94 ± .13 | 49.2 ± 1.13 | 79.74 ± 0.84 | 1.07 ± .11 | 49.65 ± 1.43 |
| | CoST | 68.4 ± 2.41 | 0.97 ± .03 | 47.7 ± 1.35 | 68.4 ± 2.41 | 0.97 ± .03 | 53.2 ± 1.02 | 69.95 ± 1.63 | 1.00 ± .02 | 53.45 ± 0.82 |
| | SlowVAE+HDF | 79.8 ± .10 | 0.64 ± .05 | 57.2 ± 2.15 | 79.8 ± .10 | 0.64 ± .05 | 61.3 ± 1.82 | 80.37 ± .05 | 0.72 ± .03 | 61.64 ± 1.52 |
| | C-DSVAE + HDF | 73.1 ± 1.01 | 0.69 ± .02 | 34.4 ± 1.89 | 73.1 ± 1.01 | 0.69 ± .02 | 38.1 ± 1.34 | 74.25 ± 0.59 | 0.73 ± .05 | 38.48 ± 1.04 |
| | SparseVAE | 67.2 ± 2.01 | 0.52 ± .02 | 24.3 ± 1.81 | 67.2 ± 2.01 | 0.52 ± .02 | 27.4 ± 1.13 | 71.79 ± 1.27 | 0.58 ± .04 | 27.77 ± 0.83 |
| | TimeCSL | 63.5 ± 1.35 | 0.38 ± .02 | 19.6 ± 1.95 | 69.3 ± 1.2 | 0.44 ± .02 | 20.3 ± 1.79 | 70.12 ± 0.91 | 0.51 ± .01 | 23.63 ± 1.49 |
| Synthetic-4 | BertNILM | - | - | 61.42 ± 3.47 | - | - | 67.61 ± 1.95 | - | - | 69.06 ± 1.43 |
| | S2S | - | - | 59.08 ± 4.15 | - | - | 68.60 ± 3.91 | - | - | 70.68 ± 3.25 |
| | Autoformer | - | - | 49.87 ± 0.92 | - | - | 51.53 ± 1.48 | - | - | 51.88 ± 1.34 |
| | Informer | - | - | 54.23 ± 1.78 | - | - | 57.70 ± 1.78 | - | - | 62.51 ± 1.55 |
| | FEDformer | - | - | 52.84 ± 1.69 | - | - | 55.83 ± 1.82 | - | - | 61.92 ± 1.57 |
| | TimesNet | - | - | 51.37 ± 2.41 | - | - | 55.35 ± 2.23 | - | - | 58.47 ± 2.21 |
| | C-DSVAE | 72.97 ± 3.44 | 1.04 ± 0.16 | 47.17 ± 2.11 | 73.60 ± 1.82 | 0.98 ± 0.14 | 52.16 ± 1.89 | 73.96 ± 2.46 | 1.11 ± 0.12 | 53.73 ± 1.79 |
| | SlowVAE | 77.41 ± 1.67 | 0.94 ± 0.15 | 46.61 ± 1.91 | 77.80 ± 1.63 | 0.95 ± 0.14 | 49.82 ± 1.71 | 79.47 ± 1.26 | 1.04 ± 0.13 | 50.88 ± 1.58 |
| | CoST | 70.75 ± 2.01 | 0.96 ± 0.09 | 48.92 ± 1.62 | 70.87 ± 2.04 | 0.96 ± 0.09 | 52.73 ± 1.34 | 71.93 ± 1.84 | 0.98 ± 0.09 | 54.46 ± 1.19 |
| | SlowVAE+HDF | 79.97 ± 0.14 | 0.72 ± 0.05 | 56.96 ± 2.34 | 79.77 ± 0.14 | 0.72 ± 0.05 | 59.75 ± 2.21 | 80.22 ± 0.07 | 0.75 ± 0.03 | 60.77 ± 2.22 |
| | C-DSVAE + HDF | 73.85 ± 0.85 | 0.69 ± 0.05 | 34.19 ± 1.47 | 73.71 ± 0.85 | 0.69 ± 0.05 | 37.53 ± 1.21 | 74.34 ± 0.56 | 0.71 ± 0.04 | 39.35 ± 1.06 |
| | TDRL | 70.86 ± 0.816 | 0.57 ± 0.041 | 32.80 ± 1.41 | 70.75 ± 0.816 | 0.57 ± 0.041 | 36.04 ± 1.16 | 71.94 ± 0.54 | 0.58 ± 0.033 | 37.83 ± 1.02 |
| | SparseVAE | 70.13 ± 1.44 | 0.61 ± 0.04 | 25.46 ± 1.10 | 70.13 ± 1.44 | 0.61 ± 0.04 | 28.99 ± 1.22 | 71.44 ± 1.30 | 0.63 ± 0.05 | 29.47 ± 1.10 |
| | TimeCSL | 66.14 ± 1.66 | 0.40 ± 0.04 | 19.81 ± 1.29 | 69.00 ± 1.41 | 0.44 ± 0.04 | 20.46 ± 1.45 | 70.41 ± 1.22 | 0.48 ± 0.03 | 22.08 ± 1.36 |

### B.9.3 COMPARISONS BETWEEN TIMECSL AND BASELINES ON KITTI DATASET

We evaluate TimeCSL on time-sequential data using preprocessed frames from the KITTI and MOTSChallenge datasets. The original KITTI image resolutions are $1080 \times 1920$ or $480 \times 640$ for MOTSChallenge, and between 370–374 pixels tall by 1224–1242 pixels wide for KITTI MOTS. The video frame rates vary from 14 to 30 fps, as described in (Milan, 2016). To preprocess the data, we apply nearest-neighbor down-sampling to reduce each frame's height to 64 pixels while maintaining the aspect ratio for the width. Using a horizontal sliding window, we extract six equally spaced windows of size $64 \times 64$ (with overlap) from each sequence in both datasets. This preprocessing produces a sequence of shape $64 \times 64 \times T$, where $T$ represents the number of time steps in the sequence. Our approach assumes reasonable invariance to horizontal translation and scale within

the dataset. Scale invariance is supported by the fact that the data was collected from a car-mounted camera, leading to varying distances to pedestrians. To validate translation invariance, we conducted an ablation study on the number of horizontal sliding windows. Using only two horizontally spaced windows, instead of six, resulted in no significant changes in key statistics, such as kurtosis (remaining within $\pm 10\%$ of the original value for $\Delta x$ transitions). This experiment results Fig. 14 demonstrates the robustness of TimeCSL to time-sequential data, showcasing its potential for applications beyond its original domain.

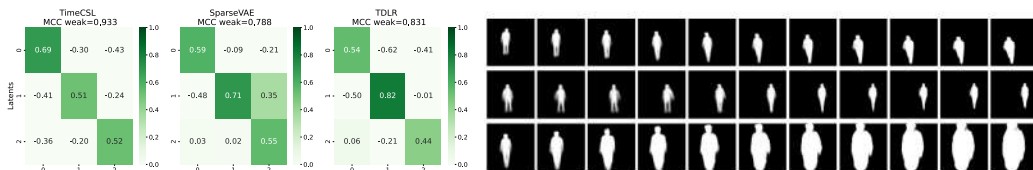

Figure 14: **Validation on KITTI dataset. Left.** MCC correlation matrix of the top 3 latents corresponding to y-position (1), x-position (2) and scale (3). **Right.** Images produced by varying the `TimeCSL` latent unit that corresponds to the corresponding row in the MCC matrix.

