# OpenReview forum: "Identifiability Guarantees For Time Series Representation via Contrastive Sparsity-inducing"
_ICLR.cc/2025/Conference — ICLR 2025 Conference Withdrawn Submission_

### Official Review · Reviewer_DZWn · 2024-11-03

**Soundness:** 1
**Presentation:** 2
**Contribution:** 2
**Rating:** 3
**Confidence:** 3

**Summary:**

This paper proposes the identification guarantees of learning representation for time series data. Specifically, this paper proposed a new method, called Contrastive Sparsity-inducing, to leverage the assumption of sparsity in data structure. Then, a TimeCSL framework is proposed to learn a disentangled representation with the constraint of sparsity. Extensive experiments are conducted to evaluate the proposed method.

**Strengths:**

1) The codes are released for better reproduction.

2) The experiments on public datasets and synthetic datasets are conducted.

3) Figure 1 proposes a good point to show the motivations.

**Weaknesses:**

I have three big concerns about this paper.

First, though some definitions and assumptions were listed, I didn't find the strict theorem and corresponding proof about the identification results. Some related work is referred to, such as  (Lachapelle et al., 2022). However, it is still not clear how this method can directly help prove the identification.

Second, lots of existing ICA-based work on identifiable disentangled time series representation is missing to compare in evaluation, like TCL, PCL, TDRL, and so on.   Discuss with them can further help highlight the contribution. Besides, it is better to show the difference between the key assumption 4.1 and assumption 6 in Sparse ICA (Ignavier Ng, 2023, Neurips).

Third, the results seem not reliable. The performance of S3VAE+HFS, C-DSVAE + HFS, SparseVAE, and TimeCSL are totally the same on two different datasets. The authenticity of experimental data is doubtful.  Due to time limitations, the code is not checked. Will check it before the rebuttal phase.

**Questions:**

Please refer to the weaknesses section.

---

> ### Author Response · Authors · 2024-11-28
>
> Dear Reviewer DZWn
>
> Thank you for your thoughtful and thorough review of our work. We greatly appreciate the time and effort you devoted to reviewing our paper and highlighting its strengths. In response to your feedback, we have carefully revised the manuscript. Below, we provide detailed responses to your comments, and we have released an updated version of the paper incorporating your valuable suggestions.
>
> 🔹  **Identifiability Results**
> Thank you again for pointing this out. The proof was originally included in the appendix, but due to an oversight, the correct version was not uploaded to OpenReview. We have now formally stated the results in **Theorem 4.2**, with proof sketches in the main text and complete proofs in Appendix A.3.
>
> 🔹  **Additional Time Series Comparisons Provided**
> Thank you for your comment. In the initial version, we compared our work to SlowVAE, which is somewhat similar to TDRL results in our experiments. In the revised paper, we have also included comparisons to TDRL, IVAE, LEAP, and TCL. This strengthens the contribution of our paper by providing a more comprehensive comparison.
>
> ☑  TDRL
> ☑  iVAE,
> ☑   LEAP
> ☑  TCL
>
> 🔹 **Assumption 4.1 and Structural Sparsity**
> We appreciate your suggestion to discuss the differences between our assumption and assumption-6 presented by Hygiene [1]. In our original version, we focused more on the assumption presented by Zheng [2] (joint work with Ignavier Ng), which is similar to the work by Lachappelle [3] (**Structural Variability**). In  the revised version of our work, we have included a more detailed discussion in **lines 267-269**. The key differences are as follows:
>
> 1. **Structural Sparsity** (Assumption-6 of [2]), ensures that each pair of sources influences distinct observed variables.
> 2. However, in real-world time series, overlapping influences often occur, presenting practical challenges (as discussed in App. A.5). Our **Partial Selective Pairing assumption** (Eq. 4.1) allows some overlap, provided that the union of shared support indices (excluding the specific source) spans all sources. This enables more flexible modeling of source dependencies under contrastive learning.
> 3- We provide an example in the Appendix A.5 to validate our assumption 4.1, as well as assumption 6 in [1].
>
>
> 🔹 **Experiments in Table 1.1**
>
> We have added additional experimental results and clarified this in **Remark C.1**. Initially, when considering only 3 factors (**{FR, THR, HTR}**, where FR = Fridge, DW = Dishwasher, WM = Washing Machine, HTR = Heater, LT = Lighting), the results for S3VAE+HFS, C-DSVAE+HFS, SparseVAE, and TimeCSL were quite similar due to common signal combinations. However, when more factors (**{FR, DW, WM, HTR, LT}**) were included in training and testing, we observed distinct differences. These results are discussed in detail in **Table 5 of Appendix B.9.1**.
>
> We also want to emphasize that we have provided checkpoints and code for reproducibility. To further support clarity and reproducibility, we’ve included an example.
>
> ---
> 🔹 Checkpoints & Code
> ---
> - Please there is no ``main.py``, we use ``train.py``, and ``src_timecsl/evaluation.py`` for evaluation
> - The architecture of the network is depicted in Figure 3 in our paper, with the implementation available in ``/src_timecsl/models/timecsl.py``.
> - Run command:
> ```bash
> cd src_timecsl/
> python train.py --dataset_path "./datasets/data/ukdale.csv" --model_name "TimeCSL" --num_slots 5 --epochs 200 --use_generalization_loss True
> ```
>
> ---
> \
> **Thank you once again for your insightful feedback. We believe these revisions strengthen the paper, and we look forward to your further thoughts. We would appreciate it if you could consider adjusting the rating accordingly.**
>
>
> Best regards,  \
> Authors,
>
>
> **References:**
>
> [1] Ng, I., et al. On the identifiability of sparse ICA without assuming non-Gaussianity. NeurIPS 2023
>
> [2] Zheng, Y., et al. On the identifiability of nonlinear ICA: Sparsity and beyond. NeurIPS 2022
>
> [3] Lachapelle, et al. Nonparametric partial disentanglement via mechanism sparsity: Sparse actions, interventions, and sparse temporal dependencies.

---

### Official Review · Reviewer_JQPq · 2024-11-03

**Soundness:** 2
**Presentation:** 2
**Contribution:** 3
**Rating:** 5
**Confidence:** 2

**Summary:**

The work discusses identifiability that guarantees disentangled representations via Contrastive Sparsity-inducing. Following this, a new framework called TimeCSL is proposed to learn a generalised disentangled representation that maintains compositionality. The results show the efficacy of the proposed method compared to various baselines.

**Strengths:**

1. The paper is easy to follow.
2. The evaluation in Table 1 is very comprehensive with many cases and baselines considered.
3. All pre-trained models are accessible along with guidelines.

**Weaknesses:**

1. There are a few typos and broken references in the paper for e.g. guarantee (misspelled in Line 77), broken reference in line 409, figure reference broken in line 502. I will advise doing a spell check etc.
2. Could the authors discuss using Normalising Flows (or Diffusion Models) instead of a VAE in their framework? Flows, for example, are bijective transformations. Some results for identifiability with Flows exist for image data [1]; maybe they can be directly applied to temporal data.
3. Could the authors provide justification or empirical evidence for the realism of Assumption 4.1 in practical time series scenarios? It will be interesting to see how sensitive the method's performance is to violations of this assumption.

[1] DISENTANGLEMENT BY NONLINEAR ICA WITH GENERAL INCOMPRESSIBLE-FLOW NETWORKS (GIN) by Peter Sorrenson, Carsten Rother, Ullrich Kothe

**Questions:**

See weaknesses

---

> ### Author Response · Authors · 2024-11-29
> **Include Normalizing Flows, Diffusion VAE and Practical Time Series Scenarios**
>
> Dear Reviewer JQPq,
>
> Thank you for your thoughtful and thorough review of our work. We greatly appreciate the time and effort you put into reviewing our paper and highlighting its strengths. In response to your feedback, we have carefully revised the manuscript. Below, we provide detailed responses to your comments, and we have released an updated version of the paper incorporating your valuable suggestions.
>
> ### Addressing weaknesses and Questions:
>
> 1. 🔹 **Misspelling in Line 77**
>    We have thoroughly reviewed the manuscript and corrected the spelling error you pointed out.
>
> 2. 🔹 **Extending Normalizing Flows or Diffusion Models**
>    Thank you for your insightful suggestion. Normalizing Flows (NF) and Diffusion Models indeed offer powerful, flexible transformations, and could potentially enhance our framework. We have discussed the use of Diffusion Models in combination with VAE (D3VAE) [1] in our experiments. We are actively exploring this direction and plan to include [2] as additional experiments in a future version of the paper.
>
> 3. 🔹 **Justification for Assumption 4.1 in Practical Time Series Scenarios**
>
>    Thank you for raising this point. Assumption 4.1 is crucial to our approach, and we understand the need for empirical justification regarding its realism in real-world time series. The assumption suggests that the influence of sources on observed variables is partial and selective, i.e., some sources are more influential at different times. In comparison to other work [3,4,5,6], such as **Structural Sparsity** [5] or **Sparse Variability** [6], our assumption is more relaxed but still effective.
>
> We conducted experiments with both synthetic (Tables 7-8 in Appendix B.9.2) and real-world data, demonstrating that our model performs well even when **Assumption 4.1** is not perfectly satisfied. However, we acknowledge that deviations from this assumption can impact performance. To address this:
>
>    - We provide further discussion in Section 4.1 and Appendix A.5 on the realism of **Assumption 4.1**.
>    - We propose grouping sources, i.e., considering groups of sources as active at the same time, as a relaxation of this assumption. This approach still allows us to achieve effective disentanglement in real-world applications.
>
> **Thank you once again for your insightful feedback. We believe these revisions strengthen the paper, and we look forward to your further thoughts. We would appreciate it if you could consider adjusting the rating accordingly.**
>
> Best regards,  \
> Authors,
>
>
> **References:**
> - [1] Li, Y., et. al. Generative time series forecasting with diffusion, denoise, and disentanglement. NeurIPS 2022
> - [2] Sorrenson, et al. Disentanglement by nonlinear ica with general incompressible-flow networks (gin), ICLR 2020
> - [3] Zheng, Y., et al. On the identifiability of nonlinear ICA: Sparsity and beyond. NeurIPS 2022
> - [4] Lachapelle, et al. Nonparametric partial disentanglement via mechanism sparsity: Sparse actions, interventions, and sparse temporal dependencies.
> - [5] Ng, I., et al. On the identifiability of sparse ICA without assuming non-Gaussianity. NeurIPS 2023
> - [6] Zheng, Y., et al. On the identifiability of nonlinear ICA: Sparsity and beyond. NeurIPS 2022

---

### Official Review · Reviewer_agV7 · 2024-11-03

**Soundness:** 1
**Presentation:** 1
**Contribution:** 1
**Rating:** 1
**Confidence:** 5

**Summary:**

The paper "IDENTIFIABILITY GUARANTEES IN TIME SERIES REPRESENTATION VIA CONTRASTIVE SPARSITY-INDUCING" proposes Contrastive Sparsity-Inducing Learning to help improve model generalization and interpretability. A slot-wise identifiability has been proved in the theorem part. Additionally, it implements the TimeCSL framework, which enhances performance across various existing models. The authors provided lots of experiments to support their conclusions.

**Strengths:**

- Question is interesting.
- The motivation is clear.
- The figure is intuitive.

**Weaknesses:**

- Section 2: The setting is not clear.
    - The author claims that $x$ is an additive mixture of y from $y _ 1$ to $y _ n$ and noise. However, in the next paragraph, the dataset is defined as $D= ${$ x _ i, y _ i $}$  _ {i=1} ^ N$, which indicates that $y$ is the label. This is confusing.
    - If in the first paragraph, y is the latent variable. It is also confusing about the mixing function. What does $n$ means in line 119. Does it mean that at each time step, y is a $n$ dimensional vector?
    - If so, in line 117, when $C=1$, does it infer that the mixing function is not injective?
- Equation 2.1: The reconstructed in reconstrcution loss should be $x$ rather than $y$.
- Line 144: What is gθ♯p(z)? What is $M^1 _ + (X)$? Definition is needed.
- Line 162: What is 'recouvering'? Is it 'recovering'?
- Line 162: y is not given in Eq 2.2 (generating process) and emerge here sharply, it is not clear what this sentence mean here.
- Definition 2.2: Usually we call it identifiability rather than disentanglement.
- Line 257: unfinished setence, if what?
- Section 4.1: Why call entries i with $\frac{|\mu _ i|}{\sigma _ i}>1$ importatant? Some discussion is needed.
- Line 262: Usually the term component-wise identifiability is used.
- Assumption 4.1: assumption is not aligned with Equation 4.1. Text says two samples x and x', while Equation says all $i$ that does not include k, which infers much more observables.
- Experiments are not reliable
    - Table 1: The performances of S3VAE+HFS, C-DSVAE+HFS, SparseVAE, TimeCSL are EXACTLY the same, for REFIT and REDD. As far as i know, the two dataset is not that same.
    - Table 1: why is R2 larger than 1 (in the table it shows "RMIG", I am not sure what it is)
    - line 404: split 70/20/20, which sums up to more than 100
    - Table 2: why "higher is worse" for MCC？Why the larger R2 is, the smaller MCC is, which is strange.

**Questions:**

I am interested in the detailed implementation of the model. At the same time, the code seems not runnable. It will be helpful if a more detailed document can be added to the code.
- Code is not runnable.
    - There is no file named 'main.py'.
    - The code looks like for image data rather than sequence data.

---

> ### Author Response · Authors · 2024-11-28
> **Part1 - Clarification of the Setting and nonlineair ICA**
>
> Dear reviewer agV7,
>
> Thank you for your initial review of our work. We greatly appreciate the time and effort you put into reviewing our paper and highlighting its strengths. In response to your feedback, we have carefully revised and updated the manuscript. Below, we provide detailed answers to your comments, and we have also released an updated version of the paper incorporating your valuable recommendations.
>
> We hope that these revisions address your concerns and enhance the clarity of our work.
>
> ### Section 2 - **Clarification of the Setting:**
> - First, this problem separation is kindly studied in the context of nonlinear ICA [1]. We consider a mixing signal $\mathbf{x} \in \mathbb{R}^{C \times T}$ with $C = 1$ (one feature) and $T$ time steps. The signal $\mathbf{x}$ is the sum of $n$ sources $\mathbf{y} = \{y\_1, \dots, y\_n\}$, where each $y\_k \in \mathbb{R}^{T}$ contributes to $\mathbf{x}$ at each time step.
>
> - The index $i$ in Eq. (2.2) refers to a sample $\mathbf{x}\_{i}$ and its corresponding sources $\mathbf{y}\_{i}$  (i.e., the decomposition of $\mathbf{x}_{i}$).
>
> 🔹 **Does it infer that the mixing function is not injective?** No. When $C = 1$, the observed data $\mathbf{x}$ is essentially a single-channel time series. If $T$ (the length of the time series, in our case $T = 256$) is sufficiently large relative to $n$ (the number of sources, in our case $n = 3$ and $n = 2$), the injectivity of the mixing function can still be preserved.
>
> 🔹 **Difference between $\mathbf{y}$ and $\mathbf{x}$ in Eq. 21:**
>
> We apologize for any confusion and have clarified the notation as follows:  $\mathbf{x}\_{i} \in \mathbb{R}^{C \times T}$ represents a single sample from the dataset, and we consider a set of $N$ samples across the entire dataset.  In our setting, there are $n$ sources $\mathbf{y} = \{y\_1, \dots, y\_{n}\}$, where each source $y\_k \in \mathbb{R}^{T}$ contribute to the mixed signal $\mathbf{x}$. Specifically, $\mathbf{x}$ is the sum of $y\_k$ at each timestamp $t \in \{1, \dots, T\}$.  The index $i$ in Eq. (2.2) refers to a sample $\mathbf{x}\_{i}$ and its corresponding sources $\mathbf{y}\_{i}$  (i.e., the decomposition of $\mathbf{x}\_{i}$).
>
> 🔹  **In Lines 144-145, what is $\mathcal{M}\_{+}^{1}(\mathcal{X})$?** By $\mathcal{M}\_{+}^{1}(\mathcal{X})$, we refer to the positive probability measure of the set $\mathcal{X}$. We have updated the text for further clarification. The notation $\mathbf{g}\_{\theta}♯p(\mathbf{z})$ refers to the transformation of a probability distribution $p(\mathbf{z})$ under a function $\mathbf{g}\_{\theta}$, often representing a model or transformation parameterized by $\theta$. This can be understood as the pushforward of the distribution $p(\mathbf{z})$ by the function $\mathbf{g}\_{\theta}$, where the distribution of $\mathbf{z}$ is transformed according to $\mathbf{g}\_{\theta}$. Apologies for any confusion.
>
> 🔹  **Line 162: recover?**
> As we have reconstructed the $n$ sources $\mathbf{y} = \{y_1, \dots, y_{n}\}$, we recover up to some noise from the mixed signal, thus $\mathbf{x}$.
>
> 🔹 **Definition 2.2:** Thank you for your feedback. Our definition is aligned with standard work on disentanglement and identifiability [3]. We have clarified the distinction between the two concepts and provided explicit definitions for both in **lines 193-198**. Please refer to the updated definition, as we have aimed to eliminate any potential confusion.
>
> 🔹  **The ratio $\frac{|\mathbf{\mu}\_{\theta\, k}(\mathbf{x})|}{\mathbf{\sigma}\_{\theta\, k}(\mathbf{x})}$** has been well discussed in lines 1124-1134, focusing on the impact on the sparsity of $\mathbf{\hat{z}}$.
>
> 🔹  **Assumption 4.1**  We would like to kindly clarify that our assumption is inspired by the work of Lachapelle [3] and the Structural Sparsity [4] approach. However, our focus is on pairs $(\mathbf{x}, \mathbf{x'})$ that share some active sources. The shared activation support is denoted by $\mathbf{i}$ (bold), and the union of all $\mathbf{i}$ defines the subset $\mathcal{I}$. The sufficiency is rejected based on the fact that for each factor $k$, we have enough pairs to cover all factors except $k$, denoted as $[n] \setminus k$. We provide further details and examples in Appendix A.5, where we also compare this approach with Structural Sparsity.
>
> **To ensure clarity, we have updated the section and made it more explicit. We hope this resolves the concern related to this.**
>
>
> We address other concerns in Part 2. Please refer to that section for further details.
>
>
> **References**
>
> - [1] Michalec et al., "Impact of harmonic currents on power quality," Energies, 14.12 (2021).
> - [2] Goldstein, "Auditory nonlinearity," JASA, 41.3 (1967).
> - [3] Lachapelle, et al. Nonparametric partial disentanglement via mechanism sparsity: Sparse actions, interventions and sparse temporal dependencies.
> - [4] Zheng, Y., et al. On the identifiability of nonlinear ICA: Sparsity and beyond. NeurIPS 2022

---

> ### Author Response · Authors · 2024-11-28
> **Section 2 - Metrics and Reproducible Code Provided**
>
> Dear Reviewer agV7,
>
> We truly appreciate your initial feedback. Below, we present our responses to Part 2 of your review, with updates based on your suggestions.
>
> ---
> ## Experiments
> ---
>
> 🔹 **Experiments in Table 1.1**
> We have added additional results, and Remark C.1 now clarifies that when we considered only 3 factors **{FR, THR, HTR}** (where, Fridge "FR", Dishwasher "DW", Washing Machine "WM", Heater "HTR", and Lighting "LT"), the results for S3VAE+HFS, C-DSVAE+HFS, SparseVAE, and TimeCSL were quite similar due to common signal combinations. However, when more factors **{FR, DW, WM, HTR, LT}** were included in training and testing, we observed distinct differences. These observations are thoroughly discussed in Table 5 of Appendix B.9.1.
>
> 🔹 **Experiments in Table 1.2 (MMC, $R^{2}$, Sparsity)**
> We greatly appreciate your comment. To eliminate any confusion, we have clarified that two versions of MCC are commonly used in the literature [1]. In our initial version, we reported the difference, which is why we stated "Lower is better". We have elaborated on both metrics for clarity:
> 1. The *strong MCC* refers to the value before alignment via the affine map $\Gamma$ (we provide a complete procedure in **appendix B.4.1** for this alignment).
> 2. The *weak MCC* refers to the value after alignment.
>
> In the updated version, we have chosen to present both the weak and strong MCC metrics together. We hope this clarification addresses your concern, and if it meets your satisfaction, we kindly request that you update your review.
>
> 🔹 **Why the 70%-20%-20% data split?**
> For clarity, our data split consists of 60% real data (with 10% augmentation), totaling 70% for training, and 20% each for testing and validation. This setup ensures efficient datasets for testing and validation, as explained in the 'Experimentation' section **(lines 401-405)**. We have updated the text to avoid any potential misunderstandings regarding the data split.
>
> 🔹 **Architecture of the Model:**
> The model architecture is provided in Figure 3 (page 7), with further implementation details available in Appendix B.5. We hope this answers your question. Please do not hesitate to reach out if you need additional clarification.
>
> 🔹 **Reproducing Our Results:**
> 1. We have added more clarified instructions for running the code and all requirements. Pre-trained model checkpoints are also provided in the zip file used in the paper with ``Reproducibility_Guidelines.md`` documentation.
>
> 2. **About our implementation** Thank you for your comment. You noted that some parts of the implementation seem based on image data rather than sequence data. To clarify, some components are adapted from sequential image ("video") disentanglement [2]. We've included a source map in Appendix B.1 (TimeCSL-Lib) to show the code we used to build our framework.
>
> ☑  The GMM-based VAE sampling is inspired by VaDE (Jiang et al., 2016), and
> we adapted the implementation from https://github.com/mperezcarrasco/
> Pytorch-VaDE.
>
> ☑ For the Diffusion model D3VAE (Li et al., 2023), we utilized the authors’ implementation from https://github.com/PaddlePaddle/PaddleSpatial/tree/main/research/D3VAE.
>
> ☑ TCL model was adapted from https:
> //github.com/hmorioka/TCL/tree/master/tcl, while the other models are
> derived from https://github.com/rpatrik96/nl-causal.
>
> ---
> ## Running the Code:
> ---
> - Please there is no ``main.py``, we use ``train.py``, and ``src_timecsl/evaluation.py`` for evaluation
> - The architecture of the network is depicted in Figure 3 in our paper, with the implementation available in ``/src_timecsl/models/timecsl.py``.
> - Run command:
> ```bash
> cd src_timecsl/
> python train.py --dataset_path "./datasets/data/ukdale.csv" --model_name "TimeCSL" --num_slots 5 --epochs 200 --use_generalization_loss True
> ```
>
> ---
>
> **Code Running in Terminal**
>
> ```python
> | Epoch | Valid LOSS | Valid MAE | Test LOSS | Test MAE |
> |-------|------------|-----------|-----------|----------|
> | 1     | 0.483      | 0.473     | 0.333     | 0.382    |
> | 2     | 0.456      | 0.461     | 0.331     | 0.379    |
> | 3     | 0.474      | 0.465     | 0.330     | 0.379    |
>
> (best val_loss: 0.456200, current val_loss: 0.474380)
> ````
> ---
> ---
>
> **Thank you for your feedback. We’ve addressed your concerns in the updated version and kindly ask you to reconsider the rating based on our clarifications. Please let us know if you have any further questions.**
>
> Thank you again for your time.
>
> Best, \
> Authors
>
> **References**
>
> - [1] Kivva, B. et al. Identifiability of deep generative models without auxiliary information, NeurIPS, 2022
> - [2] Li, Y., & Mandt, S. (2018). Disentangled sequential autoencoder. arXiv preprint arXiv:1803.02991.

---

### Official Review · Reviewer_Ubdd · 2024-11-04

**Soundness:** 2
**Presentation:** 2
**Contribution:** 3
**Rating:** 3
**Confidence:** 3

**Summary:**

The paper aimed at ensuring identifiability in time series representation learning by leveraging contrastive sparsity-inducing mechanisms. The authors address challenges in disentangling time series data by proposing a structured, sparsity-enforcing learning method that improves interpretability, robustness, and generalization, especially in source separation tasks.

**Strengths:**

1. **Theoretical Contributions**: The proposed framework is supported by theoretical insights that demonstrate the efficacy of contrastive sparsity in ensuring identifiable representations.

2. **Thorough Experimental Validation**: The paper offers a detailed experimental analysis, evaluating the proposed method on multiple datasets and providing insights into its performance under various settings.

**Weaknesses:**

1. **Unclear Problem Definition**: While the paper addresses time series representation learning, the data generation process presented is not inherently temporal (line 157). Furthermore, although the authors claim that their method accommodates "statistically dependent latent factors" (line 78), the source separation example provided involves independent sources, and dependent latent relationships are not clearly illustrated within the problem set. Without a well-defined problem scope and explicit assumptions, comparing the limitations with prior work becomes challenging.

2. **Unclear Theorem Proof**: A rigorous mathematical proof of identifiability, grounded in a clearly defined problem setting and set of assumptions, would strengthen the paper. Explicit derivations would provide the necessary foundation for understanding the theoretical claims presented.

3. ** Incomplete Manuscript:** Some information is missing in the main manuscript (lines 409, 502), and sections of the appendix (A3, A4, and C) appear unfinished or repetitive. This lack of completeness makes it difficult to thoroughly verify the experimental setup and interpret results accurately.

**Questions:**

1. Figure 1 is visually appealing but lacks sufficient detail to fully understand the concepts it illustrates. Could the authors provide a more comprehensive explanation of each component and its role within the model?
2. In line 76, the authors state that "sparsity alone is insufficient to ensure reliable identifiability, and thus, generalizability." Could you expand on this claim? What specific limitations of sparsity do previous studies identify in the context of identifiability?
3. Could the authors clarify the relationship between Y and Z? Is Y intended to represent the ground-truth sources (line 118) and Z the estimated latent variables (line 123)?
4. What data are provided as inputs to the model? In line 376, the tuples (x, y, x', y', i) are sampled. Are all these elements necessary in the model?
5. Since it is affine-wise identifiability, why use R2 instead of MCC as the evaluation metric?

---

> ### Author Response · Authors · 2024-11-29
>
> Dear Reviewer Ubdd,
>
> Thank you for your detailed review and constructive feedback. We appreciate the time you invested in evaluating our work and acknowledging its strengths. We have carefully addressed your concerns in the revised manuscript, which we have uploaded for your review.
>
> ---
> ### Addressing weaknesses :
> ---
>
> 🔹 **Independent vs Dependent Source Separation**
>
> We appreciate your question on this point. In our setting, we do not assume that sources are independent because, in practice, sources are often dependent. For instance, the operation of a heater and an air conditioner can be influenced by shared environmental factors such as temperature.
>
> 1. **Dependency in Time and Context**: Sources are often dependent on the time of usage. For example, in the kitchen, activating one appliance increases the likelihood of another being used shortly after.
> 2. **Approach**: Our methodology is designed to model and accommodate such dependencies, avoiding the assumption of independence on the prior.
>
> - We have clarified this in the **introduction** and added references to prior works addressing time-dependency, such as LEAP [1] and TDRL [2].
> ---
>
> 🔹 **Additional Comparisons with State-of-the-Art**
> In the revised manuscript, we expanded our comparison to include TDRL, iVAE, LEAP, TCL, and others, beyond our initial comparison to SlowVAE/DiffusionDisentangledVAE. This broader comparison enhances the paper’s contribution with a more comprehensive evaluation
>
> ---
>
> 🔹 **Provable Identifiability Results**
>
> You noted an issue with the missing proof in the initial submission. We have addressed this by formally stating the results in **Theorem 4.2**. Proof sketches are now provided in the main text, with complete proofs in **Appendix A.3** for further clarity.
>
> ---
>
> 🔹 **Appendix Manuscript Sections**
> We sincerely apologize for the oversight in our initial submission, where the incorrect appendix version was uploaded. The revised manuscript now includes the correct appendix and all the missing sections to ensure the completeness of the manuscript.
>
> ---
>
> ### 💬 Answers to questions
>
> ---
> 1. **Line 76: "Sparsity alone is insufficient to ensure reliable identifiability..."**
>    This is rooted in Darmois' theory (1953) [1] which shows that nonlinearity can cause unidentifiable representations even when sparsity the identifiability problem preexists, as sparsity does not alter the inherent properties that lead to unidentifiable representations. For example, the latent space $\boldsymbol{\hat{y}}\_{k}$ may depend on multiple latent slots $\mathbf{z}$, leading to uncertainties. We clarified this in **lines 287-289** of the revised manuscript.
>
> 2. **Relationship Between $\mathbf{y}$ and $\mathbf{z}$**
> You are correct. In our nonlinear ICA setting, $\mathbf{x}$ is generated from a latent space $\mathbf{z}$ via a nonlinear function $\mathbf{g}\_{\theta}$. For identifiability, each source $k$ is controlled by $\mathbf{z}\_{k}$: $\boldsymbol{y}\_{k} = \mathbf{g}\_{\theta, k}(\mathbf{z})$ and $\mathbf{x} = \sum\_{k=1}^{n} \mathbf{g}\_{\theta}(\mathbf{z})$. Accurately estimating the latent allows recovery of $\hat{\boldsymbol{y}}\_{k}$ (i.e, source $k$) summing to $\mathbf{x}$. $\mathbf{g}\_{\theta}(\mathbf{z})$ captures the full output, not just latent/slot $\boldsymbol{z}\_{k}$.
>
> 3. **Model Input and Framework**
>  Thank you for your question. We’ve added **Figure 4** to illustrate the framework. While labels for active sources are helpful, they are not strictly required. Instead, we use pairs $(\mathbf{x}, \mathbf{x'})$ with shared source activation. For example, in **Figure 2**, $\mathcal{S}(\mathbf{x}) = {1, 2, 4, 5}$ and $\mathcal{S}(\mathbf{x'}) = {2, 3, 4, 5}$, giving shared support indices $\mathbf{i} = {2,3,4,5}$.
>
> - Our approach can work in an unsupervised or semi-supervised manner (with a small labeled dataset), where we minimize $||\sum_{k=1}^{n}(\mathbf{\hat g}_{\theta k}(\mathbf{\hat z})) - \mathbf{x}||_2^2$.
> - In a fully supervised setting (with $\mathbf{y}$ available), we minimize $||\sum_{k=1}^{n}(\mathbf{\hat g}_{\theta k}(\mathbf{\hat z}) - \mathbf{y}_k)||_2^2$.
>
> This makes the approach more flexible, but labeled data can improve results.
>
> 4. **Evaluation Metrics**
>    We provided both strong MCC and weak MCC metrics in the original manuscript. Strong MCC refers to values before alignment via the affine map $\Gamma$, while weak MCC is measured after alignment. The procedure is detailed in **Appendix B.4.1**. We clarified this further in the revised text.
>
> ---
>
> **We hope these updates and clarifications address your concerns. If you find the revisions satisfactory, we kindly request that you consider updating your review.**
>
> ---
>
> ### References
>
> [1] G. Darmois. Analyse des liaisons de probabilité. In Proc. Stat, 1951.
> [1] Weiran et al. *Temporally Disentangled Representation Learning*. NeurIPS, 2022.
> [2] Weiran et al. *Learning Temporally Causal Latent Processes from General Temporal Data*. NeurIPS, 2021.

---

### Official Review · Reviewer_zkBJ · 2024-11-04

**Soundness:** 1
**Presentation:** 1
**Contribution:** 2
**Rating:** 3
**Confidence:** 3

**Summary:**

This paper proposes a method (TimeCSL) to obtain disentangled representations from high-dimensional time series data through Contrastive Sparsity-inducing Learning. They use Partial Selective Pairing as the contrastive objective, and train a modified VAE to obtain disentangled representations. The authors argue how this formulation improves the compositional generalization of the obtained representations. Experimentally, the paper shows the effectiveness of their formulation for the separation task.

**Strengths:**

1. The authors release a substantive set of pretrained baselines that could be useful for future research.
2. The showcased experimental results indicate that the proposed method TimeCSL outperforms the considered baselines.

**Weaknesses:**

## 1. Presentation and organization

The paper's key findings could be enhanced through a more structured organization and clearer presentation of the material. Apart from numerous grammatical/spelling errors (See Questions for a non-exhaustive list) that make it difficult to read the paper, there are several major issues due to the presentation.

### a. Unclear problem setting

I am very confused about the problem setting due to various mistakes/omissions in the presented notations.

a(i): What are the "unobserved States sources" $s_1, ..., s_5$ in Figure 1? There is no mention of these sources in the main text or appendix, however, they seem to be an important part of how the problem is being modeled.

a(ii): Lines 120-140 describe the details of a "VAE", however, the classic VAE reconstructs the input $x$ (hence, it is an *auto*-encoder). However, Equation (2.1) in the paper talks about the reconstruction of $y_i$, given $z$ and $x_i$. What is $y_i$ in this context? The notation $y_i$ was used to indicate the sources in line 118, but this does not make sense in the context of Equation (2.1). Similarly, what is $\mathcal{Y}$?

a(iii): Lines 170-173 are unclear. "$z$ must be generated by $g_\theta$", but $g_\theta$ does not generate $z$, it decodes $z$. The grammar errors in line 172-173 obfuscate the meaning entirely.

a(iv): The different "views" $x$ and $x'$ considered in the contrastive formulation are not explained. Are they different time-series samples?

Overall, it is not clear if this problem statement is the same as the source separation problem tackled in nonlinear ICA or not. If not, how?

### b. Organization of Section 4

b(i): What is $z$ vs $\hat{z}$? This is not adequately explained in Section 4. Similarly, in lines 263, what are $f_\phi$ and $\hat{f}_\phi$?

b(ii): Assumption 4.1 has missing details that make it hard to understand. What is $\mathcal{I}$? Does Equation (4.1) need to hold for any pair $x$, $x'$ or some pair? These details are unclear/missing.

b(iii): Lines 288-289: "according to Assumption 4.1, the sparsity-inducing nature ... existence of a source" - I don't see how this statement follows. What does the sparsity inducing nature mean in this context?

b(iv): It is unclear what the "claimed" theoretical contribution is. It would be beneficial to write a formal mathematical statement in the form of a theorem block and provide a proof.

b(v): The discussion about compositional generalization is difficult to follow. The first equation in Equation (4.6) doesn't quite make sense, since $g_\theta : \mathcal{X} \rightarrow \mathcal{Z}$, i.e. the image space of $g_\theta$ does not align with the domain space of $\hat{g}\_{\theta}$. Similarly, in line 356, I'm unsure how $\hat{g_\theta}(\hat{z}) \approx \hat{z}$.

b(vi): The optimization objective/algorithm is unclear. In equation (4.8), what are $z, z', \hat{z}, \hat{z}'$? Note that the latent variables $z$ appear in the objective of the VAE loss $\mathcal{L}_\text{VAE}$, and so without additional context, it is unclear how they are computed in the other terms. Additionally, what are the indices $\mathbf{i}$ and how are they calculated?


## 2. Unsubstantiated/Wrong Claims

There are several occasions in the paper where claims are unsubstantiated or (in some cases) wrong.

2(a): Lines 49-50: "The risk of ill-defined may lead to unstable and unreliable model outputs, where minor perturbations in data or hyperparameters can yield significantly different results upon retraining." - is this an observation by the authors? If so, I don't see the evidence presented in the paper. If it appears in prior work, then there should be a citation.

2(b):  Line 203-204: "This is the first identifiability study in real world of time series representation" - This seems like too strong a claim. The authors go on to cite papers that, in fact, tackle identifiability of time-series representations with real-world applications. Similarly, lines 195-196: "this work is the first to address identifiability and generalization in time series representations for separating sources in real scenarios" is wrong, see [1] for a method that does source separation with real-world applications.

2(c): Lines 209-210: "we place no assumptions on $p(z)$", however, in line 159, the authors assumed a particular form for $p(z)$, i.e. a GMM. This should be appropriately qualified in the text.

## 3. Missing Details

Several important details about the method/experimental settings are missing.

3(a): The details about the implementation of the loss-terms/neural network architectures used are not present.

3(b): Several subsections in the Appendix are empty (A.4, C).

3(c): The definitions of DCI/RMIG are not mentioned in the main text, but presented in the tables.


## References:
[1] Hyvärinen, Aapo, Ilyes Khemakhem, and Hiroshi Morioka. "Nonlinear independent component analysis for principled disentanglement in unsupervised deep learning." Patterns 4.10 (2023).

**Questions:**

1. What is a "latent slot"? This seems to be non-standard terminology. Do you mean latent dimension?

2. Line 159: "$z$ follows a Gaussian mixture model" - Can you provide an equation to show what the mixture components are, and an intuition as to why this assumption is useful?

3. In Line 144-145, what is $\mathcal{M}_+^1(\mathcal{X})$?

4. Line 257: "non-zero components of $\hat{z}$ and $\hat{z}'$", but they are technically not non-zero, rather they have a small magnitude as defined by the condition on the ratio of mean and variance.

5. The setting considered in Line 119 seems to indicate that the observed signal is a linear combination of unobserved sources. Why can't we use linear ICA? Would it make sense to include it as a baseline?

6. Comments on Figure 1.

6.a. The figure is not mentioned anywhere in the text

6.b. There are 4 OFF/ON views, but 5 state variables.

6.c. The numbering of the slots is inconsistent (1.1...1.5) and (1.2, ... 5.2).

6.d. What is "stop-gradient"?

6.e. The figure is cut off under the second view.

6.f. $x'$ is used in the caption, but $\tilde{x}$ is used in the figure.

7. Grammar/Spelling Errors

7.a. Line 43: weaker->weakly

7.b. Line 49: "risk of ill-defined <missing word?> may"

7.c. Line 77: "there <is> a need". "garantee" -> guarantee

7.d. Lines 94-95: "we propose a .... learning out-of-distribution data" -> incorrect grammar

7.e. Lines 104-105: $d$ is used in some places, $d_\mathcal{Z}$ in others.

7.f. Line 134: missing parenthesis around $x_i, y_i$.

7.g. Line 162: recouvering -> recovering

7.h. Line 173: "We give in Section 4.3. intuition and theoretical behind" -> incorrect grammar.

7.i. Line 182: extra )

7.j. Line 193: extra }

7.k. Line 195: "This work best of our knowledge.." -> incorrect grammar

7.l. Line 203: "As this is the first identifiability study in real world of time series representations" -> incorrect grammar

7.m. Line 257: "indicating that if <missing fragment>, then"...

7.n. Line 317: $k$ and $p$ were defined but not used correctly

7.o. Line 374: "leanred" -> learned

7.p. Line 377: "we show a how for" -> incorrect grammar

7.q. Line 404: "70/20/20 train/val/test split" does not sum to 100

7.r. Line 409: Missing reference (?)

7.s. Line 502: Missing reference (??)

**Details Of Ethics Concerns:**

The anonymous github README has a link to a HuggingFace repo that reveals the authors' affiliation and identity.

---

> ### Author Response · Authors · 2024-11-25
> **Part-1: Clarification on the problem setting and claims**
>
> Dear zkBJ,
>
> Thank you for your thoughtful and detailed feedback on our work. We greatly appreciate the time and effort you put into reviewing our paper and highlighting its strengths. Below, we provide a detailed response to each point raised:
>
> ### Clarification of the setting
> ----
> #### **a(i). Unobserved States Sources in Figure 1**
>
> We apologize for any confusion regarding Figure 1. The "unobserved states sources" refer to hidden components modeled as part of the generative process. To improve clarity, we will update the figure and revise the main text to ensure these components are properly defined and consistent with the modeling assumptions.
>
> ---
> #### **a(ii). Notation in Equation (2.1)**
> We apologize for any confusion and have clarified the notation as follows:
> - $\mathbf{x}_{i} \in \mathbb{R}^{C \times T}$ represents a single sample from the dataset, and we consider a set of $N$ samples across the entire dataset.
> - In our setting, there are $n$ sources $\mathbf{y} = \{y_1, \dots, y_{n}\}$, where each source $y_k \in \mathbb{R}^{T}$ contribute to the mixed signal $\mathbf{x}$. Specifically, $\mathbf{x}$ is the sum of $y_k$ at each timestamp $t \in \{1, \dots, T\}$.
> - The index $i$ in Eq. (2.2) refers to a sample $\mathbf{x}\_{i}$ and its corresponding sources $\mathbf{y}\_{i}$  (i.e., the decomposition of $\mathbf{x}_{i}$).
>
> We have updated the text to make this setting more explicit and hope this resolves the concerns raised.
>
> ---
>
> #### **a(iii). Lines 170-173 are unclear**
> We have clarified the formulation in this section. Specifically:
> - $\mathbf{z}$ is generated by $g_{\theta}^{-1}$.  Our decoder takes $\mathbf{z}$ as input and reconstructs the output $\mathbf{x}$.
>
> This updated explanation should make the relationship between $\mathbf{z}$ and $\mathbf{x}$ clearer.
>
> ---
>
> #### **a(iv). Views in the Contrastive Formulation**
> **Are $\mathbf{x}$ and $\mathbf{x'}$ different time-series samples?**  **Yes.** In the context of our method, "views" (see Fig. 2) represent different time-series samples used to train the model in a contrastive manner.
>
> We have clarified this in the revised version and linked it to the assumptions underlying our model, specifically:
> - **Partial Pairing** and support indices sharing.
> - $\mathbf{i}$ refers to the indices of sources active in both $\mathbf{x}$ and $\mathbf{x'}$.
>
> ---
>
> #### ✅ **Question: Relation to Nonlinear ICA (lines 123-127)**
> Thank you for your question. This problem is a source separation task, where the goal is to recover the sources $\mathbf{y} = \{y_1, \dots, y_{n}\}$ from the mixed signal $\mathbf{x}$. While the process involves summation, it remains nonlinear due to:
> - **Energy time series data (e.g., NILM):** Nonlinearities arise from interactions between multi-state appliances, power distortions, and continuous power fluctuations [1].
> - **Audio data:** Nonlinearities arise due to harmonic distortions and reverberations [2].
>
> ---
> ### Clarification of our claims
> ----
>
> **2(a): Lines 49-50 - Ill-defined risks leading to unstable and unreliable model outputs**
> Thank you for pointing this out. In the introduction, we mentioned a series of related works studying this issue. We have now added a more in-depth discussion to ensure the claim is properly supported.
>
> ---
> **2(b): Lines 203-204 -** "First identifiability study in real-world time series representation" -> "We clarify that our work focuses on enhancing disentanglement and generalization in time series, not claiming to be the first on identifiability". The phrasing has been revised to avoid ambiguity and properly attribute prior work. Relevant work [3], already discussed in the initial version, is now more explicitly highlighted. We appreciate your comment.
>
> ---
> **2(c): Lines 209-210 -** "We place no assumptions on..." vs. Line 159 (GMM assumption)
> - By "no assumption," we mean that we do not assume independence in the distribution (i.e. we do not assume independence of sources), as done in [3,4].
> - We clarify that the GMM prior is a weaker assumption, generalizable to exponential family mixtures [5] (Kivva et al.). Moreover, GMMs can approximate complex distributions [6], preserving the flexibility and generalization of Eq. (2.2). In our approach, we impose no constraints on: (1) ReLU architectures, (2) independence of $\mathbf{z}$ or (3) the complexity of the mixture model or neural network.
>
> See Part2
>
> **References**
> - [1] Michalec et al., "Impact of harmonic currents on power quality," Energies, 14.12 (2021).
> - [2] Goldstein, "Auditory nonlinearity," JASA, 41.3 (1967).
> - [3] Hyvärinen et al., "Nonlinear ICA for disentanglement in unsupervised deep learning," Patterns, 4.10 (2023).
> - [4] Hyvärinen et al., "Nonlinear ICA using auxiliary variables," AISTATS, 2019.
> - [5] Kivva et al., "Identifiability of deep generative models," NeurIPS, 2022.
> - [6] Nguyen & McLachlan, "On approximations via convolution-defined mixture models," Comm. Stat. Theory Methods, 2019.

---

> ### Author Response · Authors · 2024-11-25
> **Part-2: Details, Questions, and Claimed Theoretical of Section 4**
>
> Dear Reviewer zkBJ,
>
> Thank you once again for your valuable feedback. This is Part 2 (see below for Part 1), where we address the remaining points. We’re happy to answer any further questions. The revised version has already been uploaded, and we kindly request a reconsideration of the current score.
>
> ---
> ### Organization of Section 4
>
> ---
> #### **b(i): What is $ \mathbf{z} $ vs $ \hat{\mathbf{z}} $?**
> $ \hat{\mathbf{z}} $ refers to the learnable latent representation of the ground truth. While this was initially included in the notation section, we have now explicitly clarified it in Section 4.
>
> ---
> #### **b(ii): Assumption 4.1 Missing Details**
>
> Assumption 4.1 (Sufficient Partial Selective Pairing) ensures enough pairs $(\mathbf{x}, \mathbf{x'})$ share some sources except for a specific source $k$. Unlike stricter requirements in [1], our assumption allows overlaps, provided the union of shared indices spans all sources, enabling flexible modeling.
>
> ---
>
> #### **b(iii): Lines 288-289 (Sparsity-Inducing)**
> Assumption 4.1 implies that at least one source is inactive, naturally inducing sparsity. Support indices $ \mathbf{i} $ define active appliances in pairs $(\mathbf{x}, \mathbf{x'})$. We’ve clarified this further in the revised version.
>
> #### **b(iv): Claimed Theoretical Contribution**
> Thank you again for pointing this out. The proof was initially included in the appendix, but due to an oversight, it was not uploaded correctly  to OpenReview. We’ve now formally stated the results in Theorem 4.2, with proof sketches in the main text and complete proofs in the Appendix.
>
> ---
>
> #### **b(v): Compositional Generalization**.
>
> We clarified that $\hat{\mathbf{z}}$ and $\hat{\mathbf{z'}}$ are latents learned by the autoencoder $(\mathbf{\hat f}\_{\phi}, \mathbf{\hat g}\_{\theta})$ for $(\mathbf{x}, \mathbf{x'})$, respectively. The decoder enforces inversion of the encoder. Composing $\mathbf{z}\_{c}$ from $(\hat{\mathbf{z'}}, \hat{\mathbf{z}})$, decoding it via $\mathbf{\hat g}\_{\theta}$, and re-encoding with $\mathbf{\hat f}\_{\phi}$ must satisfy $\mathbf{\hat f}\_{\phi}(\mathbf{\hat g}\_{\theta}(\mathbf{z}\_{c})) = \mathbf{z}\_{c}$ to ensure compositional generalization, i.e., $\mathbf{g}\_{\theta}^{-1}\circ \mathbf{g}\_{\theta}(\mathbf{z}\_{c}) = \mathbf{z}\_{c}$. See lines 320–357 for details.
>
> ---
> ### b(vi): Optimization Objective/Algorithm
> The complete loss function and hyperparameters are now detailed in Section 4.1 for more clarity.
>
> ---
>
> ## 💬 Answers to Questions:
>
> 1) **What is a "latent slot"?**
> A "latent slot" refers to a latent vector in space, i.e., $ \mathbf{z} \in \mathbb{R}^{d \times n} $. This is defined in Section 2 and aligns with concepts familiar in the field of disentanglement, as discussed in (Locatello, 2020) [2] and (Wang, 2023) [3].
>
> 2) **Line 159: Gaussian Mixture Model (GMM)**
> - We’ve provided the full formulation in lines **162-169** and Algorithm 1 (Appendix 1.3) to clarify the components and their relevance.
>
> 3) **In Line 144-145, what is $\mathcal{M}\_{+}^{1}(\mathcal{X})$?** . By $\mathcal{M}\_{+}^{1}(\mathcal{X})$, we refer to the probability positive measure of the set $\mathcal{X}$. We have updated the text for more clarification, apologies for any confusion.
>
>
> 4) **Line 144-145, Line 257: Small Magnitude vs Zero Components**
> True, components may be near-zero due to the condition on the ratio of mean and variance in $ \log \sigma^{2}_\phi(\mathbf{x}_i) $. In practice, we observed some values à ~1e-4.  We have correctly discussed this in the revised version.
>
> ---
> 5) **Why not use linear ICA?**
> Thank you for your question. As mentioned in lines 125-128, nonlinearities such as distortions make linear ICA unsuitable. However, we have included nonlinear ICA baselines such as TCL, iVAE, SlowVAE, and TDRL.
>
> ---
>
> ## Comments on Figure 1 (Now Figure 2)
>
> - **6.a:** ✅ Please note that we have to make sure that the figure it well described in both section 4 and the introduction.
> - **6.b: There are 4 OFF/ON views, but 5 state variables** We note that the Noise contributes to the 5th state. This is clarified in the figure. ✅
> - **6.c:** Slot numbering corrected. ✅
> - **6.d: What is "stop-gradient"?** We have provided explanations for “Stop-gradient", in the main text and also in the figure.  ✅
> - **6.e:** Cropped figure fixed. ✅
> - **6.f:** Consistent variable use in the figure and caption. ✅
> - **7 Spelling Errors:**  We have thoroughly reviewed the text to address grammar and clarity issues.
> ---
>
> Thank you for your efforts and consideration. The revised version **has been uploaded** ✅, and we kindly request a reconsideration of the score. **We kindly ask that if our responses address your concerns, you consider updating your review to reflect the latest changes.**
>
> **Reference**:
>
> - [1] Hyvärinen et al., "Nonlinear ICA using auxiliary variables," AISTATS, 2019.
> - [2] Locatello, F., et al. "Object-centric learning with slot attention." NeurIPS, 2020
> - [3] Wang, Y. et al. Slot-vae ICML, 2023.

---

> > ### Comment · Reviewer_zkBJ · 2024-11-30
> >
> > Thanks for the clarifications and modifications to the paper. The authors have made numerous modifications and additions to the paper that improve the readability of the paper. Unfortunately, I still have major issues with  due to which I would like to maintain my score.
> >
> > My main concerns:
> >
> > 1.  Line 124-125: "Although the observed signal is a sum of sources, the mixing process is inherently non-linear due to interactions from multi-state appliances, power distortions, and continuously fluctuating power in NILM (Yue et al., 2020), similar to harmonic distortions and reverberations in audio (Lu et al., 2021)". I still don't see how the problem is "non-linear". The observed signal $x$ is a summation of signals $y_k = g_{\theta k}(z)$. The relationship between the observed signal $x$ and the unobserved sources $y_k$ is still linear. Mathematically, how do these non-linearities mentioned affect the source recovery process? Do the authors instead mean we are more interested in recovering the latent $z$ itself? If so, why?
> >
> > To add to this point, I would still argue that ICA is a worthwhile baseline to add if the authors can show that empirically, ICA cannot solve the problem considered in the paper.
> >
> > 2. The authors do not clarify how the support indices $\textbf{i}$ are computed in the loss terms (Eq 4.3). Theorem 4.2 simply states that if the cosine similarities between $\hat{z}$ and $\hat{z}'$ align with $\textbf{i}$, then $\hat{z}$ and $\hat{z}'$ are related by a permutation composed with element-wise invertible linear transformation. However, this theorem needs, as an "input", the support indices $\mathbf{i}$ on which $\hat{z}$ and $\hat{z}'$ agree with each other. It is unclear how these indices are computed during training, when the ground truth information about which sources are active is not present.
> >
> > 3. The proof of Theorem 4.2 contains several issues.
> > 	- There are numerous grammatical errors that make it difficult to follow the arguments.
> > 	- Lines 1036-Lines 1044 make an argument that is not mathematically precise since it is written out in words. Hence, it is difficult to assess the validity of the statements.
> > 	- Similarly, I am unsure about what the authors mean by "It is more likely to observe that:" and the following equation in Equation A.2.
> > 	- Line 1067: "Since both $g_\theta$ and $f_\phi$ are invertible linear functions", but they are not. They are assumed to be piecewise linear functions, which are not equivalent statements.
> > 	- In line 1075, I am not sure how the assertion $h_k$ is invertible linear transformation is made, using the fact that $h$ is a linear invertible transformation.
> > 	As such, I am not convinced by the validity of the theorem.
> >
> > Other minor concerns/questions:
> > 1. In Theorem 4.2, $f_{\phi}$ and $\hat{g}\_{\theta}$ are assumed to be continuous piecewise linear functions. This would preclude ReLU activations in Assumption 2.1 since the resulting decoders would not be invertible anymore. In fact, even with leaky ReLU activation, the assumption of invertible $g_\phi$ and $f_\theta$ is quite strong and unlikely to be true in practice.
> > 2. Why is the latent space of $z$ assumed to be a GMM, and why is the VaDE [1] formulation used instead of the regular VAE? This is not necessarily an issue with the approach, but since it is the non-standard choice, the authors should clarify their rationale.
> > 3. How is equation 4.5 related to 4.6? How does minimizing equation 4.5 ensure the compositional consistency? Also, how does ensuring the consistency of the latent on the in-distribution samples ensure that compositional consistency is maintained for OOD samples?
> > 4. Line 128-129 "Given a dataset of $N$ samples, denoted as $\{x_i, y_i \}_{i=1}^N$", but $y_i$ are not known a priori.
> > 5. In equation (4.7), $\mathcal{R}_{inv}$ takes $\mathbf{i}$ as input, but this is not the case in Equation (4.5)
> >
> > [1] Jiang, Z., Zheng, Y., Tan, H., Tang, B., & Zhou, H. (2016). Variational deep embedding: An unsupervised and generative approach to clustering. _arXiv preprint arXiv:1611.05148_.

---

### Author Response · Authors · 2024-11-29
**Updates**

Dear all reviewers,

Thank you for your invaluable suggestions. Based on your feedback, we have revised the paper to include additional discussions on related work, more comparison results, and other improvements. **The main changes are highlighted for each reviewer**. Below, we outline the key remarks and changes:

-  We have added Figure 3 to illustrate the entire framework, from the data preparation process to learning generalization. It also showcases the design of the model used.
-  We have included a dataset comparison between TimeCSL and baselines using the KITTI MOTS Challenge dataset, which demonstrates motion interpretability and shows that the proposed methods can be extended to a variety of time-series data. We provide an analysis of both Strong MCC and Weak MCC (after alignment), as well as the disentanglement metrics.

-  The link to the code and pre-trained models (huggingface weights) is available in the README file, which are also available in a zip file. Please note that it is kept anonymous for confidentiality purposes. You can access it on Hugging Face: https://huggingface.co/anonymousModelsTimeCSL/TimeCSL. We kindly remind reviewer **zkBJ** that the process is designed to preserve anonymity, and we hope this helps clarify any questions regarding our instructions in the code and the reproducibility of our results.

We sincerely appreciate the time and effort each reviewer has put into reviewing our manuscript. We believe we have addressed your constructive feedback, with a focus on enhancing the presentation and clarity of the paper.

We trust these updates will aid in evaluating our work more accurately. Please let us know if you have any further questions.

Thank you again.

---


Best regards,

The authors.

---

> ### Comment · Reviewer_zkBJ · 2024-11-30
> **Comment about Anonymity**
>
> I would like to clarify that in the previous version of the paper, the link in the "anonymous" GitHub repo in the paper had a README, which linked to the *author's personal Huggingface* account. This is an *oversight on the author's part*, not an attempt by me to subvert the anonymous review process. I would advise that the authors are more careful in their future submissions instead of blaming the reviewers for their own negligence.

---

> ### Author Response · Authors · 2024-11-30
> **Comment about Anonymity of the Code and README.md file**
>
> Thank you for your feedback. We would like to emphasize that we took all necessary measures to ensure anonymity during the submission process of the code. However, it appears there was an unexpected issue with the initial link, which was directed to the Huggingface account instead of the intended TimeCSL account. We are unsure how this redirection occurred, as it was not our intention, and we sincerely apologize for any confusion it may have caused. We appreciate your understanding and will double-check our processes in the future to avoid such occurrences.

---

### Note · Authors · 2025-01-22

**Comment:**

Dear all,

We deeply appreciate you taking the time to review our manuscript and share your thoughtful feedback. Your observations have provided valuable guidance on improving the clarity, relevance, and impact of our work. After reviewing. on your comments, we have decided to withdraw the paper to address your suggestions more thoroughly. This will allow us to refine our research and better highlight the significance of our work.

Thank you again for your time and for helping us strengthen our work.

**Withdrawal Confirmation:**

I have read and agree with the venue's withdrawal policy on behalf of myself and my co-authors.